# Doubly Outlier-Robust Online Infinite Hidden Markov Model

**Horace Yiu** [1 2] **Leandro Sánchez-Betancourt** [1 2] **Álvaro Cartea** [1 2] **Gerardo Duran-Martin** [2]

## Abstract

We derive a robust update rule for the online infinite hidden Markov model (iHMM) for when the streaming data contains outliers and the model is misspecified. Leveraging recent advances in generalised Bayesian inference, we define robustness via the posterior influence function (PIF), and provide conditions under which the online iHMM has bounded PIF. Imposing robustness inevitably induces an adaptation lag for regime switching. Our method, which is called Batched Robust iHMM (BR-iHMM), balances adaptivity and robustness with two additional tunable parameters. Across limit order book data, hourly electricity demand, and a synthetic high-dimensional linear system, BR-iHMM reduces one-step-ahead forecasting error by up to 67% relative to competing online Bayesian methods. Together with theoretical guarantees of bounded PIF, our results highlight the practicality of our approach for both forecasting and interpretable online learning.

## 1. Introduction

Non-stationarity is ubiquitous in fields such as finance (Stanley, 2003; Adams et al., 2019; Cartea et al., 2026), geology (Dastjerdy et al., 2023), speech recognition (Lee & Lee, 2006; Betkowska et al., 2007; Ondel, 2021), and continual learning (Liu et al., 2022; Pan & Yang, 2010). Here, the underlying data-generating process (DGP) may drift gradually, undergo abrupt regime changes, or be contaminated by outliers. Two crucial properties of the datasets that these DGPs produce are that (i) previously observed dynamics reappear, and that (ii) new regimes often emerge.

Popular approaches such as Bayesian changepoint detection (CPD) models (Adams & MacKay, 2007; Wilson et al., 2010; Fearnhead & Liu, 2007) and Kalman filters (KF)

[1]Mathematical Institute, University of Oxford [2]Oxford-Man Institute of Quantitative Finance. Correspondence to: Gerardo Duran-Martin < g.duran@me.com >.

*Proceedings of the $43^{rd}$ International Conference on Machine Learning*, Seoul, South Korea. PMLR 306, 2026. 

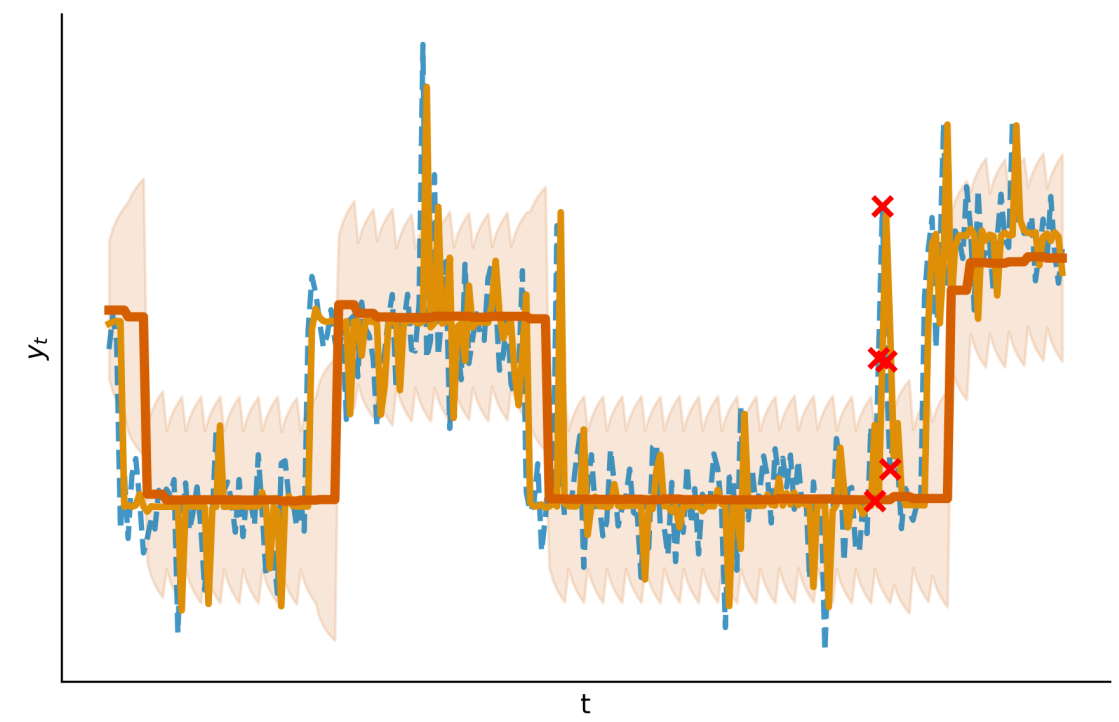

*Figure 1.* Data (blue dotted line) generated according to (139) with Student-$t$ observation noise. The outliers are marked with x. The solid red line shows our `BR-iHMM`, and the solid orange line shows the standard online iHMM. Regions around the posterior predictive mean cover $\pm 2$ standard deviations as error bounds.

(Welch et al., 1995) are able to handle non-stationarity, however, they typically reset or gradually forget the dynamics after a changepoint (CP) and cannot readily reuse previously encountered regimes. On the other hand, online infinite hidden Markov models (iHMM) are particularly well suited for such settings (Rodriguez, 2011). By maintaining a reusable library of regimes, iHMMs allow rapid returns to previously encountered dynamics while retaining the flexibility to discover new regimes when warranted.

However, this flexibility comes at a cost. Online iHMMs are highly sensitive to model misspecification and outliers: a single anomalous data point can corrupt the posterior over a regime's parameters, degrading future predictions even after the anomaly has passed. At the same time, the same outlier may trigger the creation of spurious regimes by falsely suggesting the presence of new dynamics. While state creation is appropriate when the data genuinely enter a new regime, outliers arising from sensor failures, corrupted measurements, or misspecified observation models lead to artificial states that harm interpretability and predictive performance (Stanley, 2003; Zimmerman, 1994; Liu et al., 2022; Fox et al., 2008b). Existing online iHMM formulations lack robustness in both the observation space and the latent state space.

This work addresses this gap. We introduce a provably doubly-robust online infinite hidden Markov model for online learning, forecasting, and regime detection. Our approach combines robust Generalised Bayes updates (Bissiri

et al., 2016) in the observation space with a novel batched inference mechanism that enforces robustness in the state space. To the best of our knowledge, this is the first online iHMM to provide robustness guarantees in both spaces simultaneously.

## 2. Related work

Infinite hidden Markov models belong to the broader class of hierarchical state-space models (HSSMs). Hidden Markov models (HMMs) are a widely studied subclass of HSSMs for discrete latent state spaces and provide a natural framework for modelling regime-switching dynamics (Baum & Petrie, 1966; Rabiner, 2002). The iHMM of Beal et al. (2001) extends HMMs by placing independent Dirichlet process (DP) priors over transition distributions, allowing an unbounded number of latent states. Teh et al. (2006) later formalised this construction as the hierarchical Dirichlet process (HDP), yielding explicit posterior Bayesian updates and enabling practical inference.

Existing online iHMM formulations typically rely on hybrid sequential Monte Carlo (SMC) methods such as particle learning (PL), Rao-Blackwellised particle filters (RBPF), or mixture Kalman filters (Doucet et al., 2000; Chen & Liu, 2000; Murphy & Russell, 2001; Carvalho et al., 2010). While effective for non-stationary data, these methods remain sensitive to outliers and model misspecification, often exhibiting rapid state switching or the creation of spurious regimes (Shah et al., 2006; Fox et al., 2007; Sgouralis & Pressé, 2017). Robust methods have previously been studied, but largely in isolation across different components of the model. State-space robustness has primarily been explored in offline iHMMs through modified priors or post-processing (Fox et al., 2008b; Johnson & Willsky, 2012; Van Gael et al., 2008), while observation-space robustness has been developed in robust variants of KFs and related linear state-space models (Welch et al., 1995; Karlgaard, 2015; Duran-Martin et al., 2024), as well as in Bayesian CPD models (Altamirano et al., 2023).

Our work connects these strands by providing a unified treatment of robustness in both the observation space and the latent state space within an online iHMM framework. A comprehensive review of related literature is provided in Appendix B.

## 3. Infinite Hidden Markov Models

We briefly review iHMMs and the main assumptions. Let $\boldsymbol{y}_{1:t} = \{\boldsymbol{y}_1, \ldots, \boldsymbol{y}_t\}$ with $\boldsymbol{y}_i \in \mathbb{R}^d$ denote a sequence of $d$-dimensional observations and $\boldsymbol{x}_{1:t} = \{\boldsymbol{x}_1, \ldots, \boldsymbol{x}_t\}$ with $\boldsymbol{x}_t \in \mathcal{X}$ a sequence of exogenous input features. The pair $\mathcal{D}_t = (\boldsymbol{x}_t, \boldsymbol{y}_t)$ is the data available at time $t$ and $\mathcal{D}_{1:t} = (\boldsymbol{x}_{1:t}, \boldsymbol{y}_{1:t})$. Predictive quantities for some future time $t' > t$ conditioned on $\mathcal{D}_{1:t}$ are indexed with the subscript $t'|t$.

### 3.1. Data generating process

In this work, we consider iHMMs with linear-Gaussian (LG) emissions.

Let $s_t \in \{1, 2, \ldots, t\}$ denote the latent state at time $t$. The latent state sequence $(s_t)_{t\geq 1}$, with base assignment $s_1 = 1$, evolves according to a transition probability

$$P(s_t \mid s_{t-1}) = \pi_{s_{t-1}, s_t}.$$

Let $\boldsymbol{\beta}_t = (\beta_1, \ldots, \beta_{t+1})$ denote the global state weights of an iHMM drawn from a stick-breaking process with concentration parameter $\gamma > 0$, inducing a DP prior over a countably-infinite set of latent states. For each state $s_t$, the state-specific transition probabilities are given by

$$\boldsymbol{\pi}_{s_t} = (\pi_{s_t,1}, \ldots, \pi_{s_t,t+1}) \sim \mathrm{Dir}(\alpha \boldsymbol{\beta}_t),$$

where $\alpha > 0$ is a concentration parameter that controls how strongly $\boldsymbol{\pi}_{s_t}$ is dispersed around $\boldsymbol{\beta}_t$.[1]

Given the latent state $s_t$, the exogenous covariates $\boldsymbol{x}_t$, and observation noise covariance $\mathbf{R}_t \in \mathbb{R}^{d\times d}$, the observation $\boldsymbol{y}_t$ is generated according to the LG model

$$P(\boldsymbol{y}_t \mid \boldsymbol{\theta}_{1:t}, s_t, \boldsymbol{x}_t) = \mathcal{N}(\boldsymbol{y}_t \mid f(\boldsymbol{x}_t)\, \boldsymbol{\theta}_{s_t},\ \mathbf{R}_t)\,, \quad (1)$$

where $\boldsymbol{\theta}_{1:t} = (\boldsymbol{\theta}_1, \ldots, \boldsymbol{\theta}_t)$ is the sequence of per-regime model parameters, $\boldsymbol{\theta}_{s_t} \in \mathbb{R}^m$ are the model parameters that correspond to state $s_t$, $f : \mathcal{X} \to \mathbb{R}^{d\times m}$ is a known feature transformation. Similar linear dynamical models are considered as iHMM observation models for its flexibility (Linderman et al., 2017) and are frequently used in time-series forecasting and regression (Ghahramani & Hinton, 2000; Fox et al., 2008a). Although we focus on linear models, our framework is general and accommodates a wide range of emission models.[2]

### 3.2. Online infinite hidden Markov models

Let $s_{1:t}$ be the state path up to time $t$, that is,

$$s_{1:t} = (s_{1:t-1}, s_t) \in \{1\} \times \{1, 2\} \times \ldots \times \{1, 2, \ldots, t\}.$$

For $t \geq 1$, Bayesian online inference for iHMMs involves recursive estimation of the joint posterior,

$$P(\boldsymbol{\theta}_{1:t}, s_{1:t} \mid \mathcal{D}_{1:t}) = \underbrace{P(\boldsymbol{\theta}_{1:t} \mid s_{1:t}, \mathcal{D}_{1:t})}_{\text{observation posterior}} \underbrace{P(s_{1:t} \mid \mathcal{D}_{1:t})}_{\text{path posterior}}.$$

for $t \geq 1$. These two factors are studied in Sections 3.3 and 3.4, respectively.

[1] A measure-theoretic HDP construction is in Appendix C.1, or alternatively, the work in Teh et al. (2006).

[2] See Rodriguez (2011) for instance, where emission model is specified as an exponential-family.

### 3.3. Sequential inference for linear-Gaussian models

Conditioned on a latent state path $s_{1:t}$ and data $\mathcal{D}_{1:t}$, the posterior over observation parameters $\boldsymbol{\theta}_{1:t}$ factorises such that only the parameter associated with the active state $s_t$ is updated. Under the linear-Gaussian (LG) model, the posterior admits the factorisation

$$P(\boldsymbol{\theta}_{1:t} \mid s_{1:t}, \mathcal{D}_{1:t}) = \prod_{k=1}^{t} p(\boldsymbol{\theta}_k \mid \boldsymbol{\Psi}_{t,k}), \tag{2}$$

where $\boldsymbol{\Psi}_{k,t} = (\boldsymbol{\mu}_k, \boldsymbol{\Sigma}_k)$ denotes the sufficient statistics associated with state $k$ at time $t$.

Given a datapoint $\mathcal{D}_t$ and Gaussian prior $P(\boldsymbol{\theta}_t \mid \boldsymbol{\Psi}_{t,t-1}) = \mathcal{N}(\boldsymbol{\mu}_0, \boldsymbol{\Sigma}_0)$, the posterior can be updated recursively as

$$P(\boldsymbol{\theta}_{1:t} \mid s_{1:t}, \mathcal{D}_{1:t}) \propto P(\boldsymbol{y}_t \mid \boldsymbol{\theta}_{s_t}, \boldsymbol{x}_t) \prod_{k=1}^{t} P(\boldsymbol{\theta}_k \mid \boldsymbol{\Psi}_{k,t-1}).$$

This expression shows that the observation $\boldsymbol{y}_t$ affects only the posterior for the active state $s_t$; all other state parameters are carried forward unchanged.

For $k \neq s_t$, we set $\boldsymbol{\Psi}_{k,t} = \boldsymbol{\Psi}_{k,t-1}$, while the sufficient statistics of the active state $s_t$ are updated according to the equations

$$\begin{aligned}
\hat{\boldsymbol{y}}_{s_t} &\leftarrow f(\boldsymbol{x}_t)\, \boldsymbol{\mu}_{s_t}, \\
\mathbf{S}_{s_t} &\leftarrow f(\boldsymbol{x}_t)\, \boldsymbol{\Sigma}_{s_t}\, f(\boldsymbol{x}_t)^\top + \mathbf{R}_t, \\
\mathbf{K}_t &\leftarrow \boldsymbol{\Sigma}_{s_t}\, f(\boldsymbol{x}_t)^\top\, \mathbf{S}_{s_t}^{-1}, \\
\boldsymbol{\mu}_{s_t} &\leftarrow \boldsymbol{\mu}_{s_t} + \mathbf{K}_t(\boldsymbol{y}_t - \hat{\boldsymbol{y}}_{s_t}), \\
\boldsymbol{\Sigma}_{s_t} &\leftarrow \boldsymbol{\Sigma}_{s_t} - \mathbf{K}_t\, \mathbf{S}_{s_t}\, \mathbf{K}_t^\top.
\end{aligned} \tag{3}$$

The resulting posterior predictive distribution is given by

$$\begin{aligned}
&P(\boldsymbol{y}_{t+1} \mid \mathcal{D}_{1:t},\, s_{1:t+1},\, \boldsymbol{x}_{t+1}) \\
&= \int P(\boldsymbol{y}_{t+1} \mid \boldsymbol{\theta}, \boldsymbol{x}_{t+1})\, P(\boldsymbol{\theta} \mid \boldsymbol{\Psi}_{t,s_{t+1}})\, d\boldsymbol{\theta} \\
&= \mathcal{N}(\boldsymbol{y}_{t+1} \mid \hat{\boldsymbol{y}}_{s_{t+1}}, \mathbf{S}_{s_{t+1}}).
\end{aligned} \tag{4}$$

### 3.4. Sequential inference for HDPs

Next, estimation of the path posterior $s_{1:t}$ takes the form

$$\begin{aligned}
P(s_{1:t} \mid \mathcal{D}_{1:t}) \propto P(\boldsymbol{y}_t \mid \mathcal{D}_{1:t-1}, s_{1:t}, \boldsymbol{x}_t) \\
P(s_t \mid s_{1:t-1})\, P(s_{1:t-1} \mid \mathcal{D}_{1:t-1})\,.
\end{aligned} \tag{5}$$

The first probability in (5) is defined in (2). The second probability is the posterior predictive transition probability from the terminal state $s_{t-1}$, which only depends on the path $s_{1:t-1}$ through the HDP sufficient statistics $\boldsymbol{\Phi}_{t-1}$. Specifically,

$$\begin{aligned}
&P(s_t \mid s_{1:t-1}) \\
&= P(s_t \mid s_{t-1}, \boldsymbol{\Phi}_{t-1}) \\
&= \int_{\Delta^{t-1}} \pi_{s_{t-1},s_t}\, \mathrm{Dir}(\boldsymbol{\pi}_{s_{t-1}} \mid \hat{\alpha}_{t-1}\, \hat{\boldsymbol{\beta}}_{t-1})\, d\boldsymbol{\pi}_{s_{t-1}} \\
&= \frac{n_{s_{t-1},s_t,t-1} + \hat{\alpha}_{t-1}\hat{\beta}_{s_t,t-1}}{\sum_\ell n_{s_{t-1},\ell,t-1} + \hat{\alpha}_{t-1}}\,,
\end{aligned} \tag{6}$$

where $\boldsymbol{\Phi}_t = (\hat{\alpha}_t, \hat{\boldsymbol{\beta}}_t, \hat{\gamma}_t, L_t, \mathbf{N}_t)$ are the *structural parameters* that summarises the path $s_{1:t}$ into sufficient statistics of the HDP. Here, $(\hat{\alpha}_t, \hat{\boldsymbol{\beta}}_t, \hat{\gamma}_t)$ estimates $(\alpha, \boldsymbol{\beta}, \gamma)$ respectively, $L_t$ is the number of unique states instantiated in $s_{1:t}$, and $\mathbf{N}_t \in \mathbb{N}^{t\times t}$ is the transition count matrix such that $[\mathbf{N}_t]_{\ell k} = n_{\ell,k,t}$ is the number of times transitioned from state $k$ to $\ell$ and is a function of $s_{1:t}$. The final equality in (6) follows from Dirichlet–multinomial conjugacy. A full derivation of (6), as well as the posterior updates of the structural parameters $\boldsymbol{\Phi}_t$ given $s_t$ and $\boldsymbol{\Phi}_{t-1}$, are provided in Appendix C.2.[3]

## 4. The Doubly-robust Online Infinite Hidden Markov Model

This section introduces our novel Batched Robust iHMM (BR-iHMM). We show that robustness in online iHMMs is inherently a two-fold problem, requiring control of both the observation-parameter posterior and the latent-state posterior. Addressing only one of these aspects is insufficient. We therefore introduce a doubly-robust online iHMM that combines robust observation updates with robust state inference, yielding provable guarantees against outlier contamination.

### 4.1. Quantifying robustness

We formalise robustness with respect to outliers in terms of posterior sensitivity. An *outlier* is defined as an observation $\boldsymbol{y}_t^c \in \mathbb{R}^d$ that is not generated according to the assumed DGP and whose magnitude may be arbitrarily large. Robustness is assessed by measuring the influence such an outlier has on posterior distributions.

Following Matsubara et al. (2022); Altamirano et al. (2023); Duran-Martin et al. (2024), we quantify this influence using the *posterior influence function* (PIF), defined as the Kullback–Leibler divergence (KLD) between posteriors updated with and without contamination. Let $\mathcal{D}_t^c = (\boldsymbol{x}_t, \boldsymbol{y}_t^c)$ and $\mathcal{D}_{1:t}^c = (\mathcal{D}_{1:t-1}, \mathcal{D}_t^c)$. We define the PIFs for the latent state path and the observation parameter as

$$\begin{aligned}
\mathrm{PIF}_{s_t}(\boldsymbol{y}_t^c, \boldsymbol{y}_{1:t}) &= \mathrm{KL}(P(s_{1:t} \mid \mathcal{D}_{1:t}) \,\|\, P(s_{1:t} \mid \mathcal{D}_{1:t}^c))\,, \\
\mathrm{PIF}_{\boldsymbol{\theta}_t}(\boldsymbol{y}_t^c, \boldsymbol{y}_{1:t}) &= \mathrm{KL}(P(\boldsymbol{\theta}_t \mid s_{1:t}, \mathcal{D}_{1:t}) \,\|\, P(\boldsymbol{\theta}_t \mid s_{1:t}, \mathcal{D}_{1:t}^c))\,.
\end{aligned}$$

[3] Or alternatively, Section 5.3 of Teh et al. (2006) for the HDP posterior updates.

A posterior is said to be *outlier-robust* if its PIF is uniformly bounded, i.e.

$$\sup_{\boldsymbol{y}_t^c \in \mathbb{R}^d} \mathrm{PIF}(\boldsymbol{y}_t^c, \boldsymbol{y}_{1:t}) < \infty.$$

Assuming that $\boldsymbol{\theta}_t$ has continuous support, the PIF of the joint posterior over $(\boldsymbol{\theta}_t, s_{1:t})$ admits the additive decomposition (see Appendix D.3)

$$\begin{aligned}&\mathrm{PIF}_{\boldsymbol{\theta}_t, s_t}(\boldsymbol{y}_t^c, \boldsymbol{y}_{1:t}) \\ &= \mathrm{PIF}_{s_t}(\boldsymbol{y}_t^c, \boldsymbol{y}_{1:t}) + \sum_{s_{1:t}} P(s_{1:t} \mid \mathcal{D}_{1:t})\, \mathrm{PIF}_{\boldsymbol{\theta}_{s_t}}(\boldsymbol{y}_t^c, \boldsymbol{y}_{1:t}).\end{aligned} \tag{7}$$

Equation (7) shows that robustness of the joint posterior requires boundedness of *both* the observation-space PIF and the state-space PIF.

**Batched posterior influence.** In online regime-switching settings, a single outlier should be insufficient to justify the creation of a new regime. To capture robustness against short sequences of anomalous observations, we extend the definition of the PIF to batches of size $B \geq 1$. Let $\boldsymbol{y}_{t+1:t+B}^c$ denote a batch of $B$ potentially contaminated observations, and let $\mathcal{D}_{1:t+B}^c$ denote the corresponding contaminated dataset. We define the *batched PIF* as the KLD between posteriors updated with $\mathcal{D}_{1:t+B}$ and $\mathcal{D}_{1:t+B}^c$.

A model is said to be *batch-robust of order* $B$ if the batched PIF is uniformly bounded for all contaminations of size $B$. This notion allows us to formalise a robustness–adaptivity trade-off: larger values of $B$ increase robustness to transient outliers at the cost of slower detection of genuine regime changes. We verify this empirically in the appendix with Figures E.12 and E.10 by comparing detection delay of our model on various batch sizes.

### 4.2. Robustness in observation space (WoLF)

We first address robustness in the observation space, that is, the sensitivity of the posterior over observation parameters $\boldsymbol{\theta}_t$ to extreme observations. For standard LG models, it is well known that non-robust Bayesian updates yield unbounded posterior influence: a single outlier can arbitrarily distort the posterior over $\boldsymbol{\theta}_t$ (Peña et al., 2009; Desgagné & Gagnon, 2019).

Since Bayesian updating of LG models can be seen as a special case of KFs, existing work on robust KFs can be directly leveraged to make robust updates (Lambert et al., 2022). In this work, we adopt the weighted observation likelihood filter (WoLF) (Duran-Martin et al., 2024), which replaces the standard likelihood with a weighted likelihood in a generalised Bayesian update (Bissiri et al., 2016). Thus, conditioned on a latent state path $s_{1:t}$, the generalised posterior update for the active state's parameter takes the form

$$\begin{aligned}&P(\boldsymbol{\theta}_{1:t} \mid s_{1:t}, \mathcal{D}_{1:t}) \\ &\propto P(\boldsymbol{y}_t \mid \boldsymbol{\theta}_{s_t}, \boldsymbol{x}_t)^{W(\boldsymbol{y}_t, \hat{\boldsymbol{y}}_{s_t})^2} \prod_{k=1}^{t} p(\boldsymbol{\theta}_k \mid \boldsymbol{\Psi}_{k,t-1}).\end{aligned} \tag{8}$$

where $W \colon \mathbb{R}^d \times \mathbb{R}^d \to \mathbb{R}$ is a bounded weighting function and $\hat{\boldsymbol{y}}_{s_t}$ denotes the state-specific predictive mean.

To ensure bounded observation-space influence, the weight function is required to satisfy

$$\sup_{\boldsymbol{y} \in \mathbb{R}^d} W(\boldsymbol{y}, \hat{\boldsymbol{y}}) < \infty, \qquad \sup_{\boldsymbol{y} \in \mathbb{R}^d} W(\boldsymbol{y}, \hat{\boldsymbol{y}})^2 \, \|\boldsymbol{y}\|_2^2 < \infty, \tag{9}$$

for any $\hat{\boldsymbol{y}} \in \mathbb{R}^d$. Under these conditions, WoLF yields a bounded observation-space PIF. Moreover, under the LG assumption, the weighted update preserves conjugacy and admits closed-form recursive updates, which is essential for PL. In what follows, we use the inverse-multiquadratic (IMQ) weight

$$W(\boldsymbol{y}, \hat{\boldsymbol{y}})^2 = \frac{1}{1 + c^{-2} \|\boldsymbol{y} - \hat{\boldsymbol{y}}\|_{\mathbf{R}_t}^2}, \tag{10}$$

where $c > 0$ is a soft-threshold parameter, $\mathbf{R}_t$ is the observation noise covariance, and $\|\cdot\|_{\mathbf{R}_t}$ denotes the Mahalanobis norm. We choose IMQ specifically as it is a convenient representation of the admissible class of weighting functions. With the Mahalanobis norm, the weight is scale-aware with respect to $\mathbf{R}_t$, which is known in the LG model, so it measures residual size in the natural geometry of the observation model rather than in raw coordinates.[4] We denote by $w_{s,t'|t}$ the corresponding weight assigned to observation $\boldsymbol{y}_{t'}$ relative to the prediction produced by state $s$ at time $t$.

With WoLF, the update is modified by replacing the posterior predictive variance in (3) with

$$\mathbf{S}_{s_t} \leftarrow \begin{cases} f(\boldsymbol{x}_t) \boldsymbol{\Sigma}_{s_t} f(\boldsymbol{x}_t)^\top + \mathbf{R}_t / w_{s_t,t|t-1}^2 & \sum_\ell n_{\ell,s_t,t} > 0 \\ f(\boldsymbol{x}_t) \boldsymbol{\Sigma}_{s_t} f(\boldsymbol{x}_t)^\top + \mathbf{R}_t & \text{otherwise} \end{cases} \tag{11}$$

which limits changes to the posterior mean $\boldsymbol{\mu}_{s_t}$ of any existing state $s_t$ if prediction error $\|\boldsymbol{y}_t - \hat{\boldsymbol{y}}_{s_t}\|_{\mathbf{R}_t}^2$ is large and $w_{s_t,t|t-1}^2$ is close to 0.

**Observation-space robustness is insufficient.** While WoLF guarantees bounded $\mathrm{PIF}_{\boldsymbol{\theta}_t}$, this alone does not ensure robustness of the joint posterior. Even when observation updates are robust, outliers can still induce pathological behaviour in the latent state posterior.

**Theorem 4.1** (Observation-space robustness alone is insufficient)**.** *Consider an online iHMM with linear-Gaussian*

[4] For other admissible weighting functions, see West (1981); Duran-Martin et al. (2024) and references therein.

*emissions. Suppose that, conditional on any fixed latent state path $s_{1:t}$, the observation parameters are updated using WoLF with a weight function satisfying* (9). *Then the joint* $\mathrm{PIF}_{\boldsymbol{\theta}_t, s_t}$ *is unbounded, as the state-space* $\mathrm{PIF}_{s_t}$ *is unbounded.*

The proof is in Appendix D.4. Intuitively, as the prediction error $\|\boldsymbol{y}_t - \hat{\boldsymbol{y}}_t\|^2$ grows, the posterior increasingly favours the creation of a new state whose prior variance penalises the error less severely than existing states. As a result, the state posterior degenerates onto paths that instantiate a new regime at time $t$, leading to an unbounded $\mathrm{PIF}_{s_t}$.

This result shows that robustness in observation space, while necessary, is not sufficient. In the next subsection, we introduce a complementary mechanism that directly controls robustness in the state space.

### 4.3. Robustness in state space (batched inference)

Observation-space robustness alone does not prevent outlier-driven state creation. Even when the likelihood contribution of an extreme observation is downweighted, a single anomalous point can still dominate state inference by favouring a path that introduces a new regime. To control this effect, we introduce a complementary mechanism that enforces robustness directly in the state space.

The key idea is to infer latent states over short batches of observations rather than at every individual time step. Instead of deciding whether to switch regimes based on a single data point, the model accumulates evidence over a batch of size $B > 1$ before allowing a state transition. This pooling prevents a single outlier from making a switching path dominate the posterior, and thus bounds the influence of extreme observations on state inference.

**Batched state inference.** Following Duran-Martin et al. (2025), we consider generalised posteriors over state trajectories. For a candidate extension of the state path over the next batch, $s_{t+1:t+B}$, we define a batched (log) posterior score that approximates (5) by aggregating predictive likelihoods across the batch,

$$
\begin{aligned}
&\log \nu(s_{1:t+B};\, \mathcal{D}_{1:t+B}) \\
&= \sum_{b=1}^{B} w^2_{s_{t+b},\, t+b|t} \log P(\boldsymbol{y}_{t+b} \mid \mathcal{D}_{1:t+b-1}, \boldsymbol{x}_{t+b},\, s_{t+b}) \\
&\quad + \log \sum_{s_{1:t}} P(s_{1:t} \mid \mathcal{D}_{1:t}) P(s_{t+1} \mid s_t, \boldsymbol{\Phi}_t) \\
&\qquad \prod_{b=2}^{B} \mathbb{1}(s_{t+b-1} = s_{t+b}).
\end{aligned} \tag{12}
$$

The first term pools the WoLF-weighted predictive likelihoods over the batch, while the second term accounts for the transition into the batch. The indicator enforces persistence within the batch, so only paths that remain in a single state over the batch are admissible.

Intuitively, an isolated outlier contributes little to the pooled likelihood due to its small weight, while the remaining uncontaminated observations stabilise the posterior. As a result, switching paths cannot dominate unless there is consistent evidence across multiple observations.

**Degenerate sticky HDP prior.** We implement intra-batch persistence via a time-dependent degenerate sticky HDP prior to achieve the approximation in (12). Specifically, we modify the state-transition distribution to

$$
\boldsymbol{\pi}_{s_t} \mid \boldsymbol{\Phi}_t \sim \mathrm{Dir}(\hat{\alpha}_t \hat{\boldsymbol{\beta}}_t + \boldsymbol{e}_{s_t} \kappa_t), \tag{13}
$$

where $\boldsymbol{e}_{s_t} \in [0,1]^{t+1}$ is the standard basis vector corresponding to state $s_t$ and $\kappa_t \geq 0$ is a self-transition bias. We define

$$
\kappa_t = \begin{cases} 0, & t \equiv 1 \pmod{B}, \\ \infty, & \text{otherwise}, \end{cases} \tag{14}
$$

and interpret $\kappa_t \in \{0, \infty\}$ in the limiting sense. At inter-batch times ($t \equiv 1 (\mathrm{mod}\ B)$), the prior reduces to the standard HDP, allowing regime changes or the creation of new states. At intra-batch times, $\kappa_t = \infty$ forces self-transitions almost surely, preventing switching within the batch.

This deterministic scheduling yields what we refer to as the *batched mechanism*. State inference is performed only at inter-batch times, and the states within each batch are propagated deterministically.

**State-space robustness.** The batched mechanism directly bounds the state-space PIF. Because switching is only permitted after aggregating $B$ observations, a single extreme observation cannot induce a new state. Figure E.5 illustrates this effect: while the state-space PIF grows unbounded in the non-batched case, batching yields a uniformly bounded PIF even under large perturbations. This mechanism also mitigates the rapid-switching pathology of iHMMs, where redundant states with similar posteriors alternate frequently due to reinforcement by the HDP prior (Fox et al., 2008b). By construction, intra-batch alternation is prohibited, improving prequential stability.

**Doubly-robust inference.** Combining WoLF-based observation updates with batched state inference yields robustness in both spaces. Figure 2 illustrates the graphical model. We formalise this result below.

**Theorem 4.2** (Joint observation–state space robustness)**.** *Let* $W(\cdot,\cdot)$ *satisfy the boundedness conditions of Theorem 4.1. The batched inference posterior in* (12)*, together with the degenerate sticky HDP prior in* (13)*, yields a*

**Algorithm 1** BR-iHMM

**Input:** Number of particles $N$, ESS threshold $\tau_{\text{ESS}}$, initial $\boldsymbol{\theta}_0^{(i)}, s_0^{(i)} \sim \pi^{(i)}, \omega_0^{(i)} \leftarrow 1/N$, batch size $B > 1$, max number of states before prune MAX_STATES
**for** $t < T$ **do**
  **for** each particle $i$ **do**
    Predict $\hat{\boldsymbol{y}}_{t+1:t+B}^{(i)} = \hat{\boldsymbol{y}}_{s_t^{(i)},\, t+1:t+B|t}$
    $\boldsymbol{w}_{l,t}^{(i)} \leftarrow [W(\boldsymbol{y}_{t+b}, \hat{\boldsymbol{y}}_{l,t+b|t})]_{b=1}^{B}$
    $\omega_{t+B}^{(i)} \leftarrow P(\boldsymbol{y}_{t+1:t+B} \mid \boldsymbol{\Phi}_t^{(i)}, \boldsymbol{\Psi}_t^{(i)}, \{\boldsymbol{w}_{l,t}^{(i)}\})$
  **end for**
  **if** ESS $\leq \tau_{\text{ESS}}$ **then**
    Resample $(\boldsymbol{\Phi}_t^{(i)}, \boldsymbol{\Psi}_t^{(i)}, \boldsymbol{\psi}_t^{(i)}, \mathbf{N}_t^{(i)}) \sim \text{Mult}(\{\omega_{t+B}^{(i)}\})$
    Reset $\omega_{t+B}^{(i)} \leftarrow 1/N$
  **else**
    Normalize $\omega_{t+B}^{(i)} \leftarrow \omega_{t+B}^{(i)} / \sum_j \omega_{t+B}^{(j)}$
  **end if**
  **for** each particle $i$ **do**
    $\hat{s}_{1:t+B}^{(i)} \leftarrow$ sample from (12)
    **if** $s_{t+B}^{(i)}$ is a new state **then**
      $L_{t+B}^{(i)} \leftarrow L_t^{(i)} + 1$
    **end if**
    $\mathbf{N}_{t+B}^{(i)} \leftarrow \mathbf{N}_t^{(i)} + \text{counts}(\hat{s}_{t+1:t+B}^{(i)})$
    Prune and update (see Algorithm 8)
    **for** $b = 1$ **to** $B$ **do**
      $\mathbf{M}_{t+b}^{(i)} \sim \text{Antoniak}(\mathbf{N}_{t+B}^{(i)}, \hat{\alpha}_{t+b-1}^{(i)}, \hat{\boldsymbol{\beta}}_{t+b-1}^{(i)})$
      **if** $b > 1$ **then**
        $m_{j,j,t+b}^{(i)} \leftarrow 0 \quad \forall j$
      **end if**
      $\boldsymbol{\Phi}_{t+b}^{(i)} \leftarrow \texttt{update_struct}(\boldsymbol{\Phi}_{t+b-1}^{(i)}, \mathbf{M}_{t+b}^{(i)})$
      $\boldsymbol{\Psi}_{t+b}^{(i)} \leftarrow \texttt{WoLF_update}(\boldsymbol{\Psi}_{t+b-1}^{(i)}, \boldsymbol{y}_{t+b}, w_{\hat{s}_{t+b}, t+b|t}^{(i)})$
    **end for**
  **end for**
  $t \leftarrow t + B$
**end for**
**Return:** $\{\hat{\boldsymbol{y}}_{1:T}^{(i)}, \hat{s}_{1:T}^{(i)}, \boldsymbol{\Psi}_{1:T}^{(i)}, \boldsymbol{\Phi}_{1:T}^{(i)}, \omega_T^{(i)}\}_{i=1}^{N}$

*bounded state-space PIF. Consequently, the joint PIF is bounded.*

The proof is in Appendix D.5.

**Implementation.** Following Rodriguez (2011), we implement the online iHMM via PL. We use the superscript $(i)$ to indicate quantities associated with the $i$-th particle. The particle-specific posterior predictive of the next state is conditioned on $s_{1:t-1}^{(i)}$ to recover the form in (6). Following standard practice in SMCs, we incorporate a threshold $\tau_{\text{ESS}}$; particles are resampled whenever the effective sample size (ESS) falls below this value (Doucet et al., 2001). The formula for ESS is given in Appendix E.1. Particle weights are initialised uniformly with $\omega_0^{(i)} = N^{-1}$ for $N > 0$ particles. The classical online iHMM algorithm via PL is presented in Algorithm 6 in Appendix C.3.

The pseudocode in Algorithm 1 summarises the resulting doubly-robust online iHMM, where wolf_update refers to the update rules in (3) and (11), and update_struct refers to those in (32)-(37) in Appendix C.2. In Algorithm 1, we sample auxiliary variables $\mathbf{M}_t \in \mathbb{N}^{t \times t}$ with $[\mathbf{M}_t]_{l,k} = m_{l,k,t}$. We write its distribution as $\mathbf{M}_t \sim \text{Antoniak}(\mathbf{N}_t, \alpha_{t-1}, \boldsymbol{\beta}_{t-1})$ if its elements follow the probability law (31) in the appendix (Antoniak, 1974).

### 4.4. Computational efficiency

Beyond robustness, the batched mechanism yields substantial computational benefits. In a standard online iHMM, transitions are permitted within a batch of size $B$, the number of admissible paths grows exponentially in $B$, which is infeasible in online settings (Lee & Lee, 2006).

The degenerate sticky HDP prior introduced in Section 4 approach avoids this complexity by enforcing intra-batch state persistence. As a result, state sampling is required only once per batch. Proposition D.1 formalises the resulting computational simplifications. In particular, the transition counts are updated only once per batch, while other structural parameters are resampled conditionally. The full algorithm (Algorithm 7) and proof (Appendix D.1) are in the appendix.

**Scalability.** In long data streams, the number of instantiated states may grow unbounded. To ensure scalability, we cap the number of active states to a fixed maximum and prune stale regimes using a heuristic based on usage frequency and recency. When a state is pruned, it is permanently removed from the bookkeeping counts and the global state weights. Formal details and proofs are provided in Algorithm 8 and Proposition D.2 in the appendix.

## 5. Experiments

We evaluate the proposed BR-iHMM model on four datasets covering online forecasting and segmentation tasks. Our primary objective is to assess one-step-ahead predictive accuracy, measured by root mean squared error (RMSE); results are summarised in Table 1. We additionally evaluate regime segmentation performance using standard detection metrics (Table 2). For simplicity, we initialise $\boldsymbol{\theta} \sim \mathcal{N}(\boldsymbol{\mu}_0, \mathbf{I}_m)$, where $\mathbf{I}_m \in \mathbb{R}^{m \times m}$ is the identity. Initially, $\boldsymbol{\mu}_{s_t}$ samples from the same prior as $\boldsymbol{\theta}$, whereas $\boldsymbol{\Sigma}_{s_t} = \boldsymbol{\Sigma}_{(0)}$ is a known initial covariance matrix. We also initialise $\hat{\alpha}_0$ and $\hat{\gamma}_0$ with Gamma priors as in Escobar & West (1995) to allow conjugate posterior sampling, which we further simplify with an uninformative initialisation, i.e., $\hat{\alpha}_0, \hat{\gamma}_0 \sim \text{Gam}(1, 1)$.

Across experiments, we compare three online iHMM variants with increasing levels of robustness: (i) the original HDP-iHMM (iHMM), (ii) an iHMM with observation-space robustness via WoLF updates (WoLF-iHMM), and (iii) our batched state-space robust variant (BR-iHMM). We also include two baselines: the offline beam-sampling iHMM (offline-iHMM) (Van Gael et al., 2008), and Bayesian online changepoint detection (BOCD), which enforces a left-to-right, non-revisiting regime structure. For segmentation experiments, we further compare against the robust BOCD method of Altamirano et al. (2023) (DSM-BOCD) and the use of student-t posterior predictives with LGs of unknown-variance (iHMM (unknown-var)).

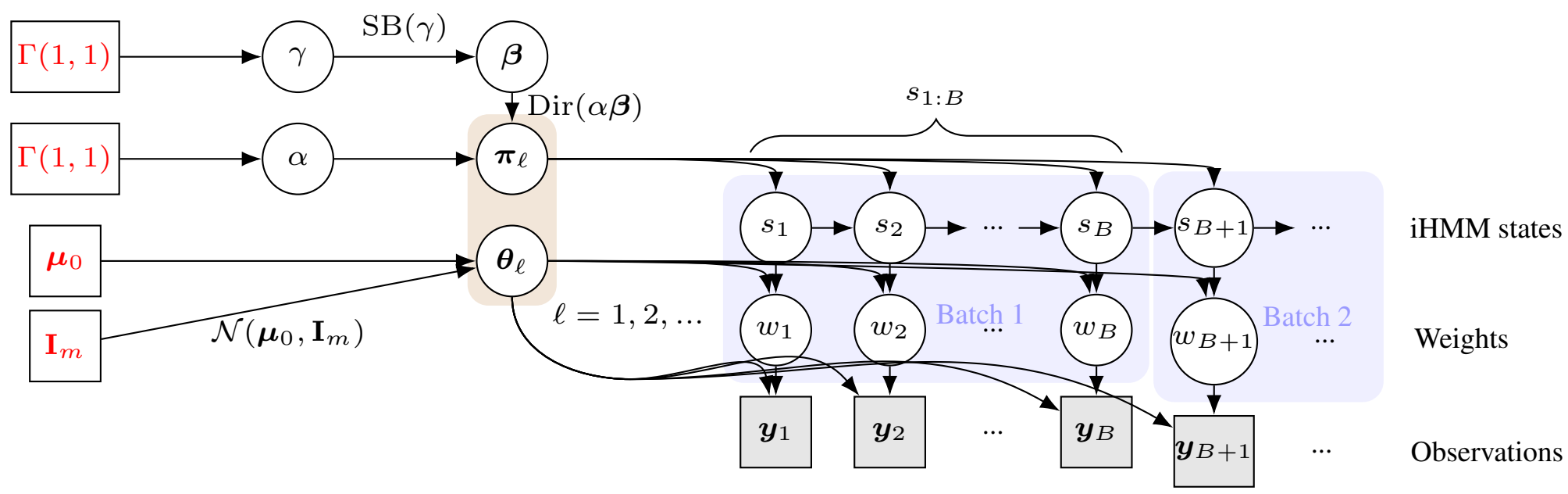

*Figure 2.* Bayesian architecture of the BR-iHMM. Variables in a circle are updated during inference. Red variables are fixed hyperparameters. For instance, the priors of $\gamma$ and $\alpha$ are both uninformed $\Gamma(1,1)$ for all our experiments.

Table C.4 in Appendix C.4 summarises the hyperparameters, which we optimise via `bayesian-optimization` on a training partition (Nogueira, 2014; Garrido-Merchán & Hernández-Lobato, 2020). In particular, the choice of $B$ is application-dependent and fixed a priori as a hyperparameter. For details on fitted hyperparameters, see in Table E.7 in the Appendix. Each experiment is repeated 100 times using the same hyperparameters. We implement the models in JAX, and run the experiments on `NVIDIA GeForce RTX 3090` with `256GB RAM`. For `offline-iHMM`, we port the MATLAB code (Gael, 2009) and run for 1000 iterations.

| Model | Synthetic | Electricity | OFI |
|---|---|---|---|
| BR-iHMM | **46.1±0.00301** | **0.47±0.04** | **0.616±0.082** |
| WoLF-iHMM | 103.8±0.01155 | 0.63±0.03 | 0.623±0.089 |
| iHMM | 101.7±0.02636 | 0.57±0.03 | 0.620±0.080 |
| offline-iHMM | 2.9 | 0.32 | 0.552 |
| BOCD | 123.12 ± 0.01365 | 0.80 ± 0.11 | 0.733 |

*Table 1.* One-step ahead RMSE (Mean ± stdev) over 100 runs. The most precise online model is bolded.

**Synthetic data: Online linear regression with regime changes.** We construct a synthetic online regression task with regime switching and extreme outliers to evaluate robustness in high-dimensional settings. The data consist of 2500 observations with a scalar response driven by a $d = 100$-dimensional, regime-dependent parameter vector. This setup reflects applications such as factor models in finance and online continual learning.

The DGP has three latent regimes with strong self-persistence (99.5% self-transition probability). Observations follow a linear model with heavy-tailed noise, and with probability 1% are replaced by extreme outliers sampled uniformly from $[-600, 600]$. Although the response is univariate, the regression depends on a high-dimensional feature vector, making the task particularly sensitive to spurious regime creation.

Forecasting performance is reported under the "Synthetic" column of Table 1. Figure 3 visualises rolling RMSE and inferred MAP states. BR-iHMM rapidly stabilises after a short calibration period, recovers the true number of regimes, and achieves forecasting accuracy comparable to the offline beam sampler. In contrast, both iHMM and WoLF-iHMM exhibit severe state fragmentation, creating more than 30 spurious regimes driven by outliers. This instability leads to pronounced RMSE spikes at outlier locations and prevents convergence to the true regimes.

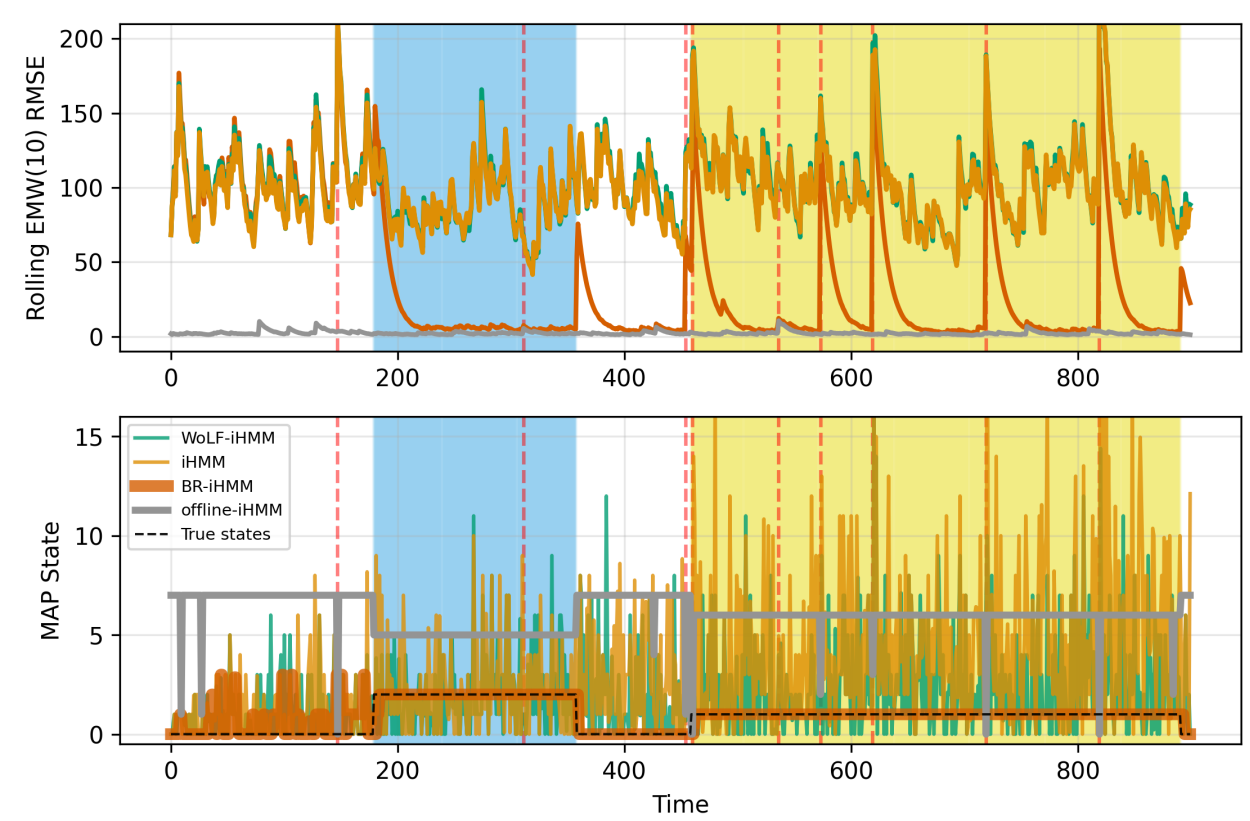

*Figure 3. Synthetic linear data.* Outliers are marked by **red** vertical dashes. **Yellow** and **blue** regions correspond to regimes 2 and 3. First subplot tracks rolling RMSE of mean predictions across the 100 runs. Bottom plot shows average MAP state. First 900 predictions are plotted. The full Figure E.11 is in the appendix.

State stability is critical in this setting: by suppressing outlier-driven regime creation, BR-iHMM maintains consistent state assignments and yields substantially more accurate predictions. While `offline-iHMM` also achieves low RMSE, it creates additional states that are primarily assigned to outliers, is not robust to extreme observations, and is offline (see lower panel of Figure 3).

**Hourly electricity demand.** We evaluate the proposed methods on a real-world hourly electricity demand dataset from Farrokhabadi (2020), spanning 2017–2020 and covering both pre-pandemic and pandemic periods. The task is one-step-ahead forecasting of electricity load demand (`Load (kW)`), using weather-related covariates. The

dataset contains 31 912 observations; hyperparameters are optimised on the first 12 000 points.

We model electricity demand using a regime-dependent LG regression (see (142) in the appendix). Figure E.15 compares rolling RMSE (top panel) and inferred latent regimes (bottom panel) for BR-iHMM, iHMM, and WoLF-iHMM. The top panel shows that BR-iHMM achieves uniformly lower rolling RMSE, with the largest improvements occurring at periods of elevated volatility. These gains are consistent with the results reported in the "Electricity" column of Table 1. In the bottom panel, BR-iHMM identifies regime transitions beginning in March 2020, aligning with the onset of COVID-19. In contrast, both iHMM and WoLF-iHMM predominantly remain in a single regime over this period, indicating limited sensitivity to structural change. Together, these results show batching improves predictive accuracy and the inference of latent structures.

**Order flow imbalance.** We evaluate the proposed methods on a limit order book (LOB) dataset for Microsoft (MSFT) from March 2020, and forecast order flow imbalance (OFI) in a prequential setting. We replicate the experimental setup of Tsaknaki et al. (2025), where trades are aggregated into fixed-size buckets and scalar observations $y_t$ are produced by summing signed volumes within the partition. The procedure used to construct the observations from the LOB and the DGP we assume is in Appendix E.10.

Figure 4 reports cumulative RMSE for one-step-ahead predictions. In this setting, hyperparameter optimisation selects a batch size of $B = 1$ for BR-iHMM, effectively recovering the WoLF-iHMM variant. This indicates that larger batch sizes degrade performance by forcing consecutive minutes into the same latent state, while regime changes in (141) can occur rapidly and are not primarily driven by outliers.

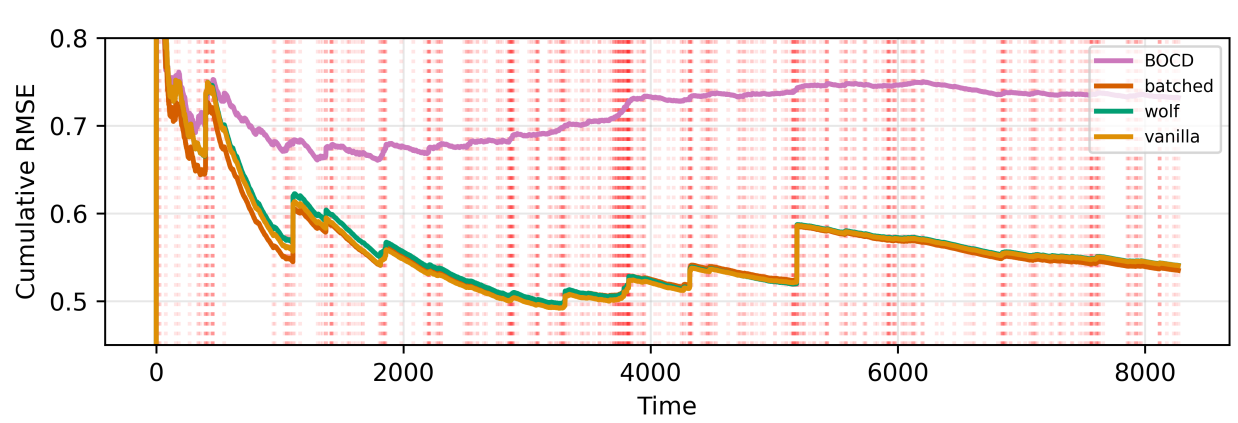

*Figure 4. Cumulative RMSE for OFI predictions.* offline-iHMM is not plotted to allow zooming into the difference between the online models. Vertical dotted red lines indicate an outlier (defined as outside 1.5 × inter-quartile range).

These results highlight an important property of BR-iHMM: batching is not imposed indiscriminately. When state persistence is inappropriate, the method adapts by selecting $B = 1$, avoiding unnecessary smoothing. Quantitative results are reported under the "OFI" column of Table 1.

**Segmentation: Well-log.** The well-log dataset consists of 4050 measurements and is first studied by Ó Ruanaidh & Fitzgerald (1996); it is a popular benchmark for inference methods on non-stationary data (Fearnhead & Clifford, 2003; Fearnhead & Liu, 2007). Studies usually remove outliers caused by transient geological events, but robustified methods such as ours and Altamirano et al. (2023) leave them in. We assume a trivial i.i.d. Gaussian–Gaussian emission. Hyperparameters are trained on the first 1700 datum. Unlike other experiments, we use a lookahead mean absolute error objective to account for outliers preceded by gradual deviations rather than isolated spikes.

Simulations are plotted in Figure E.9. BR-iHMM is clearly more resistant to the spikes between observations 1000-1500. The predictive advantage of reverting to a previously encountered state manifests at short regimes, such as those near $t = 2500$. In contrast, DSM-BOCD requires a recalibration period. In MAP state, iHMM and WoLF-iHMM both exhibit rapid state-switching in regions that were clearly stationary. DSM-BOCD was the most stable, with the exception of missing the CP at $t = 1500$, whereas all iHMMs correctly detected a new regime. offline-iHMM predictions were closest to the outliers, showing even offline models are not outlier-robust. Furthermore, a popular tactic in robust-statistics is to replace the likelihood function or a posterior predictive with heavy-tailed distributions which is more lenient to outliers. We demonstrate with iHMM (unknown-var) that such tactic does not prevent the outlier-driven state-creation pathology stated in Theorem 4.1.

**Runtimes.** Table E.6 in Appendix E summarises the average runtime of each model. BR-iHMM is faster as stated in Proposition D.1. BR-iHMM takes the least time to compute on equal machinery against iHMM and WoLF-iHMM. The speed-up is significant for the electricity exercise.

**Detection delay (DD).** The batched mechanism inherently means that there is a delay in detecting CPs if the CP is intra-batch. A CP may be assigned to the previous state under batching, producing incorrect predictions for the mismatched states. We evaluate the DD empirically with our synthetic data by comparing the positive predictive value (PPV), true positive rate (TPR), and DD (formulas in Appendix E.8). Table 2 shows BR-iHMM is more accurate in regime detection at the expense of a small DD. WoLF-iHMM and iHMM outperform offline-iHMM with smaller DD due to rapid state transitions.

Furthermore, Figures E.10 and E.12 confirm the intuition that larger batch sizes increase detection delay as the CP has larger chance to fall within a batch.

| Model | PPV | TPR | Delay |
|---|---|---|---|
| BR-iHMM | **0.963 ± 0.001** | **0.996 ± 0.001** | 3.4 ± 0.0 |
| WoLF-iHMM | 0.533 ± 0.002 | 0.344 ± 0.002 | **0.0 ± 0.0** |
| iHMM | 0.551 ± 0.002 | 0.363 ± 0.008 | 0.2 ± 0.0 |
| offline-iHMM | 0.996 | 0.997 | 0.0 |

*Table 2.* Values are averaged over 100 runs for online models. Higher PPV and TPR is better, lower DD is better.

## 6. Limitations

Our approach has several limitations. First, the batch size $B$ is fixed a priori and dataset-dependent, motivating future work on adaptive or data-driven selection. Second, the theoretical guarantees rely on LG emissions to obtain closed-form PIF expressions, the analysis to more general emission models remains open. Third, while bounded PIF ensures stability of the posterior under contamination, its precise relationship to predictive performance is not yet characterised. Finally, we do not provide an analytical characterisation of the adaptivity–robustness trade-off induced by $B$, and instead rely on empirical evidence to illustrate this effect. A more detailed discussion on the limitations is in Appendix F, where we also propose possible treatments to infer $B$ online, and provide general statements to bounding PIFs for non-LG distributions.

## 7. Conclusion

We introduced a doubly-robust online iHMM that controls sensitivity to outliers and remains efficient for online prequential settings. By integrating generalised Bayes updates in the observation model with batched state-inference, we showed that robustness at either level alone is insufficient to ensure bounded joint influence. Our contribution is not the introduction of new components, but their principled integration, we show that combining standard tools (WoLF updates, HDP priors, and particle learning) in this specific manner is necessary to achieve bounded joint PIF. The superior predictive performance ability of our BR-iHMM is demonstrated across a wide range of forecasting exercises. We find that our model is especially advantageous when the exogenous input is high-dimensional.

## Impact statement

This paper presents work whose goal is to advance the field of machine learning. There are many potential societal consequences of our work, none of which we feel must be specifically highlighted here

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

## A. Notation

We denote variables belonging to the *i*-th particle with the superscript $(\cdot)^{(i)}$. Furthermore, for any time-dependent matrix $\mathbf{X}_t$, such that $[\mathbf{X}_t]_{ij} = x_{i,j,t}$, we introduce the shorthand

$$
\begin{aligned}
x_{i,\cdot,t} &= \sum_j x_{i,j,t}, \\
x_{\cdot,j,t} &= \sum_i x_{i,j,t}, \\
x_{\cdot,\cdot,t} &= \sum_{i,j} x_{i,j,t},
\end{aligned} \tag{15}
$$

as the marginal sum of the $i$-th row, the $j$-th column, and the sum of all elements of $\mathbf{X}_t$ respectively. For any time-independent matrix $\mathbf{X}$ with elements $[\mathbf{X}]_{ij} = x_{i,j}$, the marginal sums are defined similarly without the time index. Let $\lambda_{\max}(\mathbf{X})$ be a function that maps a square matrix $\mathbf{X}$ to its largest eigenvalue, and $\det \mathbf{X}$ be its determinant. Let $\mathbf{A}, \mathbf{B}$ be two square matrices. We write $\mathbf{A} \succ 0$ if $\mathbf{A}$ is a positive definite, and $\mathbf{A} \succeq 0$ for positive semi-definiteness. The relation $\succ$ and $\succeq$ is a non-strict partial order that is asymmetric, transitive, and reflexive. If $\mathbf{A} - \mathbf{B}$ is positive definite, then $\mathbf{A} \succ \mathbf{B}$, and similarly $\mathbf{A} \succeq \mathbf{B}$ for the positive semi-definite case. We denote $\mathbb{1}(\cdot)$ as the indicator function which evaluates to 1 if its argument is true, and 0 otherwise.

Furthermore, as per Duran-Martin et al. (2025), we adopt the notational shorthand $\boldsymbol{\psi}_t = s_{1:t} = (\boldsymbol{\psi}_{t-1}, t)$ for convenience. In the supplementary material, we extend the double subscript notation $\ell, t$ to distinguish between the value of a regime-specific estimate of state $\ell \in \{1, ..., t\}$ conditioned on different data. Indeed, we need to distinguish the value at a previous time $t-1$ and its updated posterior value at time $t$ when we prove results that reference the same quantity before and after a posterior update.

**Model assumptions** We now reiterate the assumptions of our linear-Gaussian (LG) setup. For any state $\ell$, we assume a positive-definite initial covariance matrix $\boldsymbol{\Sigma}_{\ell,0} = \boldsymbol{\Sigma}_{(0)} \succ 0$. In summary, the LG model is defined as

$$
\boldsymbol{y}_t = \mathbf{F}_t \boldsymbol{\theta}_t + \boldsymbol{r}_t \, .
$$

where $\mathrm{Var}(\boldsymbol{r}_t) = \mathbf{R}_t$, and the matrices $\mathbf{R}_t \in \mathbb{R}^{d \times d}, \mathbf{F}_t \in \mathbb{R}^{d \times m}$ are real-valued and finite. The feature transformation of the input $\boldsymbol{x}_t \in \mathcal{X}$ given the feature map $f(\boldsymbol{x}_t)$ is morally the same object as $\mathbf{F}_t$, but we introduce the latter notation to make clear it is a matrix.

In this work, we refer to $\mathbb{R}^m$ as the parameter space, such that emission parameters $\boldsymbol{\theta}$ takes values in $\mathbb{R}^m$. We collect the sufficient statistics statistics at time $t$ for the observation posterior $\boldsymbol{\Psi}_t = \{\boldsymbol{\Psi}_{\ell,t}\}_\ell$ where $\boldsymbol{\Psi}_{\ell,t} = (\boldsymbol{\mu}_{\ell,t}, \boldsymbol{\Sigma}_{\ell,t})_\ell$, and the path posterior $\boldsymbol{\Phi}_t = \{\hat{\alpha}_t, \hat{\boldsymbol{\beta}}_t, \hat{\gamma}_t, L_t, \mathbf{N}_t\}$ respectively.

| Symbol | Description |
|---|---|
| $\|\cdot\|_2$ | $\ell_2$ norm. |
| $\|\cdot\|_{\mathbf{R}_t}$ | Mahalanobis norm with respect to $\mathbf{R}_t$. |
| $\mathcal{N}(\boldsymbol{y} \mid \boldsymbol{\mu}, \boldsymbol{\Sigma})$ | Multivariate Gaussian density evaluated at $\boldsymbol{y}$ with mean $\boldsymbol{\mu}$ and covariance $\boldsymbol{\Sigma}$. |
| $\mathrm{DP}(\gamma, H)$ | Dirichlet process with base measure $H$ and $\gamma > 0$ |
| $\mathrm{Dir}(\boldsymbol{\pi} \mid \boldsymbol{\beta})$ | Dirichlet density evaluated at $\boldsymbol{\pi}$ with base prior $\boldsymbol{\beta}$. |
| $SB(\gamma)$ | Stick breaking distribution with concentration $\gamma$ |
| $\mathrm{Mult}(\boldsymbol{n} \mid \boldsymbol{\pi})$ | Multinomial density evaluated at $\boldsymbol{n} \in \mathbb{N}^m$ with discrete likelihood $\boldsymbol{\pi} \in [0,1]^m$. |
| $\delta_a$ | Dirac measure concentrated at $a$, unit probability mass on $a$ and zero elsewhere. |
| $\mathbf{I}_d \in \mathbb{R}^{d\times d}$ | $d$-dimensional identity. |
| $y_t \in \mathbb{R}$ | Scalar observations at time $t$. |
| $\boldsymbol{y}_t \in \mathbb{R}^d$ | Observations at time $t$ with dimension $d \in \mathbb{N}$. |
| $\boldsymbol{x}_t \in \mathcal{X}$ | Inputs at time $t$. |
| $\boldsymbol{u}_{1:t} = \{\boldsymbol{u}_1, ..., \boldsymbol{u}_t\}$ | Sequence of vectors ordered sequentially through time. |
| $\mathcal{D}_t = (\boldsymbol{x}_t, \boldsymbol{y}_t)$ | Datapoint at time $t$. |
| $\ell \in \mathbb{N}$ | Indicator for state index. |
| $(\Omega, \mathcal{A})$ | Measurable space with universe $\Omega$ and $\sigma$-algebra $\mathcal{A}$. |
| $H^*$ | Latent probability measure on $(\Omega, \mathcal{A})$ in the iHMM's DGP. |
| $\gamma > 0$ | Latent concentration parameter for global prior's Dirichlet Process in the iHMM's DGP. |
| $\boldsymbol{\xi} = (\xi_1, \xi_2, ...)$ | Infinite-dimensional stick weights sampled from $\mathrm{SB}(\gamma)$. |
| $\boldsymbol{\Xi}_t = \{A_1, ..., A_t, A_{t+1}\}$ | Partition on $\mathbb{R}^d$ to obtain the finite-dimensional approximation $\boldsymbol{\beta}_t$ |
| $\boldsymbol{\beta}_t = (\beta_1, \beta_2, ..., \beta_{t+1})$ | Latent global prior over the distribution of countably infinite states in the iHMM's DGP. |
| $\alpha > 0$ | Latent concentration parameter for Dirichlet process of state-specific transition probabilities in the iHMM's DGP. |
| $\boldsymbol{\pi}_\ell = (\boldsymbol{\pi}_{\ell 1}, \boldsymbol{\pi}_{\ell 2}, ...)$ | Latent transition probabilities from state $l$ in the iHMM's DGP. |
| $\boldsymbol{\theta}_\ell \in \mathbb{R}^m$ | Random samples from $H$, parametrising the observation model corresponding to state $l$ in the iHMM's DGP. |
| $\boldsymbol{\Theta} = \{\boldsymbol{\theta}_1, \boldsymbol{\theta}_2, ...\}$ | Collection of all parameters in the iHMM's DGP observation models. |
| $s_t \in \{1, ..., t\}$ | Latent state at time $t$ in the iHMM's DGP. |
| $s_{1:t} = (s_1, ..., s_t)$ | Latent state trajectory up until time $t$ with base assignment $s_1 = 1$ (random variable). |
| $f : \mathcal{X} \to \mathbb{R}^{d\times m}$ | feature transformation that takes in values from the input space $\mathcal{X}$ and maps to a linear projection for $\boldsymbol{\theta} \in \mathbb{R}^m$. |
| $\mathbf{F}_t = f(\boldsymbol{x}_t) \in \mathbb{R}^{d\times m}$ | Projection matrix in the SPN model, is morally the same object as $f(\boldsymbol{x}_t)$. |
| $\mathbf{R}_t \in \mathbb{R}^{d\times d}$ | Covariance matrix for observation noise in the SPN model. |
| $\boldsymbol{r}_t \in \mathbb{R}^d$ | Zero-mean observation noise vector with known covariance $\mathbf{R}_t$ in the SPN model. |
| $\boldsymbol{\mu}_{\ell,t} \in \mathbb{R}^d$ | Estimated posterior mean of Gaussian observation model for state $l$ at time $t$. |
| $\mathbf{P}_{\ell,t} \in \mathbb{R}^{d\times d}$ | Estimated posterior covariance of Gaussian observation model for state $l$ at time $t$. |
| $\boldsymbol{u}_{t'\|t}$ | Predictive value of the quantity $\boldsymbol{u}$ at time $t'$ given sufficient statistics at time $t$. |
| $\hat{\boldsymbol{\beta}}_t, \hat{\boldsymbol{\alpha}}_t, \hat{\boldsymbol{\psi}}_t$ | Posterior estimates of $\boldsymbol{\beta}, \boldsymbol{\alpha}, \boldsymbol{\psi}_t$ at time $t$. |
| $\mathbf{N}_t = [\mathbf{N}_t]_{\ell k} = n_{\ell,k,t} \in \mathbb{N}^{t\times t}$ | Matrix of transition counts; $n_{\ell,k,t}$ denotes the number of transitions from state $l$ to state $k$ up to time $t$. |
| $L_t \leq t$ | Number of unique values in $\boldsymbol{\psi}_t$. |
| $\boldsymbol{\Phi}_t = (\mathbf{N}_t, L_t, \hat{\alpha}_t, \hat{\boldsymbol{\beta}}_t, \hat{\gamma}_t)$ | Posterior estimates and sufficient statistics of the structural parameters at time $t$ conditional on $\mathcal{D}_{1:t}$. |
| $\boldsymbol{\Psi}_t = \{\boldsymbol{\Psi}_{\ell,t}\}_{\ell=1}^{L_t+1} = \{(\boldsymbol{\mu}_{\ell,t}, \mathbf{P}_{\ell,t})\}_{\ell=1}^{L_t+1}$ | Posterior estimates of the LG emission parameters at time $t$ conditional on $\mathcal{D}_{1:t}$. |
| $\mathbf{M}_t \in \mathbb{N}^{t\times t}$ | Auxiliary variables sampled from $\mathrm{Antoniak}(\mathbf{N}_t, \alpha_{t-1}, \boldsymbol{\beta}_{t-1})$. |
| $B \in \mathbb{N}$ | Batch size of the robust batched iHMM. |
| $N > 0$ | Number of particles in particle learning. |
| $\omega_t^{(i)}$ | Weight of particle $i \in [1, N]$ at time $t$. |
| $\boldsymbol{y}_t^c \in \mathbb{R}^d$ | Contaminated and unbounded observation at time $t$. |
| $\mathcal{D}_t^c = (\boldsymbol{y}_t^c, \boldsymbol{x}_t)$ | Contaminated data point at time $t$ |
| $\mathcal{D}_{1:t}^c = (\boldsymbol{y}_{1:t}^c, \boldsymbol{x}_{1:t})$ | Contaminated data points from time 1 to $t$ |

*Table A.3.* Table of notations.

## B. Related work

### B.1. Online inference HMMs

Exact posterior inference for HMMs is intractable in the online setting due to the exponential growth of latent state trajectories (Schaeffer et al., 2021). Online HMMs therefore rely on sequential Monte Carlo (SMC) methods (Doucet et al., 2001) to approximate the posterior over latent regimes, while propagating sufficient statistics for the observation parameters via Rao-Blackwellised particle filters (Murphy & Russell, 2001).

Particle learning (PL) schemes exploit conjugate observation models to enable closed-form parameter updates within each particle (Carvalho et al., 2010). While computationally efficient, individual particles remain sensitive to outliers. Prior work has explored particle-level robustification (Maiz et al., 2009; Boustati et al., 2020), but posterior updates within particles can still exhibit unbounded influence, leading to outlier-sensitive state inference and predictions.

### B.2. Robustness in the state space of iHMMs

The behaviour of iHMMs under noisy or misspecified conditions is widely studied. Despite their flexibility in state cardinality, classical iHMMs are prone to rapid state switching and sensitivity to anomalous observations (Shah et al., 2006; Fox et al., 2007; Sgouralis & Pressé, 2017). To mitigate this behaviour, several works modify the HDP prior to encourage state persistence, typically by increasing self-transition probabilities (Murphy, 2002; Fox et al., 2008b; Johnson & Willsky, 2012; Dewar et al., 2012).

Other approaches introduce heavy-tailed observation models (Van Gael et al., 2008), hierarchical structures to isolate outliers (Betkowska et al., 2007; Lee & Lee, 2006), or post-hoc pruning of artefact states (Aguirre et al., 2024). These methods are primarily developed for offline inference and segmentation tasks, and often break conjugacy or require numerical approximations, limiting their applicability in online particle learning settings.

**State-space robustness via heavy-tailed distributions** A common tactic to improve robustness in filtering tasks is to use distributions with heavy tails, a natural choice would be the Student-t distribution (Chen & Liu, 2000; Van Gael et al., 2008). Whilst the Student-t distribution is robust to some extent as its heavy tails accommodate outlier observations, it does not admit a bounded PIF as well. In other words, given large enough outliers, we recover the same pathology illustrated in Lemmas D.4 and D.5. In Section E.5, we demonstrate the behaviour of an iHMM with LG emissions and unknown variances, which admits a Student-t posterior predictive.

**Retrospective smoothing** Backward smoothing is a standard technique in state-space models that refines latent state estimates by incorporating future observations, typically computing smoothed distributions of the form $P(s_t \mid \boldsymbol{y}_{1:T})$. We emphasise that this is not possible in the online prequential setting as the algorithm produces the next $B$ predictions, then performs samples the state-paths once the next $B$ observations are collected. In online settings, fixed-lag smoothing variants approximate $P(s_t \mid \boldsymbol{y}_{1:t+B})$, allowing a limited lookahead window to stabilise inference and mitigate the effect of transient anomalies. While such approaches can improve retrospective state estimation and segmentation accuracy, they are fundamentally incompatible with the prequential forecasting setting considered in this work. In particular, one-step-ahead predictions at time $t$ must be formed using only information available up to time $t-1$. In the batched setting, smoothing methods condition on future observations $\boldsymbol{y}_{t+1:t+B}$ are not available at prediction time. Consequently, any gains from backward smoothing arise from delayed or retrospective inference rather than genuine online prediction. It is possible that retrospective smoothing may improve state allocations post-hoc and improve parametric convergence, however this requires relaxing the constraints imposed in the online-prequential setting. Furthermore, the computational overhead to performing a backward pass per observation/batch remains a risk to computation efficiency.

Our objective is instead to ensure robustness within the filtering distribution itself, so that predictions remain stable under outliers without relying on future data.

### B.3. Observation-space robustness via Generalised Bayes

Robustness in the observation space has been extensively studied through Generalised Bayes (GB) inference, particularly in Bayesian online changepoint detection and Kalman filtering. In these settings, the likelihood is replaced by loss functions or divergences to obtain bounded-influence updates (Karlgaard, 2015; Hooker & Vidyashankar, 2014; Knoblauch et al., 2018; Boustati et al., 2020; Matsubara et al., 2022; Altamirano et al., 2023).

While effective, GB methods have typically been developed outside hierarchical state-space settings and therefore have seen limited integration with models such as HDP-based switching systems. Recent work on weighted-likelihood updates for Kalman filters demonstrates that robust GB-style updates can be incorporated into SSMs while preserving conjugacy and computational efficiency (Duran-Martin et al., 2024). This result motivates extending GB-style robustness to more general hierarchical SSMs.

**Mixture Kalman filters** One such method to approximate Student-t noise models with KFs is to use mixture KFs (Chen & Liu, 2000). In mixture KFs, the filtering distribution is approximated as a finite mixture of Gaussian components, where each component corresponds to a trajectory of a latent indicator process and is propagated via a KF conditional on that trajectory. The same structure arises naturally in LG models, where marginalising over discrete latent variables yields an exponentially growing Gaussian mixture representation of the posterior. From this perspective, HMMs with LG emissions, such as our `BR-iHMM` under LG assumptions, can be interpreted as defining a (potentially infinite) mixture Kalman filter, in which each latent state sequence induces a distinct Kalman filter component. Coupled with the PL approximation, the online `BR-iHMM` algorithm is effectively a mixture KF with trial distributions weighted by the LG per state scaled by the particle weight.

**Other tempered likelihoods** The use of tempered or weighted likelihoods in robust Bayesian inference is closely connected to classical robust estimation procedures such as iteratively reweighted least squares (IRLS) and M-estimation. In IRLS, parameter updates take the form of weighted least squares, where each observation is assigned a weight $w_t$ that depends on its residual $r_t = y_t - \hat{y}_t$. For example, Huber's loss induces weights of the form $w_i = \psi(r_i)/r_i$, where $\psi(\cdot)$ is a bounded influence function, yielding robustness to outliers. This perspective aligns directly with generalised Bayesian updates using a tempered likelihood, where the standard likelihood contribution is replaced by $P(y_t \mid \theta)^{w_t}$, effectively downweighting observations with large residuals. Under this correspondence, the implied weight satisfies the boundedness conditions listed under (9) as it is quadratic in the central region and constant in the tails, hence, the robustness guarantees of Theorem 4.2 carry over directly.

A key connection is that IRLS procedures can often be interpreted as maximum likelihood estimation under implicit heavy-tailed observation models. In particular, the Student-$t$ distribution admits a scale-mixture-of-normals representation,

$$y_t \mid \lambda_t \sim \mathcal{N}(\mu, \sigma^2/\lambda_t), \quad \lambda_t \sim \text{Gamma}\left(\frac{\nu}{2}, \frac{\nu}{2}\right),$$

under which the marginal distribution of $y_t$ is Student-$t$. Conditional on $\lambda_t$, inference reduces to weighted least squares with weights $w_t = \mathbb{E}[\lambda_t \mid y_t]$, yielding an IRLS scheme. This establishes a well-known equivalence between IRLS, robust M-estimation, and parametric heavy-tailed models.

From this viewpoint, tempered likelihood methods can be interpreted as implicitly defining a robust observation model through their weighting function. For instance, a weighting scheme of the form

$$w_t = \frac{1}{1 + r_t^2/c^2}, \qquad c^2 \in \mathbb{R},$$

resembles the posterior expectation of latent precisions in student-$t$ models, suggesting an implicit correspondence with a heavy-tailed distribution. In our setting, the `WoLF-iHMM` update plays exactly this role by replacing the Gaussian likelihood with a weighted likelihood, using an IMQ weight based on the Mahalanobis residual; under the LG assumption, this preserves conjugacy and yields closed-form recursive updates. At the level of a single state's emission model, IRLS/Huber/Student-t noise all correspond to some form of residual-dependent reweighting, and `WoLF-iHMM` is best viewed as the generalised-Bayes analogue of this paradigm. Our online multiplicative weighting framework can further be interpreted as a single-pass, prequential analogue of IRLS. Rather than iterating to convergence at each time step, we compute residuals from the prior predictive, evaluate the IMQ weight once, and perform a single re-weighted Kalman update. This avoids the inner loop of IRLS entirely, which is essential for real-time prequential inference where revisiting past observations is not permitted. The choice of IMQ over Huber-derived weights in this context is motivated by smoothness. Unlike the piecewise-defined Huber weight, which has a discontinuous derivative at its threshold and requires case-tracking at each step, the IMQ weight is everywhere smooth and preserves the closed-form structure of the conjugate Kalman update.

The key difference is that `BR-iHMM` is not only about downweighting outliers in the emission model. The main failure mode of an online iHMM is that a single outlier can also corrupt the latent-state posterior, inducing spurious regime creation or

excessive switching. This is precisely why observation-space robustness alone is insufficient in our model (see Theorem 4.1). Our batched state-space mechanism addresses that second problem by delaying switching decisions and enforcing intra-batch persistence through the degenerate sticky HDP prior, which bounds the state-space influence of a single anomalous point . Classical Student-t or IRLS-style methods do not have an analogue of this state-posterior regularisation; they are robust to residuals, but they do not prevent an outlier from changing the latent segmentation structure.

## C. Background on iHMMs

In this section, we provide a more detailed review of the online iHMMs and present the algorithms in full. We recite, in detail, the online iHMM implemented via particle learning (PL) from (Rodriguez, 2011), then present our batched modifications to the classical online iHMM algorithm. A proof of algorithmic correctness is provided for our batched algorithm.

### C.1. Dirichlet process and HDP construction

In this appendix, we provide the full probabilistic construction underlying the iHMM data-generating process.

Let $(\Omega, \mathcal{F}, \mathbb{P})$ be a probability space and let $H$ be a probability measure on the measurable space $(\mathbb{R}^m, \mathcal{B}(\mathbb{R}^m))$, where $\mathbb{R}^m$ denotes the parameter space. The probability space supports a collection of independent beta-distributed random variables

$$v_i \sim \mathrm{Beta}(1, \gamma), \qquad \gamma > 0,$$

and random vectors $\boldsymbol{\theta}_i$ for $i \in \mathbb{N}$ such that, for any $A \in \mathcal{B}(\mathbb{R}^m)$,

$$\mathbb{P}(\{\omega \in \Omega : \boldsymbol{\theta}_i(\omega) \in A\}) = H(A).$$

Define $\xi_1 = v_1$ and, for $i \geq 2$,

$$\xi_i = v_i \prod_{j=1}^{i-1} (1 - v_j). \tag{16}$$

The resulting random vector $\boldsymbol{\xi} = (\xi_1, \xi_2, \ldots)$ follows a stick-breaking distribution, denoted $\boldsymbol{\xi} \sim \mathrm{SB}(\gamma)$.

Using this construction, we define a Dirichlet process (DP)

$$G : \Omega \times \mathcal{B}(\mathbb{R}^m) \to [0, 1], \qquad G \sim \mathrm{DP}(\gamma, H),$$

which admits the almost-sure representation

$$G(\omega, A) = \sum_{i=1}^{\infty} \xi_i(\omega)\, \delta_{\boldsymbol{\theta}_i(\omega)}(A),$$

for $\omega \in \Omega$ and $A \in \mathcal{B}(\mathbb{R}^m)$, where $\delta_{\boldsymbol{\theta}}(A) = \mathbb{1}_A(\boldsymbol{\theta})$ denotes the Dirac measure. The indicator $\mathbb{1}_A(\boldsymbol{\theta})$ is one if $\boldsymbol{\theta} \in A$ and zero otherwise. By construction, for each $\omega \in \Omega$, $G(\omega)$ is a probability measure on $\mathcal{B}(\mathbb{R}^m)$.

Let $t \in \mathbb{N}$ and fix a finite partition

$$\Xi_t = (A_1, \ldots, A_{t+1}) \tag{17}$$

of $\mathbb{R}^m$ such that $A_i = \{\boldsymbol{\theta}_i(\omega)\}$ for $i \leq t$ and

$$A_{t+1} = \mathbb{R}^m \setminus \bigcup_{\ell=1}^{t} A_\ell.$$

*Remark* C.1. At time $t$, the iHMM has $t$ active states, corresponding to the parameters specified by $A_1, ..., A_t$. The remaining partition $A_{t+1}$, the complement to the union of the active states, encodes all other uninstantiated states. Altogether, the iHMM has $t+1$ choices as to which state is selected to produce $s_{t+1}$, hence the $t+1$-cardinality of $\Xi_t$.

From the defining property of the Dirichlet process, the random vector

$$\boldsymbol{\beta}_t(\omega) = \big(G(\omega, A_1), \ldots, G(\omega, A_{t+1})\big)$$

follows the Dirichlet distribution

$$\boldsymbol{\beta}_t \sim \mathrm{Dir}\big(\gamma H(A_1), \ldots, \gamma H(A_{t+1})\big); \tag{18}$$

see Ferguson (1973). Writing $\beta_\ell := G(A_\ell)$, the realised vector

$$\boldsymbol{\beta}_t(\omega) = (\beta_1(\omega), \ldots, \beta_{t+1}(\omega))$$

lies in the $t$-dimensional simplex

$$\Delta^t = \left\{ \mathfrak{b} \in [0,1]^{t+1} : \sum_{i=1}^{t+1} \mathfrak{b}_i = 1 \right\}.$$

In the iHMM context, $\boldsymbol{\beta}_t$ defines a finite approximation to the global prior over a countably infinite collection of latent states.

We next consider a collection of conditionally independent *child* Dirichlet processes $G_1, \ldots, G_t$, each with precision parameter $\alpha > 0$ and base measure $G(\omega)$, such that

$$G_\ell \sim \mathrm{DP}(\alpha, G(\omega)), \qquad \ell = 1, \ldots, t.$$

For each $\ell$ and $\omega_\ell \in \Omega$, the random measure $G_\ell(\omega_\ell)$ is a probability measure on $\mathcal{B}(\mathbb{R}^m)$. Using the same partition $\Xi$, define the random vectors

$$\boldsymbol{\pi}_{\ell,t}(\omega_\ell) = (\pi_{\ell,1,t}(\omega_\ell), \ldots, \pi_{\ell,t+1,t}(\omega_\ell)), \qquad \pi_{\ell,j,t}(\omega_\ell) = G_\ell(\omega_\ell, A_j),$$

which satisfy

$$\boldsymbol{\pi}_{\ell,t} \sim \mathrm{Dir}(\alpha \boldsymbol{\beta}_t(\omega)).$$

The vector $\boldsymbol{\pi}_{\ell,t}$ defines the state-specific transition probabilities for state $\ell$ in the iHMM at time $t$.

Hereafter, we omit explicit reference to the underlying outcome $\omega \in \Omega$ and write $X$ instead of $X(\omega)$ for all random variables $X$. For notational simplicity, we write $P(X(\omega))$ for the probability mass or density evaluated at the realisation $X(\omega)$, depending on whether $X$ has discrete or continuous support.

Given $\boldsymbol{\pi}_{\ell,t}$ for each $\ell$, the latent state sequence $(s_t)_{t\geq 1}$ evolves according to

$$P(s_t = k \mid s_{t-1} = \ell) = \pi_{\ell,k,t-1}.$$

Each state $\ell$ corresponds to the singleton partition $A_\ell = \{\boldsymbol{\theta}_\ell\}$ and is associated with an observation parameter $\boldsymbol{\theta}_\ell \in \mathbb{R}^m$. Let

$$\boldsymbol{\theta}_{1:t} = \{\boldsymbol{\theta}_1, \boldsymbol{\theta}_2, \ldots, \boldsymbol{\theta}_t\} \subseteq \mathbb{R}^{t\times m}$$

denote the collection of state parameters.

Given the latent state $s_t$, its associated parameter $\boldsymbol{\theta}_{s_t}$, and known covariates $\boldsymbol{x}_t$, the observation $\boldsymbol{y}_t$ is generated according to the linear-Gaussian model

$$P(\boldsymbol{y}_t \mid \boldsymbol{\theta}_{1:t}, s_t, \boldsymbol{x}_t) = \mathcal{N}(\boldsymbol{y}_t \mid f(\boldsymbol{x}_t)\, \boldsymbol{\theta}_{s_t}, \mathbf{R}_t),$$

where $f : \mathcal{X} \to \mathbb{R}^{d\times m}$ is a known feature transformation and $\mathbf{R}_t \in \mathbb{R}^{d\times d}$ is a known observation noise covariance.

### C.2. Inference on iHMMs

We briefly mention the Bayesian inference scheme which the online iHMM relies on in the update step. The online iHMM is scalable because it relies on a series of conjugate prior-likelihood pairs and propagates information from a bottom-up fashion in the iHMM hierarchy. Recall, the sequential inference for iHMMs pertains to computing the joint posterior

$$P(\boldsymbol{\theta}_{1:t}, s_{1:t} \mid \mathcal{D}_{1:t}) = P(\boldsymbol{\theta}_{1:t} \mid s_{1:t}, \mathcal{D}_{1:t})\, P(s_{1:t} \mid \mathcal{D}_{1:t}) \tag{19}$$

where we refer to $P(\boldsymbol{\theta}_{1:t} \mid s_{1:t}, \mathcal{D}_{1:t})$ as the emission posterior, and $P(s_{1:t} \mid \mathcal{D}_{1:t})$ the path posterior. One can conceptualise the iHMM as separating the HDP and LG model, first sampling the HDP, then the LG model conditioned on which state is active from the previous inference step.

Again, we reintroduce the double subscript notation $\ell, t$ to distinguish between the value of a regime-specific estimate of state $\ell \in \{1, \ldots, t\}$ conditioned on $\mathcal{D}_{1:t}$.

**Sequential inference in LG models.** With the LG model, conditioned on $s_{1:t}$, we are interested in the sequential inference of $P(\boldsymbol{\theta}_{1:t} \mid s_{1:t}, \mathcal{D}_{1:t})$. Again, factorisation into sufficient statistics is possible with

$$P(\boldsymbol{\theta}_{1:t} \mid s_{1:t}, \mathcal{D}_{1:t}) = \prod_{k=1}^{t} P(\boldsymbol{\theta}_k \mid \boldsymbol{\Psi}_{k,t}). \tag{20}$$

Now consider the decomposition

$$\begin{aligned}
&P(\boldsymbol{\theta}_{1:t} \mid s_{1:t}, \mathcal{D}_{1:t}) \\
&\quad \propto P(\boldsymbol{\theta}_t \mid \boldsymbol{\theta}_{1:t-1}, s_{1:t}, \mathcal{D}_{1:t})\, P(\boldsymbol{\theta}_{1:t-1} \mid s_{1:t}, \mathcal{D}_{1:t}) \\
&\quad \propto P(\boldsymbol{y}_t \mid \boldsymbol{\theta}_{1:t}, s_{1:t}, \mathcal{D}_{1:t-1}, \boldsymbol{x}_t)\, P(\boldsymbol{\theta}_t \mid \boldsymbol{\theta}_{1:t-1}, s_{1:t}, \mathcal{D}_{1:t-1}, \boldsymbol{x}_t) P(\boldsymbol{\theta}_{1:t-1} \mid s_{1:t}, \mathcal{D}_{1:t}) \\
&\quad = P(\boldsymbol{y}_t \mid \boldsymbol{\theta}_{s_t}, \boldsymbol{x}_t)\, P(\boldsymbol{\theta}_{s_t})\, \prod_{k=1}^{t} P(\boldsymbol{\theta}_k \mid \boldsymbol{\Psi}_{t-1,k})\,,
\end{aligned}$$

where the simplification in the second proportionality to $P(\boldsymbol{\theta}_{s_t})$ is the prior of the parameter.

The update rules remain consistent to that of the LG models, given the sufficient statistics,

$$\begin{aligned}
\mathbf{F}_t &\leftarrow f(\boldsymbol{x}_t) \\
\mathbf{K}_t &\leftarrow \boldsymbol{\Sigma}_{s_t,t-1}\, \mathbf{F}_t^{\intercal}\, \mathbf{S}_{s_t,t}^{-1}, \\
\mathbf{S}_{s_t,t} &\leftarrow \mathbf{F}_t \boldsymbol{\Sigma}_{s_t,t-1} \mathbf{F}_t^{\top} + \mathbf{R}_t. \\
\boldsymbol{\mu}_{s_t,t} &\leftarrow \boldsymbol{\mu}_{s_t,t-1} + \mathbf{K}_t(\boldsymbol{y}_t - \hat{\boldsymbol{y}}_{s_t,t}), \\
\boldsymbol{\Sigma}_{s_t,t} &\leftarrow \boldsymbol{\Sigma}_{s_t,t-1} - \mathbf{K}_t\, \mathbf{S}_t\, \mathbf{K}_t^{\intercal}.
\end{aligned} \tag{21}$$

For all $k \neq s_t$, the sufficient statistics remain the same, that is $\boldsymbol{\Psi}_{k,t-1} = \boldsymbol{\Psi}_{k,t}$.

Next, the induced posterior predictive of the LG model is given by

$$P(\boldsymbol{y}_t \mid \boldsymbol{\Psi}_{t-1}, s_t, \boldsymbol{x}_t) = \int_{\mathbb{R}^m} P(\boldsymbol{y}_t \mid \boldsymbol{\theta}_t, \boldsymbol{x}_t)\, P(\boldsymbol{\theta} \mid \boldsymbol{\Psi}_{t-1}, s_t, \boldsymbol{x}_t)\, d\boldsymbol{\theta} \tag{22}$$

$$= \mathcal{N}(\boldsymbol{y}_t \mid \hat{\boldsymbol{y}}_{s_t,t}, \mathbf{S}_{s_t,t}), \tag{23}$$

with

$$\begin{aligned}
\hat{\boldsymbol{y}}_{s_t,t} &= \mathbf{F}_t \boldsymbol{\mu}_{s_t,t-1}, \\
\mathbf{S}_{s_t,t} &= \mathbf{F}_t \boldsymbol{\Sigma}_{s_t,t-1} \mathbf{F}_t^{\top} + \mathbf{R}_t.
\end{aligned} \tag{24}$$

and $f(\boldsymbol{x}_t) = \mathbf{F}_t$.

Furthermore, a $b$-step ahead prediction is Gaussian with mean and variance

$$\begin{aligned}
\hat{\boldsymbol{y}}_{s_t,t+b|t} &= \mathbf{F}_{t+b} \boldsymbol{\mu}_{s_t,t} \\
\mathbf{S}_{s_t,t+b|t} &= \mathbf{F}_{t+b} \boldsymbol{\Sigma}_{s_t,t} \mathbf{F}_{t+b}^{\top} + \sum_{b'=1}^{b} \mathbf{R}_{t+b'}\,.
\end{aligned} \tag{25}$$

**Sequential inference in HDPs — the path posterior.** Next, the path posterior (5) takes the form

$$\begin{aligned}
P(s_{1:t} \mid \mathcal{D}_{1:t}) &= P(s_{1:t} \mid \mathcal{D}_{1:t-1}, \mathcal{D}_t) \\
&\propto P(\boldsymbol{y}_t \mid s_{1:t}, \mathcal{D}_{1:t-1}, \boldsymbol{x}_t)\, P(s_{1:t} \mid \mathcal{D}_{1:t-1}) \\
&= P(\boldsymbol{y}_t \mid s_{1:t}, \mathcal{D}_{1:t-1}, \boldsymbol{x}_t)\, P(s_t \mid s_{1:t-1}, \mathcal{D}_{1:t-1})\, P(s_{1:t-1} \mid \mathcal{D}_{1:t-1}) \\
&= P(\boldsymbol{y}_t \mid s_t, \boldsymbol{\Psi}_{t-1}, \boldsymbol{x}_t)\, P(s_t \mid s_{t-1}, \boldsymbol{\Phi}_{t-1})\, P(s_{1:t-1} \mid \mathcal{D}_{1:t-1})\,,
\end{aligned} \tag{26}$$

where the last equality indicates the sufficient statistics required to calculate the respective probabilities. In particular, $\boldsymbol{\Phi}_t$ is constructed from $s_{1:t}$ and $\boldsymbol{\Psi}_t$ is constructed from $s_{1:t}$ and $\mathcal{D}_{1:t}$.

The first probability is the conditional probability of $\boldsymbol{y}_t$ given it is assigned to state $s_t$ and is in (22). The second probability follows from Multinomial-Dirichlet conjugacy. Consider the transition probabilities

$$\begin{aligned}
P(s_{1:t} \mid s_{1:t-1}, \boldsymbol{\Phi}_{t-1}) &= P(s_t \mid s_{t-1}, \boldsymbol{\Phi}_{t-1}) \\
&= \int_{\Delta^{t-1}} P(s_t \mid s_{t-1}, \boldsymbol{\pi}_{s_{t-1}}, \boldsymbol{\Phi}_{t-1}) P(\boldsymbol{\pi}_{s_{t-1}} \mid s_{t-1}, \boldsymbol{\Phi}_{t-1})\, d\boldsymbol{\pi}_{s_{t-1}} \\
&= \int_{\Delta^{t-1}} \pi_{s_{t-1}, s_t} P(\boldsymbol{\pi}_{s_{t-1}} \mid \hat{\alpha}_{t-1}, \hat{\boldsymbol{\beta}}_{t-1}, \mathbf{N}_{t-1})\, d\boldsymbol{\pi}_{s_{t-1}}
\end{aligned}$$

with $\boldsymbol{\pi}_{s_{t-1}}$ marginalised out and $\Delta^{t-1} = \{\boldsymbol{\pi} : \boldsymbol{\pi} \in [0,1]^t, \sum_{\ell=1}^t \boldsymbol{\pi}_{s_{t-1}\ell} = 1\}$ is the $(t-1)$ dimensional simplex. Firstly, note that $\boldsymbol{\pi}_{s_{t-1}}$ takes values in $\Delta^{t-1}$ with density following a Dirichlet distribution parametrised by $\hat{\alpha}_{t-1}$ and $\hat{\boldsymbol{\beta}}_{t-1}$. Consequently, the integral is defined over the natural measure of the simplex $\Delta^{t-1}$. Secondly, this integral is the expectation of $\pi_{s_{t-1},s_t}$. We now demonstrate that $\boldsymbol{\pi}_{s_{t-1}}$ follows a Dirichlet distribution, given the multinomial observations recorded by $\mathbf{N}_{t-1}$. Due to Markovian dependencies, only transitions from $s_{t-1}$ are relevant; denote $[n_{s_{t-1},\ell,t-1}]_\ell$ as the $s_{t-1}$-th row in $\mathbf{N}_{t-1}$. By Dirichlet-multinomial conjugacy,

$$\begin{aligned}
P(\boldsymbol{\pi}_{s_{t-1}} \mid \hat{\alpha}_{t-1}, \hat{\boldsymbol{\beta}}_{t-1}, [n_{s_{t-1},\ell,t-1}]_\ell) &\propto P([n_{s_{t-1},\ell,t-1}]_\ell \mid \boldsymbol{\pi}_{s_{t-1}})\, P(\boldsymbol{\pi}_{s_{t-1}} \mid \hat{\alpha}_{t-1}, \hat{\boldsymbol{\beta}}_{t-1}) \\
&= \mathrm{Mult}([n_{s_{t-1},\ell,t-1}]_\ell \mid \boldsymbol{\pi}_{s_{t-1}}) \mathrm{Dir}(\boldsymbol{\pi}_{s_{t-1}} \mid \hat{\alpha}_{t-1} \hat{\boldsymbol{\beta}}_{t-1}) \\
&= \frac{\Gamma(\sum_{\ell=1}^t \hat{\alpha}_{t-1} \hat{\boldsymbol{\beta}}_{\ell,t-1})}{\sum_{\ell=1}^t \Gamma(\hat{\alpha}_{t-1} \hat{\boldsymbol{\beta}}_{\ell,t-1})} \prod_{\ell=1}^t \boldsymbol{\pi}_{s_{t-1},\ell}^{n_{s_{t-1},\ell,t-1} + \hat{\alpha}_{t-1} \hat{\beta}_{\ell,t-1} - 1} \\
&\propto \mathrm{Dir}(\hat{\alpha}_{t-1} \hat{\beta}_{1,t-1} + n_{s_{t-1},1,t-1}, ..., \hat{\alpha}_{t-1} \hat{\beta}_{t,t-1} + n_{s_{t-1},t,t-1})
\end{aligned} \tag{27}$$

and conclude the posterior distribution of $\boldsymbol{\pi}_{s_{t-1}}$ (Beal et al., 2001). Finally, the transition probabilities are given by

$$P(s_t \mid s_{t-1}, \boldsymbol{\Phi}_{t-1}) = \mathbb{E}[\pi_{s_{t-1},s_t}] = \frac{\hat{\alpha}_{t-1} \hat{\beta}_{s_t,t-1} + n_{s_{t-1},s_t,t-1}}{\alpha_{t-1} + n_{s_{t-1},t-1}}\,. \tag{28}$$

In practice, if we had sufficient compute, we sample $s_{1:t}$ by enumerating over all admissible paths $s_{1:t}$, normalise their weights in (6). However, since the domain for $s_{1:t}$ grows exponentially with $t$, we sample the incremental assignment $\hat{s}_t$ given a fixed $\hat{s}_{1:t-1}$.

Once we have sampled $\hat{s}_t$ we can proceed to update the sufficient statistics from $\boldsymbol{\Phi}_{t-1}$ to $\boldsymbol{\Phi}_t$. The process begins by incrementing $\mathbf{N}_{t-1}$ to $\mathbf{N}_t$ such that

$$[\mathbf{N}_t]_{i,j} = \begin{cases} [\mathbf{N}_{t-1}]_{i,j} + 1 & i = \hat{s}_{t-1},\ j = \hat{s}_t \\ [\mathbf{N}_{t-1}]_{i,j} & \text{otherwise} \end{cases} \tag{29}$$

and the running number of active states $L_t$ in the sample path $\hat{s}_{1:t}$. If $\hat{s}_t$ is a new state, we need to update the dimensions of $\hat{\boldsymbol{\beta}}_{t-1}$ from $\mathbb{R}^{L_{t-1}+1}$ to $\mathbb{R}^{L_t+1}$. To increase the dimension of $\hat{\boldsymbol{\beta}}_{t-1}$, we first update the implied partition $\Xi_{t-1} = (A_1, ..., A_{t-1}, A_t)$ to $\Xi_t = (A_1, ..., A_t, A_{t-1})$ on $\mathbb{R}^m$, where $A_t = \{\boldsymbol{\theta}_t\}$ is maps to the singleton partition containing the parameters of the newly instantiated state. Following the stick-breaking representation, we $v \sim \mathrm{Beta}(1, \hat{\gamma}_{t-1})$ and expand the new representation to

$$\hat{\boldsymbol{\beta}}_{t-1} \leftarrow \left( \hat{\beta}_{1,t-1}, ..., \hat{\beta}_{L_{t-1},t-1}, v\hat{\beta}_{L_t,t-1}, (1-v)v\hat{\beta}_{L_t,t-1} \right) \in \mathbb{R}^{L_t+1}. \tag{30}$$

Note that updating the dimensions of $\hat{\boldsymbol{\beta}}_{t-1}$ does not alter its distribution and is not a posterior update. If $\hat{s}_t$ is not a new state, then the expansion in the dimensions of $\hat{\boldsymbol{\beta}}_{t-1}$ is not necessary.

Then, we consider a collection of random variables $m_{\ell,k,t}$ for $\ell, k \in \{1, ..., L_t + 1\}$, such that their probability law is defined by

$$P(m_{\ell,k,t} = m \mid \mathbf{N}_t, \hat{\boldsymbol{\beta}}_{t-1}, \hat{\alpha}_{t-1}) = \frac{\Gamma(\hat{\alpha}_{t-1} \hat{\beta}_{k,t-1})}{\Gamma(\hat{\alpha}_{t-1} \hat{\beta}_{k,t-1} + n_{\ell,k,t})} |S(n_{\ell,k,t}, m)| (\hat{\alpha}_{t-1} \hat{\beta}_{k,t-1})^m \tag{31}$$

for $m \in \{0, ..., n_{\ell k}\}$, where $|S(n, m)|$ is the unsigned Stirling number of the first kind (Antoniak, 1974). We sample $m_{\ell,k,t}$ by calculating the normalised probabilities of (31) for each $m$ and draw $\hat{m}_{\ell,k,t}$ as the sample of a multinomial distribution. We collect all such samples into a matrix $\mathbf{M}_t \in \mathbb{N}^{(L_t+1)\times(L_t+1)}$ such that $[\mathbf{M}_t]_{\ell,k} = m_{\hat{\ell,k,t}}$. Note, the sampling of $\mathbf{M}_t$ is independent to the previous iteration's $\mathbf{M}_{t-1}$, and is only dependent on $\mathbf{N}_t, \hat{\boldsymbol{\beta}}_{t-1}$, and $\hat{\alpha}_{t-1}$. The purpose of sampling such matrix $\mathbf{M}_t$ is to allow direct closed-form updates to $\hat{\alpha}_t, \hat{\gamma}_t$, and $\hat{\boldsymbol{\beta}}_t$, given $\mathbf{N}_t$ and $\hat{\alpha}_{t-1}, \hat{\gamma}_{t-1}$, and $\hat{\boldsymbol{\beta}}_{t-1}$.

Given $\mathbf{M}_t$, we sample for $\ell \in \{1, ..., L_t\}$

$$u_\ell \mid \hat{\alpha}_{t-1}, \mathbf{N}_t \sim \mathrm{Beta}(\hat{\alpha}_{t-1} + 1, n_{\ell,.,t}) \tag{32}$$

$$v_\ell \mid \hat{\alpha}_{t-1}, \mathbf{N}_t \sim \mathrm{Bern}\left(\frac{n_{\ell,.,t}}{n_{\ell,.,t} + \hat{\alpha}_{t-1}}\right) \tag{33}$$

$$\varphi \mid \mathbf{M}_t, \hat{\gamma}_{t-1} \sim \mathrm{Beta}(\hat{\gamma}_{t-1} + 1, m_{..t}) \tag{34}$$

$$\alpha_t \mid \mathbf{u}, \mathbf{v}, \mathbf{M}_t \sim \mathrm{Gam}(a_\alpha + m_{.,.,t} - \sum_{\ell=1}^{L_t} v_\ell,\, b_\alpha - \sum_{\ell=1}^{L_t} \log u_\ell) \tag{35}$$

$$\gamma_t \mid \varphi \sim \varepsilon\, \mathrm{Gam}(a_\gamma + L_t,\, b_\gamma - \log\varphi) + (1-\varepsilon)\, \mathrm{Gam}(a_\gamma + L_t - 1,\, b_\gamma - \log\varphi) \tag{36}$$

$$\boldsymbol{\beta}_t \mid \mathbf{M}_t \sim \mathrm{Dir}(m_{.,1,t}, ..., m_{.,L_t,t}, \hat{\gamma}_t) \tag{37}$$

where

$$\frac{\varepsilon}{1-\varepsilon} = \frac{a_\gamma + L_t - 1}{m_{.,.,t}(b_\gamma - \log\varphi)}.$$

The derivation for sampling $\alpha_t, \gamma_t$ can be found in the appendix of (Teh et al., 2006) (see (A.4-A.6)). Taken together, we update bookkeeping variables $\mathbf{N}_t, \mathbf{M}_t, L_t$, then we draw samples $\hat{\alpha}_t, \hat{\gamma}_t, \hat{\boldsymbol{\beta}}_t$ from their posterior distributions to update the sufficient statistics

$$\boldsymbol{\Phi}_t = (\mathbf{N}_t, \mathbf{M}_t, L_t, \hat{\alpha}_t, \hat{\gamma}_t, \hat{\boldsymbol{\beta}}_t). \tag{38}$$

from $\boldsymbol{\Phi}_{t-1}$.

### C.3. Algorithms on Online iHMMs

The online iHMM is implemented via PL, the algorithm for the `iHMM` online iHMM of (Rodriguez, 2011) is recited in Algorithm 6. The algorithm including `WoLF-iHMM` and our batched implementation is provided in Algorithm 7.

**Initialisation** Initialise the LG parameters with $\boldsymbol{\Psi}_0^{(i)} = \{(\boldsymbol{\mu}_{j,0}^{(i)}, \boldsymbol{\Sigma}_{j,0}^{(i)}\}_{j=1}^{\texttt{MAX_STATES}}$ where

$$\boldsymbol{\mu}_{j,0}^{(i)} \sim \mathcal{N}(\boldsymbol{\mu}_0, \mathbf{I}_m), \qquad \boldsymbol{\Sigma}_{j,0}^{(i)} = \sigma_0^2 \mathbf{I}_m$$

for some hyperparameter $\boldsymbol{\mu}_0 \in \mathbb{R}^m$ and $\sigma_0 > 0$. Initialise the structural parameters $\boldsymbol{\Phi}_0^{(i)} = (\hat{\alpha}_0^{(i)}, \hat{\boldsymbol{\beta}}_0^{(i)}, \hat{\gamma}_0^{(i)}, L_0^{(i)}, \mathbf{N}_0^{(i)})$ where

$$\begin{aligned}
\hat{\alpha}_0^{(i)} &\sim \Gamma(1,1), \\
\hat{\gamma}_0^{(i)} &\sim \Gamma(1,1), \\
\hat{\boldsymbol{\beta}}_0^{(i)} &\sim \mathrm{SB}(\hat{\gamma}_0^{(i)}), \\
L_0^{(i)} &= 1, \\
\mathbf{N}_0^{(i)} &= [0]_{j,k}.
\end{aligned}$$

We initialise $\hat{s}_0^{(i)} = 1$ as the initial state.

**Algorithm 2** `update_counts`

**Input:** $\mathbf{N}_{t-1}^{(i)}, \hat{s}_t^{(i)}, \hat{s}_{t-1}^{(i)}, \hat{\boldsymbol{\beta}}_{t-1}^{(i)}$.
Increment $\mathbf{N}_t$ from $\mathbf{N}_{t-1}$ from (29)
Sample $\mathbf{M}_t^{(i)} = \{m_{\ell,k,t}^{(i)}\}$ from (31)
Update max state $L_t^{(i)} \leftarrow \max\{\hat{s}_t^{(i)}, L_{t-1}^{(i)}\}$
**if** $L_t^{(i)} > L_{t-1}^{(i)}$ (new state) **then**
  Update dimension of $\hat{\boldsymbol{\beta}}_{t-1}^{(i)}$ by sampling $v \sim \mathrm{Beta}(1, \hat{\gamma}_{t-1}^{(i)})$
  $\hat{\boldsymbol{\beta}}_{t-1}^{(i)} \leftarrow \left(\hat{\beta}_1, \ldots, \hat{\beta}_{L_{t-1}^{(i)}}, v\hat{\beta}_{L_t^{(i)}}, (1-v)\hat{\beta}_{L_{t-1}^{(i)}}\right)$
**end if**
**Return** $\mathbf{N}_t^{(i)}, \mathbf{M}_t^{(i)}, L_t^{(i)}, \hat{\boldsymbol{\beta}}_{t-1}^{(i)}$

**Algorithm 3** `update_struct`

**Input:** $\boldsymbol{y}_t$, $\boldsymbol{\Phi}_{t-1}^{(i)}$, $\hat{s}_t^{(i)}$, $\mathbf{N}_t^{(i)}, \mathbf{M}_t^{(i)}, L_t^{(i)}$.
Sample $u_\ell^{(i)}$ and $v_\ell^{(i)}$ auxiliary parameters for $l = 1, \ldots, L_t^{(i)}$ following (32) and (33) respectively.
Update $\hat{\alpha}_t^{(i)}$ by sampling (35) given $u_\ell^{(i)}$ and $v_\ell^{(i)}$.
Sample auxiliary parameter $\varphi^{(i)}$ following (34)
Update $\hat{\gamma}_t^{(i)}$ by sampling (36) given $\varphi^{(i)}$.
Update $\hat{\boldsymbol{\beta}}_t^{(i)}$ following (37).
**Return** $\boldsymbol{\Phi}_t^{(i)} = (\mathbf{N}_t^{(i)}, \mathbf{M}_t^{(i)}, L_t^{(i)}, \hat{\alpha}_t^{(i)}, \hat{\gamma}_t^{(i)}, \hat{\boldsymbol{\beta}}_t^{(i)})$

**Algorithm 4** `lg_update`

**Input:** $\mathcal{D}_t$, $\boldsymbol{\Psi}_{t-1}^{(i)}, \hat{s}_t^{(i)}$.
$\mathbf{F}_t \leftarrow f(\boldsymbol{x}_t)$
$\mathbf{K}_t \leftarrow \boldsymbol{\Sigma}_{\hat{s}_t^{(i)},t-1}\,\mathbf{F}_t^\intercal\,\mathbf{S}_{\hat{s}_t^{(i)},t}^{-1}$
$\mathbf{S}_{\hat{s}_t^{(i)},t} \leftarrow \mathbf{F}_t \boldsymbol{\Sigma}_{\hat{s}_t^{(i)},t-1} \mathbf{F}_t^\top + \mathbf{R}_t$
$\boldsymbol{\mu}_{\hat{s}_t^{(i)},t} \leftarrow \boldsymbol{\mu}_{\hat{s}_t^{(i)},t-1} + \mathbf{K}_t(\boldsymbol{y}_t - \hat{\boldsymbol{y}}_{\hat{s}_t^{(i)},t})$
$\boldsymbol{\Sigma}_{\hat{s}_t^{(i)},t} \leftarrow \boldsymbol{\Sigma}_{\hat{s}_t^{(i)},t-1} - \mathbf{K}_t\,\mathbf{S}_t\,\mathbf{K}_t^\intercal$
**Return** $\boldsymbol{\Psi}_t^{(i)}$

**Algorithm 5** `WoLF_update`

**Input:** $\mathcal{D}_t$, $\boldsymbol{\Phi}_{t-1}^{(i)}, \boldsymbol{\Psi}_{t-1}^{(i)}, \hat{s}_t^{(i)}$, $w_{\hat{s}_t^{(i)},t|t-1}^{(i)}$.
$\mathbf{F}_t \leftarrow f(\boldsymbol{x}_t)$
**if** $n_{\cdot,\hat{s}_t^{(i)},t-1} = 0$ **then**
  $\mathbf{S}_{\hat{s}_t^{(i)},t} \leftarrow \mathbf{F}_t \boldsymbol{\Sigma}_{\hat{s}_t^{(i)},t-1} \mathbf{F}_t^\top + \mathbf{R}_t$
**else**
  $\mathbf{S}_{\hat{s}_t^{(i)},t} \leftarrow \mathbf{F}_t \boldsymbol{\Sigma}_{\hat{s}_t^{(i)},t-1} \mathbf{F}_t^\top + \mathbf{R}_t / (w_{\hat{s}_t^{(i)},t|t-1}^{(i)})^2$
**end if**
$\mathbf{K}_t \leftarrow \boldsymbol{\Sigma}_{\hat{s}_t^{(i)},t-1}\,\mathbf{F}_t^\intercal\,\mathbf{S}_{\hat{s}_t^{(i)},t}^{-1}$
$\boldsymbol{\mu}_{\hat{s}_t^{(i)},t} \leftarrow \boldsymbol{\mu}_{\hat{s}_t^{(i)},t-1} + \mathbf{K}_t(\boldsymbol{y}_t - \hat{\boldsymbol{y}}_{\hat{s}_t^{(i)},t})$
$\boldsymbol{\Sigma}_{\hat{s}_t^{(i)},t} \leftarrow \boldsymbol{\Sigma}_{\hat{s}_t^{(i)},t-1} - \mathbf{K}_t\,\mathbf{S}_t\,\mathbf{K}_t^\intercal$
**Return** $\boldsymbol{\Psi}_t^{(i)}$

**Algorithm 6** Online iHMM via Particle Learning (`iHMM`)

**Input:** $N$ number of particles to use, $\tau_{\text{ESS}}$ minimum ESS before resampling, $(\boldsymbol{\Psi}_0^{(i)}, \boldsymbol{\Phi}_0^{(i)}, \hat{s}_0^{(i)})_{i=1}^N$.
**for** $i \leftarrow 1$ to $N$ **do**
  Assign equal weight $\omega_0^{(i)} \leftarrow 1/N$
**end for**
**for** $t \leftarrow 0$ to $T-1$ **do**
  Compute candidate weight of each particle as ratio of likelihoods with respect to incoming observation:

$$\omega_t^{(i)} = \omega_{t-1}^{(i)} \times \frac{P(\boldsymbol{y}_t \mid \hat{s}_{t-1}^{(i)}, \boldsymbol{x}_t, \boldsymbol{\Phi}_{t-1}^{(i)}, \boldsymbol{\Psi}_{t-1}^{(i)})}{\sum_{j=1}^N P(\boldsymbol{y}_t \mid \hat{s}_{t-1}^{(i)}, \boldsymbol{x}_t, \boldsymbol{\Phi}_{t-1}^{(j)}, \boldsymbol{\Psi}_{t-1}^{(j)})}$$

  where $P(\boldsymbol{y}_t \mid \boldsymbol{x}_t, \boldsymbol{\Phi}_{t-1}^{(i)}, \boldsymbol{\Psi}_{t-1}^{(i)}) = \sum_{\ell=1}^{L_{t-1}^{(i)}} \nu_{\ell,t}^{(i)}$ and $\nu_{\ell,t}^{(i)} = \underbrace{P(\boldsymbol{y}_t \mid s_t^{(i)}, \boldsymbol{x}_t, \boldsymbol{\Psi}_{t-1}^{(i)})}_{\text{see (22)}} \underbrace{P(s_t^{(i)} \mid \hat{s}_{t-1}^{(i)}, \boldsymbol{\Phi}_{t-1}^{(i)})}_{\text{see (6)}}$.

  **if** ESS $\leq \tau_{\text{ESS}}$ (see (124)) **then**
    Sample $N$ particles with replacement wrt to weights $(\omega_t^{(i)}, \hat{s}_{t-1}^{(i)}, \boldsymbol{\Psi}_{t-1}^{(i)}, \boldsymbol{\Phi}_{t-1}^{(i)}) \sim \sum_{i=1}^N \omega_t^{(i)} \delta_{(\hat{s}_{t-1}^{(i)}, \boldsymbol{\Psi}_{t-1}^{(i)}, \boldsymbol{\Phi}_{t-1}^{(i)})}$
    Reset weights to $w_t^{(i)} \leftarrow 1/N$ for all $i$
  **end if**
  Sample next state $\hat{s}_t^{(i)} \mid \hat{s}_{t-1}^{(i)} \sim \text{Mult}(\{1, \dots, L_{t-1}^{(i)} + 1\}, \{\nu_{\ell,t}^{(i)}\}_{\ell=1}^{L_{t-1}^{(i)}+1})$
  $\mathbf{N}_t^{(i)}, \mathbf{M}_t^{(i)}, L_t^{(i)}, \hat{\boldsymbol{\beta}}_{t-1}^{(i)} \leftarrow \texttt{update_counts}(\mathbf{N}_{t-1}^{(i)}, \hat{s}_t^{(i)}, \hat{s}_{t-1}^{(i)}, \hat{\boldsymbol{\beta}}_{t-1}^{(i)})$
  $\boldsymbol{\Phi}_{t+b}^{(i)} \leftarrow \texttt{update_struct}(\boldsymbol{\Phi}_{t+b-1}^{(i)}, \boldsymbol{y}_{t+b}, w_{\hat{s}_{t+b}, t+b|t}^{(i)}, \mathbf{N}_t^{(i)}, \mathbf{M}_t^{(i)}, L_t^{(i)})$
  $\boldsymbol{\Psi}_{t+b}^{(i)} \leftarrow \texttt{lg_update}(\boldsymbol{\Psi}_{t+b-1}^{(i)}, \boldsymbol{y}_{t+b}, w_{\hat{s}_{t+b}, t+b|t}^{(i)})$
**end for**

**Algorithm 7** Batched Online iHMM with WoLF (BR-iHMM)

**Input:** $N$ number of particles, $B > 0$ batch size, $\tau_{\text{ESS}}$ minimum ESS beforr resampling, MAX_STATES max number of states to keep, $\tau_N \in (0, \texttt{MAX_STATES}]$, $(\boldsymbol{\Psi}_0^{(i)}, \boldsymbol{\Phi}_0^{(i)}, \hat{s}_0^{(i)})_{i=1}^N$.
**for** $i \leftarrow 1$ to $N$ **do**
  Assign equal weight $\omega_0^{(i)} \leftarrow 1/N$
**end for**
Set $t \leftarrow 0$
**while** $t < T$ **do**
  Compute candidate weight per particle:

$$\omega_{t+B}^{(i)} \propto \omega_t^{(i)} \frac{P(\boldsymbol{y}_{t+1:t+B} \mid \hat{s}_t^{(i)}, \boldsymbol{\Phi}_t^{(i)}, \boldsymbol{\Psi}_t^{(i)}, \boldsymbol{x}_{t+1:t+B})}{\sum_{j=1}^N P(\boldsymbol{y}_{t+1:t+B} \mid \hat{s}_t^{(i)}, \boldsymbol{\Phi}_t^{(j)}, \boldsymbol{\Psi}_t^{(j)}, \boldsymbol{x}_{t+1:t+B})}$$

  where $P(\boldsymbol{y}_{t+1:t+B} \mid \boldsymbol{\Phi}_t^{(i)}, \boldsymbol{\Psi}_t^{(i)}, \boldsymbol{x}_{1:t+B}) = \sum_{\ell=1}^{L_t^{(i)}} \nu_{\ell,t}^{(i)}$ and

$$\log \nu_{\ell,t}^{(i)} = \log P(s_{t+1}^{(i)} = \ell \mid \hat{s}_t^{(i)}, \boldsymbol{\Phi}_t^{(i)}) + \sum_{b=1}^{B} \left(w_{\ell,\,t+b|t}^{(i)}\right)^2 \log P(\boldsymbol{y}_{t+b} \mid \boldsymbol{\Psi}_{\ell,t}^{(i)}, \boldsymbol{x}_{t+b}).$$

  **if** ESS $\leq \tau_{\text{ESS}}$ (see (124)) **then**
    Sample $N$ particles with replacement

$$(\omega_{t+B}^{(i)}, \{w_{\ell,t+1:t+b}^{(i)}\}_\ell, \hat{s}_t^{(i)}, \boldsymbol{\Psi}_t^{(i)}, \boldsymbol{\Phi}_t^{(i)}) \sim \sum_{i=1}^N \omega_{t+B}^{(i)}\, \delta_{(\{w_{\ell,t+1:t+b}^{(i)}\}_\ell, \hat{s}_t^{(i)}, \boldsymbol{\Psi}_t^{(i)}, \boldsymbol{\Phi}_t^{(i)})}$$

    Reset $w_{t+B}^{(i)} \leftarrow 1/N$ for all $i$
  **end if**
  Sample $\hat{s}_{t+1}^{(i)} \mid \hat{s}_t^{(i)} \sim \text{Mult}\left(\{1, \dots, L_t^{(i)} + 1\}, \{\nu_{\ell,t}^{(i)} / \sum_{\ell=1}^{L_t^{(i)}+1} \nu_{\ell,t}^{(i)}\}_{\ell=1}^{L_t^{(i)}+1}\right)$
  $\mathbf{N}_{t+b}^{(i)}, \mathbf{M}_t^{(i)}, L_t^{(i)}, \hat{\boldsymbol{\beta}}_{t-1}^{(i)} \leftarrow \texttt{update_counts}(\mathbf{N}_{t-1}^{(i)}, \hat{s}_t^{(i)}, \hat{s}_{t-1}^{(i)}, \hat{\boldsymbol{\beta}}_{t-1}^{(i)})$
  $\hat{\boldsymbol{\beta}}_{t-1}^{(i)}, \mathbf{N}_{t+b}^{(i)} \leftarrow \texttt{prune}(\texttt{MAX_STATE}, \tau_N, \mathbf{N}_{t+b}^{(i)}, \hat{\boldsymbol{\beta}}_{t-1}^{(i)}, \boldsymbol{\Psi}_{t-1}^{(i)})$
  **for** $b \leftarrow 1$ to $B$ **do**
    Set $\hat{s}_{t+b}^{(i)} \leftarrow \hat{s}_{t+1}^{(i)}$
    $\boldsymbol{\Psi}_{t+b}^{(i)} \leftarrow \texttt{WoLF_update}(\mathcal{D}_t, \boldsymbol{\Psi}_{t+b-1}^{(i)}, \boldsymbol{\Phi}_{t-1}^{(i)}, \hat{s}_{t+b}^{(i)}, w_{\hat{s}_{t+b}^{(i)}, t+b|t}^{(i)})$
    Sample $\mathbf{M}_{t+b}^{(i)}$ given $\mathbf{N}_{t+B}^{(i)}$ following (31) and zero diagonals $m_{j,j,t+b}^{(i)} \leftarrow 0$
    $\boldsymbol{\Phi}_{t+b}^{(i)} \leftarrow \texttt{update_struct}(\boldsymbol{\Phi}_{t+b-1}^{(i)}, \boldsymbol{y}_{t+b}, w_{\hat{s}_{t+b}, t+b|t}^{(i)}, \mathbf{N}_{t+B}^{(i)}, \mathbf{M}_{t+b}^{(i)}, L_{t+b}^{(i)})$
  **end for**
  Set $t \leftarrow t + B$
**end while**

**Algorithm 8** `prune` (Beam search)

1: **Input:** `MAX_STATES`, popularity threshold $\tau_N$, $\mathbf{N}_t$, $\boldsymbol{\beta}_t$, $\boldsymbol{\Psi}_{t-1}$, timestamps `last_upd`
2: **if** $s_t$ = `MAX_STATES` **then**
3: Candidates $\mathcal{C} \leftarrow \{\tau_N$ states with smallest number of visits $\mathbf{N}_{\ell,\cdot,t} = \sum_{k=1}^{L_t} n_{\ell,k,t}\}$
4: $\ell^\dagger \leftarrow \arg\min_{\ell \in \mathcal{C}}$ `last_upd`$[\ell]$
5: Remove $\ell^\dagger$-th row and column from $\mathbf{N}_t$
6: Remove entry $\boldsymbol{\beta}_{\ell^\dagger}$
7: Normalize $\boldsymbol{\beta}_t \leftarrow \boldsymbol{\beta}_t / \sum_{j \neq \ell^\dagger} \beta_{j,t}$
8: Remove emission parameters $\boldsymbol{\Psi}_t \leftarrow \boldsymbol{\Psi}_t \setminus \{\boldsymbol{\mu}_{\ell^\dagger,t}, \boldsymbol{\Sigma}_{\ell^\dagger,t}\}$
9: **end if**
10: **Return** $\boldsymbol{\beta}_t$, $\boldsymbol{\Psi}_t$

### C.4. Hyperparameters

All iHMM variables are optimised using `bayesian-optimization` on a training partition. This is not an exhaustive list and the number of hyperparameters may increase depending on the task.

| Parameter | Model | |
|---|---|---|
| $\tau_{\text{ESS}}$ | all | `BayesOpt` |
| $c^2$ | `WoLF-iHMM`, `BR-iHMM` | `BayesOpt` |
| $B$ | `BR-iHMM` | `BayesOpt` |

*Table C.4.* Summary of hyperparameters across iHMMs.

## D. Proofs of Theoretical Results

### D.1. Proof of Proposition D.1

We modify the sticky HDP (SHDP) prior from (Fox et al., 2008b) and introduce a time-dependent degenerate sticky prior. In particular, we take

$$G_\ell \sim \text{DP}\left(\hat{\alpha}_t + \kappa_t, \frac{\hat{\alpha}_t \hat{\boldsymbol{\beta}}_t + \kappa_t \delta_{\boldsymbol{\theta}_\ell}}{\hat{\alpha}_t + \kappa_t}\right), \tag{39}$$

where

$$\kappa_t = \begin{cases} 0, & t \equiv 1 \pmod{B}, \\ \infty, & o/w \end{cases} \tag{40}$$

determines whether the iHMM is forced to self-transition or can perform state inference normally. At the limit where $\kappa_t \to \infty$, the random measure follows $\text{DP}(\infty, \delta_{\boldsymbol{\theta}_\ell})$, then draws from such distributions $G_\ell = \delta_{\boldsymbol{\theta}_\ell}$ almost surely. This modification enforces a fixed-batch sizes, i.e., consider the same partition $\Xi_t$ in Section 3, the transition probabilities is Dirichlet distributed with

$$\boldsymbol{\pi}_{\ell,t} \mid \boldsymbol{\Phi}_t \sim \text{Dir}(\hat{\alpha}_t\, \hat{\boldsymbol{\beta}}_t + \boldsymbol{e}_{\ell,t}\, \kappa_t). \tag{41}$$

where $\boldsymbol{e}_{\ell,t} \in \{0,1\}^{t+1}$ is zero everywhere except for the $\ell$-th component. Recall from (28), the transition probabilities is marginalised over $\boldsymbol{\pi}_{\ell,t}$ for $s_t = \ell$, and at the limit of $\kappa_t \to \infty$

$$\lim_{\kappa_t \to \infty} P(s_{t+1} \mid s_t = \ell, \boldsymbol{\Phi}_t) = \lim_{\kappa_t \to \infty} \frac{\hat{\alpha}_t \hat{\beta}_{\ell,t} + \kappa_t \mathbb{1}(s_{t+1} = \ell) + n_{\ell, s_{t+1}, t}}{\hat{\alpha}_t + \kappa_t + n_{\ell,\cdot,t}} = \begin{cases} 1, & s_{t+1} = \ell\,, \\ 0, & s_{t+1} \neq \ell\,. \end{cases} \tag{42}$$

which enforces $s_{t+1} = \ell$ for a fixed batch size due to deterministic scheduling of $\kappa_t$.

We now derive the posterior update rules for Algorithm 7 and prove that steps 13-20 are the correct conditional dependencies for sampling $\mathbf{M}_t$ and subsequently updating the sufficient statistics $\boldsymbol{\Phi}_t$ given its previous values $\boldsymbol{\Phi}_{t-1}$ and $\hat{s}_t$.

**Proposition D.1.** *Consider a batch of size $B$ spanning from times $t+1$ to $t+B$. Let $\mathbf{N}_t$ and $\mathbf{N}_{t+B}$ denote the count matrices before and after observing the batch $\boldsymbol{y}_{t:t+b}$. Then, under the self-transition bias $\kappa_t$, defined in (14), the transition matrix is only incremented once from $t$ to $t+B$. In contrast, the structural HDP parameters $\hat{\alpha}_t, \hat{\boldsymbol{\beta}}_t, \hat{\gamma}_t$, and the auxiliary counts $\mathbf{M}_t$ are resampled every intra-batch step conditional on the same $\mathbf{N}_{t+B}$. Furthermore, for all intra-batch times, the diagonal entries of $\mathbf{M}_t$ are set to zero.*

*Proof.* Define the batch boundaries as $\tau_B = \{t : t \equiv 1 \pmod B\}$, such that $\forall t \in \tau_B, \kappa_t = 0$. Conversely, $\forall t \notin \tau_B, \kappa_t = \infty$.

In this proof, we justify the implementation shortcut in steps 13-20 in Algorithm 7 where (i) we only increment the transition matrix $\mathbf{N}_t$ if $t \in \tau_B$ is a batch boundary and (ii) justify subsequent resampling steps for intra-batch updates.

In the original work of Fox et al. (2008b), an override variable is introduced to explicitly model when a transition is forced to remain in the current state due to the self-transition bias $\kappa_t$ (which is uninformative to the HDP in the Bayesian sense), separating this mechanism from the base transition distribution. In this spirit, we isolate the transition matrix into two parts, an intra-batch count which only increments when $\kappa_t = 0$ and when transitions are driven by the HDP, versus a cross-batch count which only increments when $\kappa_t = \infty$ when transitions are forced,

$$\mathbf{N}_t = \mathbf{N}_t^{\text{intra}} + \mathbf{N}_t^{\text{cross}} \tag{43}$$

such that

$$[\mathbf{N}_t^{\text{intra}}]_{\ell k} = n_{\ell,k,t}^{\text{intra}} = \sum_{t' \notin \tau_B} \mathbb{1}(s_{t'-1} = \ell)\mathbb{1}(s_{t'} = k)$$
$$[\mathbf{N}_t^{\text{cross}}]_{\ell k} = n_{\ell,k,t}^{\text{cross}} = \sum_{t' \in \tau_B} \mathbb{1}(s_{t'-1} = \ell)\mathbb{1}(s_{t'} = k)$$

and increment the $\mathbf{N}_t^{\text{cross}}$ if $t \in \tau_B$ and $\mathbf{N}_t^{\text{intra}}$ otherwise. Note, $\mathbf{N}_t^{\text{intra}}$ is a diagonal matrix as it records intra-batch transitions only.

During inference, we first sample $\mathbf{M}_t$ given $\mathbf{N}_t$, then follow Fox et al. (2008b) and adjust the auxiliary counts in $\mathbf{M}_t$ to $\bar{\mathbf{M}}_t$ by subtracting the number of overriden transitions $o_{j,i,t}$ due to the self-transition bias. In other words, $o_{j,i,t}$ tracks the (expected) number of transitions that should be excluded as it is uninformative to the HDP Bayesian hierarchy. The prior distribution of the override variable is Bernoulli,

$$o_{j,i,t} \sim \text{Bern}\left(\frac{\kappa_t \mathbb{1}(i=j)}{\alpha_{t-1} + \kappa_t \mathbb{1}(i=j)}\right). \tag{44}$$

As introduced in Fox et al. (2007), we interpret $o_{j,i,t} = 1$ as the HDP being overriden by the bias (causing a self-transition) and is therefore uninformative to the HDP prior, versus $o_{j,i,t} = 0$ indicating the cases where the transition is induced by the HDP and should be included in the matrix of auxiliary counts $\mathbf{M}_t$.

The idea is that updates to the posterior distribution should only consider the cases where $o_{j,i,t} = 0$. More precisely, we seek the elements of the adjusted auxiliary counts. Let $\bar{\mathbf{M}}_t$ be the adjusted $\mathbf{M}_t$ matrix with the self-transition effects isolated, such that $[\bar{\mathbf{M}}_t]_{jk} = \bar{m}_{j,k,t}$, and

$$\bar{m}_{j,k,t} = \begin{cases} m_{j,k,t}, & j \neq k \\ m_{j,j,t} - \sum_{i=1}^{m_{j,j,t}} o_{j,i,t}, & j = k \end{cases} \tag{45}$$

where $[\mathbf{M}_t]_{jk} = m_{j,k,t}$ has distribution $\mathbf{M}_t \sim \text{Antoniak}(\mathbf{N}_t, \hat{\alpha}_{t-1}, \hat{\boldsymbol{\beta}}_{t-1})$ as defined in Antoniak (1974).

We focus on intra-batch times, as the override construction is only required when the self-transition bias is active. At inter-batch times $t \in \tau_B$, we set $\kappa_t = 0$, which reduces the transition prior to the HDP without stickiness, and hence no overridden transitions need to be accounted for. Given the degenerate SHDP prior, recall from (31), to perform closed-form updates to $\hat{\alpha}_t, \hat{\gamma}_t, \hat{\boldsymbol{\beta}}_t$ given $\mathbf{N}_t, \hat{\alpha}_{t-1}, \hat{\gamma}_{t-1}, \hat{\boldsymbol{\beta}}_{t-1}$, we sample from a series of auxiliary random variables $m_{j,k,t} \in \{0, ..., n_{j,k,t}\}$ to obtain $[\mathbf{M}_t]_{j,k}$. Since $t \notin \tau_B$, which has $\kappa_t = \infty$ in the limiting sense, the limiting distribution of (31) becomes

$$\begin{aligned} &P(m_{j,k,t} \mid n_{j,k,t}, \hat{\boldsymbol{\beta}}_{t-1}, \hat{\alpha}_{t-1}) \\ &\quad = \lim_{\kappa_t \to \infty} \frac{\Gamma(\hat{\alpha}_{t-1}\hat{\beta}_{k,t-1} + \kappa_t \mathbb{1}(j=k))}{\Gamma(\hat{\alpha}_{t-1}\hat{\beta}_{k,t-1} + \kappa_t \mathbb{1}(j=k) + n_{j,k,t})} \left|S(n_{j,k,t}, m_{j,k,t})\right| (\hat{\alpha}_{t-1}\hat{\beta}_{k,t-1} + \kappa_t \mathbb{1}(j=k))^{m_{j,k,t}} \end{aligned} \tag{46}$$

which gets dominated by the exponential term raised to $m_{j,k,t}$. Importantly, the asymptotic behaviour between the ratio of two Gammas is well documented in Olver et al. (2025, Equation (5.11.12)). We adopt the asymptotic form given in Tricomi et al. (1951) and study the case where $z \to \infty$ as $\kappa_t \to \infty$. Let $z = \hat{\alpha}_{t-1}\hat{\beta}_{j,t-1} + \kappa_t$, and consider the case $j = k$ and $\kappa_t \to \infty$,

$$\begin{aligned} P(m_{j,j,t} \mid n_{j,j,t}, \hat{\boldsymbol{\beta}}_{t-1}, \hat{\alpha}_{t-1}) \\ &= \lim_{\kappa_t \to \infty} \frac{\Gamma(\hat{\alpha}_{t-1}\hat{\beta}_{j,t-1} + \kappa_t)}{\Gamma(\hat{\alpha}_{t-1}\hat{\beta}_{j,t-1} + \kappa_t + n_{j,j,t})} \, |S(n_{j,j,t-1}, m_{j,j,t})| \times (\hat{\alpha}_{t-1}\hat{\beta}_{j,t-1} + \kappa_t)^{m_{j,j,t}} \\ &= \lim_{z \to \infty} \frac{\Gamma(z)}{\Gamma(z + n_{j,j,t})} \, |S(n_{j,j,t}, m_{j,j,t})| \times z^{m_{j,j,t}}. \end{aligned} \tag{47}$$

Specifically,

$$\lim_{z \to \infty} \frac{\Gamma(z)\, z^{m_{j,j,t}}}{\Gamma(z + n_{j,j,t})} = z^{m_{j,j,t} - n_{j,j,t}} \left(1 - \frac{n_{j,j,t}(n_{j,j,t} - 1)}{2z} + O(z^{-2})\right). \tag{48}$$

Since $m_{j,j,t} \leq n_{j,j,t}$, it is clear that the right hand side resolves to 1 only when $m_{j,j,t} = n_{j,j,t}$. If $0 \leq m_{j,j,t} < n_{j,j,t}$ and $z \to \infty$, then $z^{m_{j,j,t} - n_{j,j,t}} \to 0$ and the brackets go to 1. Hence $m_{j,j,t} = n_{j,j,t}$ almost surely and the diagonals of $\mathbf{M}_t$ and $\mathbf{N}_t$ are almost surely equal.

We now show that, for intra-batch resampling, the diagonal entries of the adjusted auxiliary count matrix $\bar{\mathbf{M}}_t$ are zero as a result of the determinisitic batch scheduling. For our case of $\kappa_t$, the override's value is deterministic. Specifically, as $\kappa_t \to \infty$ for $t \notin \tau_B$, the Bernoulli override $o_{j,j,t} \to 1$. Similarly, for $\kappa_t = 0$ in intra-batch times $t \in \tau_B$, the override is automatically $o_{j,i,t} = 0$. Thus, the diagonal entries of $\mathbf{M}_t$ reduce to 0:

$$\bar{m}_{j,j,t} = m_{j,j,t} - \sum_{j=1}^{m_{j,j,t}} 1 = 0 \qquad \forall t \notin \tau_B. \tag{49}$$

In contrast, the off-diagonals are unaffected as the bias $\kappa_t$ is only applicable for self-transitions, the possible outcomes that $m_{j,k,t}$ can takes values in remain $\{0, ..., n_{j,k,t}\} = \{0, ..., n^{\text{cross}}_{j,k,t}\}$, since intra-batch off-diagonals are 0. Therefore, it suffices to only keep track of $\mathbf{N}^{\text{cross}}_t$, the shortcut being only tracking increments at batch boundaries.

Now consider the Bayesian updates for the remaining structural parameters

$$\begin{aligned} P(\boldsymbol{\beta}_t, \gamma_t, \alpha_t, \mathbf{M}_t \mid \boldsymbol{\Phi}_{t-1}, s_t) \propto P(\boldsymbol{\beta}_t \mid \mathbf{M}_t, \hat{\gamma}_t, \kappa_t)\, P(\alpha_t \mid \mathbf{M}_t, L_t, \kappa_t) \\ \times P(\gamma_t \mid \mathbf{M}_t, L_t, \hat{\gamma}_{t-1}, \kappa_t)\, P(\mathbf{M}_t \mid \boldsymbol{\Phi}_{t-1}, \mathbf{N}_t). \end{aligned}$$

Although the count matrix itself is not incremented during intra-batch resampling, the distribution of $\mathbf{M}_t \mid \hat{\alpha}_{t-1}, \hat{\boldsymbol{\beta}}_{t-1}$ differs from that of $\mathbf{M}_{t-1} \mid \hat{\alpha}_{t-2}, \hat{\boldsymbol{\beta}}_{t-2}$ because the hyperparameters evolve over time (i.e., $\hat{\alpha}_{t-1} \neq \hat{\alpha}_{t-2}$ or $\hat{\boldsymbol{\beta}}_{t-1} \neq \hat{\boldsymbol{\beta}}_{t-2}$).

In the online inference procedure, we first sample $\mathbf{M}_t$ and subsequently update the hyperparameters $\hat{\alpha}_t$ and $\hat{\boldsymbol{\beta}}_t$. Consequently, within a batch of size $B$, $\mathbf{M}_t$ must still be resampled as the hyperparameters change, even though no new transition counts are introduced. The off-diagonal entries of $\mathbf{M}_t$ therefore remain stochastic through this resampling step.

The sampling updates for $\hat{\alpha}_t$ and $\hat{\boldsymbol{\beta}}_t$ follow Fox et al. (2008b) and remain unchanged in our implementation, provided that the transition counts $\mathbf{N}_t$ and adjusted auxiliary counts $\mathbf{M}_t$ are correctly resampled. □

### D.2. Correctness of Algorithm 8

This section shows that hard deleting a state by dropping its corresponding values in $\boldsymbol{\beta}_t$ and $\mathbf{N}_t$ is theoretically sound such that the implied distribution post-pruning remains a Dirichlet distribution.

**Proposition D.2.** *Let $\boldsymbol{\pi} \sim \mathrm{Dir}(\hat{\alpha}_t\hat{\boldsymbol{\beta}}_t)$ correspond to the partition $\Xi_t$ on $\mathbb{R}^d$ (defined in Section C.1). Removing a pruned state $l^\dagger \leq t$ by removing $\pi_{\ell^\dagger}$ remains a Dirichlet distribution after re-normalisation.*

*Proof.* The proof will largely follow equations (22) and (23) in (Teh et al., 2006). Given the partition $\Xi_t$, we have

$$\boldsymbol{\pi} = (\pi_1, \pi_2, ..., \pi_t, \pi_{t+1}) \sim \mathrm{Dir}\left(\hat{\alpha}_t\hat{\beta}_1, \hat{\alpha}_t\hat{\beta}_2, ..., \hat{\alpha}_t\hat{\beta}_t, \hat{\alpha}_t(1 - \sum_{i=1}^{t} \hat{\beta}_i)\right) \tag{50}$$

under the stick-breaking formulation. Under the standard properties of the Dirichlet distribution, removing $l^\dagger$ gives

$$\frac{1}{1-\pi_{\ell^\dagger}}\Big(\pi_1, ..., \pi_{\ell^\dagger-1}, \pi_{\ell^\dagger}, ..., \pi_{t+1}\Big) \sim \mathrm{Dir}\Big(\hat{\alpha}_t\hat{\beta}_1, ..., \hat{\alpha}_t\hat{\beta}_{\ell^\dagger-1}, \hat{\alpha}_t\hat{\beta}_{\ell^\dagger+1}, ..., \hat{\alpha}_t\hat{\beta}_t, \hat{\alpha}_t(1-\sum_{i=1}^{t}\hat{\beta}_i)\Big). \tag{51}$$

From the right hand side, we simply removed the contributing weight of $\hat{\beta}_{\ell^\dagger}$. □

Technically, re-normalisation is not needed for $\hat{\boldsymbol{\beta}}_t$ post-prune, but we do so in the implementation for numerical purposes.

### D.3. Decomposition of joint PIF

A useful result we used to decompose the joint PIF in (7) is the chain rule of KLDs (Braverman, 2011).

**Proposition D.3.** *Let $P(X, Y)$ and $Q(X, Y)$ be two joint distributions on $(X, Y)$, with corresponding marginals $P(X), Q(X)$, and conditionals $P(Y \mid X), Q(Y \mid X)$. If $X$ has discrete support, then*

$$\mathrm{KL}\big(P(X, Y) \,\|\, Q(X, Y)\big) = \mathrm{KL}\big(P(X) \,\|\, Q(X)\big) + \sum_X P(X)\Big[\mathrm{KL}\big(P(Y \mid X) \,\|\, Q(Y \mid X)\big)\Big]. \tag{52}$$

*Proof.*

$$\begin{aligned}
\mathrm{KL}\big(P(X, Y) \,\|\, Q(X, Y)\big) &= \mathbb{E}_{P(X,Y)}\left[\log \frac{P(X, Y)}{Q(X, Y)}\right] \\
&= \mathbb{E}_{P(X,Y)}\left[\log \frac{P(X)P(Y \mid X)}{Q(X)Q(Y \mid X)}\right] \\
&= \mathbb{E}_{P(X,Y)}\left[\log \frac{P(X)}{Q(X)}\right] + \mathbb{E}_{P(X,Y)}\left[\log \frac{P(Y \mid X)}{Q(Y \mid X)}\right] \\
&= \mathbb{E}_{P(X)}\left[\log \frac{P(X)}{Q(X)}\right] + \mathbb{E}_{P(X)}\Big[\mathbb{E}_{P(Y|X)}\big[\log \frac{P(Y \mid X)}{Q(Y \mid X)}\big]\Big] \\
&= \mathrm{KL}\big(P(X) \,\|\, Q(X)\big) + \mathbb{E}_{P(X)}\big[\mathrm{KL}\big(P(Y \mid X) \,\|\, Q(Y \mid X)\big)\big].
\end{aligned} \tag{53}$$

In particular, the expectation term can be made explicit if $X$ has discrete support, such that

$$\mathbb{E}_{P(X)}\big[\mathrm{KL}\big(P(Y \mid X) \,\|\, Q(Y \mid X)\big)\big] = \sum_X P(X)\Big[\mathrm{KL}\big(P(Y \mid X) \,\|\, Q(Y \mid X)\big)\Big].$$

□

Substitute $X$ with $s_{1:t}$ and $Y$ with $\boldsymbol{\theta}_t$ to obtain (7).

We define the PIF using KLD not merely for tractability, but because it is the natural measure of perturbation for Bayesian inference. Bayesian updating can be characterised as a KL projection, and KL directly quantifies the degradation in log-predictive performance induced by contamination. Moreover, KL is the only standard divergence that (i) admits a chain-rule decomposition over latent-variable models, which is essential for separating observation- and state-space robustness, and (ii) provides a meaningful notion of unbounded influence, unlike bounded divergences such as total variation. While alternative definitions, such as the $\epsilon$-contamination model (Huber, 1981), are possible, they are functional-specific and do not extend naturally to full posterior distributions or hierarchical settings.

### D.4. Proof of Theorem 4.1

We first show that for an arbitrarily large (contaminated) observation $\boldsymbol{y}_t^c$, the iHMM path posterior predictive at time $t$ concentrates around the new state $t$. Then, we use this result to show that the joint PIF is unbounded.

Let $\mathcal{I}_t$ be the set of states that are inactive (have not been updated), more precisely

$$\mathcal{I}_t = \{\ell \leq t : n_{\cdot,\ell,t} = 0\}. \tag{54}$$

**Lemma D.4.** *Consider the LG model (1) and a path $s_{1:t-1}$. Then, the assignment $s_{1:t} = (s_{1:t-1}, \ell)$ is almost surely for some $\ell \in \mathcal{I}_t$ as $\|\boldsymbol{y}_t^c\|_2^2 \to +\infty$. That is, the iHMM assigns the auxiliary variable to a new state given an arbitrarily large outlier.*

*Proof.* Let $\mathcal{D}_t^c = (\boldsymbol{x}_t, \boldsymbol{y}_t^c)$ and $\mathcal{D}_{1:t}^c = (\mathcal{D}_{1:t-1}, \mathcal{D}_t^c)$. Recall that

$$P(s_{1:t} \mid \mathcal{D}_{1:t}) \propto P(\boldsymbol{y}_t \mid \mathcal{D}_{1:t-1}, s_{1:t}, \boldsymbol{x}_t) P(s_t \mid s_{1:t-1})\, P(s_{1:t-1} \mid \mathcal{D}_{1:t-1}). \tag{55}$$

Define

$$\begin{aligned}
\nu_{k,t} &= P(s_t = k \mid s_{1:t-1})\, P(\boldsymbol{y}_t^c \mid \boldsymbol{x}_t, \boldsymbol{\Psi}_{t-1}, s_t = k),\\
\zeta_{k,\ell} &\triangleq \log(\tfrac{\nu_{k,t}}{\nu_{\ell,t}}) \\
\Delta_{k,\ell} &= \log P(\boldsymbol{y}_t^c \mid s_t = k, \boldsymbol{\Psi}_{t-1}, \boldsymbol{x}_t) - \log P(\boldsymbol{y}_t^c \mid s_t = \ell, \boldsymbol{\Psi}_{t-1}, \boldsymbol{x}_t),
\end{aligned} \tag{56}$$

and,

$$\zeta_{k,\ell} = \Delta_{k,\ell} + P(s_t = k \mid s_{1:t-1}) - P(s_t = \ell \mid s_{1:t-1}). \tag{57}$$

Fixing and conditioning on $s_{1:t-1}$ decomposes the path posterior to

$$P(s_{1:t} \mid \mathcal{D}_{1:t}) = P(s_{1:t-1} \mid \mathcal{D}_{1:t-1}) P(s_t \mid s_{1:t-1}, \mathcal{D}_t^c), \tag{58}$$

where

$$\begin{aligned}
P(s_t = t \mid s_{1:t-1}, \mathcal{D}_{1:t}^c) &= \frac{P(s_t = t, s_{1:t-1} \mid \mathcal{D}_{1:t}^c)}{P(s_{1:t-1}, \mathcal{D}_{1:t}^c)}\\
&= \frac{P(s_t = t, s_{1:t-1} \mid \mathcal{D}_{1:t}^c)}{\sum_{\ell=1}^t P(s_t = \ell, s_{1:t-1} \mid \mathcal{D}_{1:t}^c)}\\
&= \frac{P(s_{1:t-1} \mid \mathcal{D}_{1:t-1}) P(s_t \mid s_{1:t-1}) p(\boldsymbol{y}_t^c \mid \boldsymbol{x}_t, s_{1:t-1}, s_t)}{\sum_\ell P(s_{1:t-1} \mid \mathcal{D}_{1:t-1}) P(s_t = \ell \mid s_{1:t-1}) P(\boldsymbol{y}_t^c \mid \boldsymbol{x}_t, s_{1:t-1}, s_t = \ell)}\\
&= \frac{P(s_t = t \mid s_{1:t-1}) P(\boldsymbol{y}_t^c \mid \boldsymbol{x}_t, s_{1:t-1}, s_t = t)}{\sum_\ell P(s_t = \ell \mid s_{1:t-1}) P(\boldsymbol{y}_t^c \mid \boldsymbol{x}_t, s_{1:t-1}, s_t = \ell)}\\
&= \frac{\nu_{t,t}}{\sum_{\ell=1}^t \nu_{\ell,t}}\\
&= \frac{1}{\sum_{\ell=1}^t \exp(\zeta_{\ell,t})}.
\end{aligned} \tag{59}$$

Recall from (21), that the posterior predictive of the LG model is given by $P(\boldsymbol{y}_t \mid s_t, \boldsymbol{\Psi}_{t-1}, \boldsymbol{x}_t) = \mathcal{N}(\boldsymbol{y}_t \mid \hat{\boldsymbol{y}}_{s_t,t|t-1}, \mathbf{S}_{s_t,t|t-1})$. Here, the log-density of a $d$-dimensional Gaussian $\mathcal{N}(\boldsymbol{y} \mid \boldsymbol{\mu}, \mathbf{S})$ is given by

$$\begin{aligned}
\log \mathcal{N}(\boldsymbol{y} \mid \boldsymbol{\mu}, \mathbf{S}) &= \frac{-1}{2}\Big[(\boldsymbol{y} - \boldsymbol{\mu})^\top \mathbf{S}^{-1}(\boldsymbol{y} - \boldsymbol{\mu}) + \log\det \mathbf{S} + d \log(2\pi)\Big]\\
&= \frac{-1}{2}\Big[\boldsymbol{y}^\top \mathbf{S}^{-1}\boldsymbol{y} - 2\boldsymbol{\mu}^\top \mathbf{S}^{-1}\boldsymbol{y} + \boldsymbol{\mu}^\top \mathbf{S}^{-1}\boldsymbol{\mu} + \log\det \mathbf{S} + d\log(2\pi)\Big].
\end{aligned}$$

Hence, the difference in log-marginal likelihoods comprises of four differences.

$$\begin{aligned}
2\,\Delta_{k,\ell} = &(\boldsymbol{y}_t^c)^\top S_{\ell,t}^{-1}(\boldsymbol{y}_t^c) - (\boldsymbol{y}_t^c)^\top S_{k,t}^{-1}(\boldsymbol{y}_t^c) &(60)\\
&+ 2\hat{\boldsymbol{y}}_{k,t|t-1}^\top \mathbf{S}_{k,t}^{-1}\boldsymbol{y}_t^c - 2\hat{\boldsymbol{y}}_{\ell,t|t-1}^\top \mathbf{S}_{\ell,t}^{-1}\boldsymbol{y}_t^c &(61)\\
&+ \hat{\boldsymbol{y}}_{\ell,t|t-1}^\top \mathbf{S}_{\ell,t}^{-1}\hat{\boldsymbol{y}}_{\ell,t|t-1} - \hat{\boldsymbol{y}}_{k,t|t-1}^\top \mathbf{S}_{k,t}^{-1}\hat{\boldsymbol{y}}_{k,t|t-1} &(62)\\
&+ \log\det \mathbf{S}_{\ell,t} - \log\det \mathbf{S}_{k,t}. &(63)
\end{aligned}$$

and $\ell, k \in \{1, ..., t\}$.

Now consider the case where $k \in \mathcal{I}_t$ and $\ell \notin \mathcal{I}_t$. Clearly (62) and (63) are independent of $\boldsymbol{y}_t^c$. Next, (60) + (61) $\in O(\|\boldsymbol{y}_t^c\|_2^2)$, so that $\Delta_{k,\ell} \to \infty$ as $\|\boldsymbol{y}_t\|_2^2 \to \infty$. For the unbounded outlier $\|\boldsymbol{y}_t^c\|_2^2 \to \infty$,

$$\lim_{\|\boldsymbol{y}_t^c\|_2^2 \to \infty} (\boldsymbol{y}_t^c)^\top (\mathbf{S}_{\ell,t}^{-1} - \mathbf{S}_{k,t}^{-1})(\boldsymbol{y}_t^c) \to \infty \iff \mathbf{S}_{\ell,t}^{-1} \succ \mathbf{S}_{k,t}^{-1}.$$

Since, $\mathbf{A} \succ \mathbf{B} \iff \mathbf{B}^{-1} \succ \mathbf{A}^{-1}$, consider

$$\begin{aligned} \mathbf{S}_{k,t} \succ \mathbf{S}_{\ell,t} &\iff \mathbf{F}_t \mathbf{\Sigma}_{k,t-1} \mathbf{F}_t^\top + \mathbf{R}_t \succ \mathbf{F}_t \mathbf{\Sigma}_{\ell,t-1} \mathbf{F}_t^\top + \mathbf{R}_t \\ &\iff \mathbf{\Sigma}_{k,t-1} \succ \mathbf{\Sigma}_{\ell,t-1}. \end{aligned} \tag{64}$$

Therefore it suffices to show $\mathbf{\Sigma}_{k,t-1} \succ \mathbf{\Sigma}_{\ell,t-1}$. From the LG algorithm, we know $\mathbf{\Sigma}_{\ell,t-1}$ is the posterior covariance after updating $n_{\cdot,\ell,t-1}$ times. Since, the LG posterior covariance is independent of the observations $\boldsymbol{y}_{\ell,1:t-1}$, we can index them by the number of times they are updated,

$$\begin{aligned} \mathbf{\Sigma}_{\ell,t-1} &= \mathbf{\Sigma}_{(n_{\cdot,\ell,t-1})}, \quad \forall \ell \in A_t, \\ \mathbf{\Sigma}_{k,t} &= \mathbf{\Sigma}_{(n_{\cdot,k,t-1})} = \mathbf{\Sigma}_{(0)}, \quad \forall k \notin A_t. \end{aligned}$$

The difference between $\mathbf{\Sigma}_{(0)} - \mathbf{\Sigma}_{(1)}$ is

$$\mathbf{\Sigma}_{(0)} - \mathbf{\Sigma}_{(1)} = \mathbf{K}_{(1)} \mathbf{S}_{(1)} \mathbf{K}_{(1)}^\top ,$$

where we know $\mathbf{S}_{(1)} = \mathbf{F}_1 \mathbf{\Sigma}_{(0)} \mathbf{F}_1^\top + \mathbf{R}_1 \succ 0$ by construction and conclude $\mathbf{\Sigma}_{(0)} - \mathbf{\Sigma}_{(1)} \succ 0$. By transitivity, $\mathbf{\Sigma}_{(0)} \succ \mathbf{\Sigma}_n$ for all $n > 0$. Therefore, (64) holds and we conclude that as $\|\boldsymbol{y}_t^c\|_2^2 \to \infty$, $\Delta_{k,l} \to \infty$, and $\log \frac{v_{k,t}}{\nu_{\ell,t}} \to \infty$.

Conversely, if $\ell \in \mathcal{I}_t$, and $\mathbf{S}_{k,t} = \mathbf{S}_{\ell,t} = \mathbf{S}_{(0)}$, (60) goes to zero $\Delta_{k,\ell} \in O(\|\boldsymbol{y}_t^c\|)$. Then the second difference simplifies to

$$(61) = 2(\hat{\boldsymbol{y}}_{k,t|t-1} - \hat{\boldsymbol{y}}_{\ell,t|t-1})^\top \mathbf{S}_{(0)}^{-1} \boldsymbol{y}_t^c. \tag{65}$$

Consider the decomposition $\boldsymbol{y}_t^c = \|\boldsymbol{y}_t^c\|_2 \tilde{\boldsymbol{y}}_t^c$ where $\tilde{\boldsymbol{y}}_t^c$ is a unit vector in the direction of $\boldsymbol{y}_t^c$. Now consider the case where

$$\begin{aligned} &2(\hat{\boldsymbol{y}}_{k,t|t-1} - \hat{\boldsymbol{y}}_{\ell,t|t-1})^\top \mathbf{S}_{(0)}^{-1} \tilde{\boldsymbol{y}}_t^c = 0 \\ \iff &2(\hat{\boldsymbol{y}}_{k,t|t-1} - \hat{\boldsymbol{y}}_{\ell,t|t-1})^\top \mathbf{S}_{(0)}^{-1} \boldsymbol{y}_t^c = 0 \end{aligned} \tag{66}$$

then, either $\hat{\boldsymbol{y}}_{k,t|t-1} = \hat{\boldsymbol{y}}_{\ell,t|t-1}$, and $\Delta_{k,\ell} = 0$, assigning equal probability to both states $k$ and $\ell$, or, $2\Delta_{k,\ell} = (62) \in O(1)$ assigning positive probability to both states independent of $\boldsymbol{y}_t^c$. Alternatively, without loss of generality, consider the case where

$$2(\hat{\boldsymbol{y}}_{k,t|t-1} - \hat{\boldsymbol{y}}_{\ell,t|t-1})^\top \mathbf{S}_{(0)}^{-1} \tilde{\boldsymbol{y}}_t^c > 0. \tag{67}$$

Then, as $\|\boldsymbol{y}_t^c\|_2 \to \infty$,

$$2\Delta_{k,\ell} = 2\|\boldsymbol{y}_t^c\|_2 (\hat{\boldsymbol{y}}_{k,t|t-1} - \hat{\boldsymbol{y}}_{\ell,t|t-1})^\top \mathbf{S}_{(0)}^{-1} \tilde{\boldsymbol{y}}_t^c + O(1) \to \infty, \tag{68}$$

and state $k$ is infinitely more likely to be assigned to than $\ell$.

Finally, recall

$$P(s_t = \ell \mid s_{t-1}, \mathcal{D}_t^c, \mathbf{\Psi}_{t-1}, \mathbf{\Phi}_{t-1}) = \frac{\nu_{\ell,t}}{\sum_{k' \notin \mathcal{I}_t} \nu_{k',t} + \sum_{k \in \mathcal{I}_t} \nu_{k,t}} = \frac{1}{\sum_{k' \notin \mathcal{I}_t} \exp(\zeta_{k',\ell}) + \sum_{k \in \mathcal{I}_t} \exp(\zeta_{k,\ell})}. \tag{69}$$

Now consider

$$P(s_t \in \mathcal{I}_t \mid s_{t-1}, \mathcal{D}_t^c, \mathbf{\Psi}_{t-1}, \mathbf{\Phi}_{t-1}) = \frac{\sum_{k \in \mathcal{I}_t} \nu_{k,t}}{\sum_{k' \notin \mathcal{I}_t} \nu_{k',t} + \sum_{k \in \mathcal{I}_t} \nu_{k,t}} = \left[ \frac{\sum_{k' \notin \mathcal{I}_t} \nu_{k',t}}{\sum_{k \in \mathcal{I}_t} \nu_{k,t}} + 1 \right]^{-1}. \tag{70}$$

It suffices to show that at the limit $\|\boldsymbol{y}_t^c\|_2^2 \to \infty$,

$$\frac{\sum_{k' \notin \mathcal{I}_t} \nu_{k',t}}{\sum_{k \in \mathcal{I}_t} \nu_{k,t}} \to 0. \tag{71}$$

Since there exists at least one $k \in \mathcal{I}_t$, observe that

$$\frac{\sum_{k' \notin \mathcal{I}_t} \nu_{k',t}}{\sum_{k \in \mathcal{I}_t} \nu_{k,t}} \leq \sum_{k' \notin \mathcal{I}_t} \frac{\nu_{k',t}}{\nu_{k,t}} = \sum_{k' \notin \mathcal{I}_t} \exp(\zeta_{k',k}) = \sum_{k' \notin \mathcal{I}_t} \exp(-\zeta_{k,k'}) \to 0, \tag{72}$$

where the first inequality is due to the non-negativity of $\nu_{k,t}$.

Hence, $P(s_t \in \mathcal{I}_t \mid s_{t-1}, \mathcal{D}_t^c, \mathbf{\Psi}_{t-1}, \mathbf{\Phi}_{t-1}) \to 1$ as required. □

**Lemma D.5.** *The iHMM has unbounded* $\mathrm{PIF}_{s_t}$.

*Proof.* We now show that the PIF is also unbounded in the state-space $\mathrm{PIF}_{s_t}$. Since the KL divergence is between two discrete distributions, the PIF takes the form

$$\mathrm{PIF}_{s_t}(\boldsymbol{y}_t^c, \boldsymbol{y}_{1:t}) = \sum_{s_{1:t}} P(s_t \mid s_{t-1}, \mathcal{D}_t, \boldsymbol{\Psi}_{t-1}, \boldsymbol{\Phi}_{t-1}) \log \frac{P(s_t \mid s_{t-1}, \mathcal{D}_t, \boldsymbol{\Psi}_{t-1}, \boldsymbol{\Phi}_{t-1})}{P(s_t \mid s_{t-1}, \mathcal{D}_t^c, \boldsymbol{\Psi}_{t-1}, \boldsymbol{\Phi}_{t-1})} \tag{73}$$

and sums over all possible trajectories $s_{1:t}$.

In the following, we show PIF unboundedness by first showing that the uncontaminated log-posterior assigns non-zero mass to all possible paths. We then contrast this by noting that the contaminated case only assigns zero mass to some state path $\psi_t^\dagger$ in the limit of $\|\boldsymbol{y}_t^c\|_2 \to \infty$ as stated by Lemma D.4 where we know $P(s_t \mid s_{t-1}, \mathcal{D}_t^c, \boldsymbol{\Psi}_{t-1}, \boldsymbol{\Phi}_{t-1})$ is non-zero only if $s_t \in \mathcal{I}_t$.

Firstly, we present the argument that all possible trajectories have positive mass when an outlier is not present. More precisely, we first show that HDP prior $P(s_{1:t} \mid s_{1:t-1}) > 0$ assigns non-zero weights to all components of the transition probabilities $\boldsymbol{\pi}_{s_{t-1},t} \in \Delta^{t-1}$, assuming well-behaved marginal likelihood $P(\boldsymbol{y}_t \mid s_t, \boldsymbol{\Psi}_{t-1}, \boldsymbol{x}_t) \in (0, 1)$ for non-outlier $\boldsymbol{y}_t$. Recall the recursive sampling of subsequent state estimate $\hat{s_t}$ is sampled from the posterior distribution

$$P(s_t \mid \boldsymbol{\pi}_{s_{t-1},t}) \sim \mathrm{Mult}(\{1, ..., t\} \mid \boldsymbol{\pi}_{s_{t-1},t}).$$

The iHMM inference process samples the HDP for the next state using the previous iteration's $\boldsymbol{\Phi}_{t-1}$. If $\boldsymbol{\beta}_{t-1}$ is a vector of non-zero values, and all of its components are less than one, then by definition of Dirichlet distributions, the posterior transition probabilities $\boldsymbol{\pi}_{s_{t-1},t} \sim \mathrm{Dir}(\alpha\boldsymbol{\beta}_{t-1} + [n_{s_{t-1},\ell\, t-1}]_\ell)$ is also a random vector that only takes values in vectors without zeros in any of its components. Recall the partition $\Xi_{t-1} = (A_1, ..., A_t)$ on the parameter space defined in Section 3. Since $\boldsymbol{\xi} = (\xi_1, \xi_2, ...) \sim \mathrm{SB}(\gamma_{t-1})$, therefore we know for any $i$, $\xi_i > 0$ almost surely. Furthermore, by the stick breaking construction, we know

$$\beta_\ell = G(A_\ell) = \sum_{i=1}^{\infty} \xi_i \delta_{\boldsymbol{\theta}_i}(A_\ell) > 0,$$

as $A_\ell = \{\boldsymbol{\theta}_\ell\}$ by construction. Therefore, the vector $\boldsymbol{\beta}_{t-1}$ is non-zero for any of its components. Thus, it follows that $\boldsymbol{\beta}_{t-1}$ is also non-zero for any of its components, such that

$$P(s_t \mid s_{t-1}, \boldsymbol{\Phi}_{t-1}) = \mathbb{E}[\boldsymbol{\pi}_{s_{t-1},t}] \propto \hat{\alpha}_{t-1}\hat{\boldsymbol{\beta}}_{t-1} + [n_{s_{t-1},\ell,t-1}]_\ell$$

is the marginalised HDP prior. Finally, we conclude the argument by applying the non-zero property of $\boldsymbol{\beta}_t$ and $\boldsymbol{\pi}_t$ for all $t$ and conclude any valid trajectory is assigned a non-zero mass under an uncontaminated log-posterior, i.e.

$$P(s_t \mid s_{t-1}, \mathcal{D}_t, \boldsymbol{\Psi}_{t-1}, \boldsymbol{\Phi}_{t-1}) > 0 \tag{74}$$

for all $s_t \in \{1, 2, ..., t\}$ and specifically $P(s_t = t \mid s_{t-1}, \mathcal{D}_t, \boldsymbol{\Psi}_{t-1}, \boldsymbol{\Phi}_{t-1})$.

In contrast, Lemma D.4 shows $P(s_t = \ell \mid s_{t-1}, \mathcal{D}_t^c, \boldsymbol{\Psi}_{t-1}, \boldsymbol{\Phi}_{t-1}) \to 0$ for $\ell \notin \mathcal{I}_t$ when $\|\boldsymbol{y}_t^c\|_2^2 \to \infty$. The divergence in (73) is therefore unbounded at all paths $s_{1:t}$ such that $s_t \notin \mathcal{I}_t$. □

Hence, even if $\mathrm{PIF}_{\boldsymbol{\theta}_t}$ is bounded using the weighted-observation likelihood filter (WoLF) from (Duran-Martin et al., 2024), the joint PIF explodes due to an unbounded $\mathrm{PIF}_{s_t}$.

**Lemma D.6.** $\mathrm{PIF}_{\boldsymbol{\theta}_t}$ *is unbounded when* $\|\boldsymbol{y}_t^c\|_2^2 \to \infty$.

*Proof.* From Lemma D.5, we know as $\|\boldsymbol{y}_t^c\|_2^2 \to \infty$, $s_t \in \mathcal{I}_t$ almost surely. From (11), posterior updates for $s_t \in \mathcal{I}_t$ follow the standard KF posterior

$$\mathbf{S}_{s_t} \leftarrow \mathbf{F}_t \boldsymbol{\Sigma}_{s_t} \mathbf{F}_t^\top + \mathbf{R}_t \tag{75}$$

and is therefore unbounded by Duran-Martin et al. (2024, Lemma C.1). □

Therefore, the proof for Theorem 4.1 follows from Lemmas D.5 to D.6, where we have an additive decomposition of two unbounded PIFs.

### D.5. Proof of Theorem 4.2

Let $\mathcal{D}^c_{s:t} = \{(\boldsymbol{y}^c_u, \boldsymbol{x}_u)\}_{u=s}^t$ denote a contaminated batch. We first extend the definition of PIFs to accommodate multiple datapoints and batched posteriors. For a batch size of $B$, the state PIF $\mathrm{PIF}_{s_t}$ and observations PIF $\mathrm{PIF}_{\boldsymbol{\theta}_t}$ is defined by the KLD between the batched posteriors that are conditional on an uncontaminated batch of data $\mathcal{D}_{1:t+b}$ and a contaminated batch $\mathcal{D}_{1:t}, \mathcal{D}^c_{t+1:t+b}$. More precisely,

$$\mathrm{PIF}^B_{s_{t+B}}(\boldsymbol{y}^c_{t+1:t+B}, \boldsymbol{y}_{1:t}) = \mathrm{KL}(P(s_{1:t+B} \mid \mathcal{D}_{1:t+B}) \mid\mid P(s_{1:t+B} \mid \mathcal{D}_{1:t}, \mathcal{D}^c_{t+1:t+B}), \tag{76}$$

$$\mathrm{PIF}^B_{\boldsymbol{\theta}_{t+B}}(\boldsymbol{y}^c_{t+1:t+B}, \boldsymbol{y}_{1:t}) = \mathrm{KL}(P(\boldsymbol{\theta}_{t+B} \mid \mathcal{D}_{1:t+B}) \mid\mid P(\boldsymbol{\theta}_{t+B} \mid \mathcal{D}_{1:t}.\mathcal{D}^c_{t+1:t+B}). \tag{77}$$

Theorem 4.2 consists of two subclaims, which we establish separately as lemmas.

The first subclaim concerns the boundedness of the state-space PIF under the batched posterior defined in (12), i.e.,

$$\mathrm{PIF}^B_{s_{t+B}}(\boldsymbol{y}^c_{t+1:t+B}, \boldsymbol{y}_{1:t}) = \mathrm{KL}(\nu(s_{1:t+B} \mid \mathcal{D}_{1:t+B}) \mid\mid \nu(s_{1:t+B} \mid \mathcal{D}_{1:t}, \mathcal{D}^c_{t+1:t+B}) < \infty. \tag{78}$$

The second subclaim addresses the boundedness of the observation-space PIF under batched posterior updates, including the case where the batch consists entirely of contaminated observations. Specifically, we show by induction that the observation-space PIF remains bounded throughout the batch, provided that the observation-space PIF of each successive prior is bounded prior to the sequence of updates.

$$\mathrm{PIF}^B_{\boldsymbol{\theta}_{t+B}}(\boldsymbol{y}^c_{t+1:t+B}, \boldsymbol{y}_{1:t}) = \mathrm{KL}(P(\boldsymbol{\theta}_{t+B} \mid \mathcal{D}_{1:t+B}) \mid\mid P(\boldsymbol{\theta}_{t+B} \mid \mathcal{D}_{1:t}.\mathcal{D}^c_{t+1:t+B}) < \infty, \tag{79}$$

where $P(\boldsymbol{\theta}_{t+B} \mid ...)$ follows the WoLF posterior (whose sufficient statistics are updated following Algorithm 5).

With these two boundedness results in place, the proof of Theorem 4.2 follows directly: since the joint PIF

$$\mathrm{PIF}^B_{s_{t+B}, \boldsymbol{\theta}_{t+B}}(\boldsymbol{y}^c_{t+1:t+B}, \boldsymbol{y}_{1:t}) = \mathrm{PIF}^B_{s_{t+B}}(\boldsymbol{y}^c_{t+1:t+B}, \boldsymbol{y}_{1:t}) + \sum_{s_{1:t+B}} P(s_{1:t+B} \mid \mathcal{D}_{1:t+B}) \, \mathrm{PIF}^B_{\boldsymbol{\theta}_{t+B}}(\boldsymbol{y}^c_{t+1:t+B}, \boldsymbol{y}_{1:t}), \tag{80}$$

can be written as the sum of the state-space and observation-space PIFs by Proposition D.3, its boundedness is an immediate consequence.

Finally, as a side note, we know from Lemma C.1. of (Duran-Martin et al., 2024) that $\mathrm{PIF}_{\boldsymbol{\theta}_t}$ is bounded under the introduction of a weighting function $W(\cdot, \cdot)$ following the same conditions as those presented in Theorem 4.2 for one posterior update. A small caveat to note that in (Duran-Martin et al., 2024), the second-order boundedness assumption is $\sup_{\boldsymbol{y} \in \mathbb{R}^d} W(\boldsymbol{y}, \hat{\boldsymbol{y}})^2 \|\boldsymbol{y}\|_2 < \infty$. Instead, we write

$$\begin{aligned} \sup_{\boldsymbol{y} \in \mathbb{R}^d} W(\boldsymbol{y}, \hat{\boldsymbol{y}}) &\leq C_1 < \infty \\ \sup_{\boldsymbol{y} \in \mathbb{R}^d} W(\boldsymbol{y}, \hat{\boldsymbol{y}})^2 \|\boldsymbol{y}\|_2^2 &\leq C_2 < \infty \,, \end{aligned} \tag{81}$$

therefore, the condition

$$\sup_{\boldsymbol{y} \in \mathbb{R}^d} W(\boldsymbol{y}, \hat{\boldsymbol{y}})^2 \|\boldsymbol{y}\|_2 \leq \max\{C_1^2, C_2\} = C_3 < \infty \tag{82}$$

is implied by considering $\|\boldsymbol{y}\|_2 \geq 1$ and $\|\boldsymbol{y}\|_2 < 1$ separately.

**Lemma D.7.** *The batched iHMM with generalised posteriors presented in (12) for the auxiliary variable has bounded* $\mathrm{PIF}_{s_t}$ *for any weighting function* $W : \mathbb{R}^d \times \mathbb{R}^d \to \mathbb{R}$ *satisfying (9), i.e.,*

$$\mathrm{PIF}^B_{s_{t+B}}(\boldsymbol{y}^c_{t+1:t+B}, \boldsymbol{y}_{1:t}) < \infty. \tag{83}$$

*Proof.* It suffices to show the generalised posterior

$$P(s_{t+1:t+B} \mid s_t, \boldsymbol{\Phi}_t, \boldsymbol{\Psi}_t, \mathcal{D}_{1:t}, \mathcal{D}^c_{t+1:t+B})$$

with a contaminated batch $\mathcal{D}^c_{t+1:t+B}$ and weights $w^2_{\ell,t+b|t} = W(\boldsymbol{y}_{t+b}, \hat{\boldsymbol{y}}_{\ell,t+b|t})^2$ for $b \in [1, B]$ (with respect to the $\ell$-th state) assigns a non-zero mass.

More precisely, our formulation of the degenerate SHDP prior, in which transition probabilities are given by

$$P(s_{t+1} \mid s_t = \ell, \boldsymbol{\Phi}_t) = \frac{\hat{\alpha}_t \hat{\beta}_{\ell,t} + \kappa_t \mathbb{1}(s_{t+1} = \ell) + n_{\ell,s_{t+1},t}}{\hat{\alpha}_t + \kappa_t + n_{\ell,\cdot,t}} \tag{84}$$

shows that a non-zero mass (in an uncontaminated posterior) is only assigned to paths of the form

$$s_{1:t+B} \in \{\boldsymbol{\psi} : \boldsymbol{\psi} = (s_{1:t}, \ell, \ldots, \ell), \ell \leq \lceil t/B \rceil\}$$

since the transition from $s_t$ to $s_{t+1}$ determines the assignments for $s_{t+2:t+B}$ (see Section D.1).

We employ a similar tactic to Lemma D.4 and consider the log-difference between the unnormliased posteriors. Let $\nu^c_{\ell,t+B}$ denote the contaminated and $\nu_{\ell,t+B}$ the uncontaminated (unnormalised) weight for assigning the batch of states with size $B$, $s_{t+1}$ to $s_{t+B}$, to some state $\ell \in \{1, ..., \lceil t/B \rceil\}$,

$$\begin{aligned}
\log \nu^c_{\ell,t+B} &= \log \nu(s_{1:t+B}; \mathcal{D}_{1:t}, \mathcal{D}^c_{t+1:t+B}) \\
&= \left[ \sum_{b=1}^{B} w^2_{\ell,t+b|t} \log P(\boldsymbol{y}^c_{t+b} \mid \boldsymbol{x}_{t+b}, \boldsymbol{\Psi}_t, s_{t+b}) \right] \\
&\qquad + \log \sum_{s_{1:t}} P(s_{1:t} \mid \mathcal{D}_{1:t}) P(s_{t+1} \mid s_t, \boldsymbol{\Phi}_t) \prod_{b=2}^{B} \mathbb{1}(s_{t+b-1} = s_{t+b}) \,.
\end{aligned}$$

Again, we separate the contaminated weight into the summation of marginal likelihoods (inside square brackets) that are dependent on the contaminated observations and transition probabilities that are independent of contaminations. Denote

$$\begin{aligned}
\Delta^c_{\ell,k,t+b} = w^2_{\ell,t+b|t} &\log P(\boldsymbol{y}^c_{t+b} \mid \boldsymbol{x}_{t+b}, \boldsymbol{\Psi}_t, s_{t+b} = \ell) \\
&- w^2_{k,t+b|t} \log P(\boldsymbol{y}^c_{t+b} \mid \boldsymbol{x}_{t+b}, \boldsymbol{\Psi}_t, s_{t+b} = k)
\end{aligned} \tag{85}$$

as the log-difference between the lookahead marginals of $\boldsymbol{y}_{t+b}$ between state $\ell$ and $k$, i.e. the contribution that depends on the outlier. Then the difference in unnormalised weights have the form

$$\log \nu^c_{\ell,t+B} - \log \nu^c_{k,t+B} = \sum_{b=1}^{B} \Delta^c_{\ell,k,t+b} + C_{\ell k} \qquad \ell, k \leq \lceil t/B \rceil \tag{86}$$

where $C_{\ell k}$ is a finite constant independent of $\mathcal{D}^c_{t+1:t+B}$ (representing the likelihood contributed from the transition probabilities). Consider, for any $b \in [1, B]$,

$$\begin{aligned}
\Delta^c_{\ell,k,t+b} = \frac{-w^2_{\ell,t+b|t}}{2} &\left[ (\boldsymbol{y}^c_{t+b} - \hat{\boldsymbol{y}}_{\ell,t+b|t})^\top \mathbf{S}^{-1}_{\ell,t+b|t} (\boldsymbol{y}^c_{t+b} - \hat{\boldsymbol{y}}_{\ell,t+b|t}) + \log\det \mathbf{S}_{\ell,t+b|t} + d\log(2\pi) \right] \\
&+ \\
\frac{w^2_{k,t+b|t}}{2} &\left[ (\boldsymbol{y}^c_{t+b} - \hat{\boldsymbol{y}}_{k,t+b|t})^\top \mathbf{S}^{-1}_{k,t+b|t} (\boldsymbol{y}^c_{t+b} - \hat{\boldsymbol{y}}_{k,t+b|t}) + \log\det \mathbf{S}_{k,t+b|t} + d\log(2\pi) \right]
\end{aligned}$$

where $\mathbf{S}_{k,t+b|t}$ is defined in (25). As before, the difference in log-marginals comprises four terms

$$\begin{aligned}
2\Delta^c_{\ell,k,t+b} = {} & w^2_{k,t+b|t} (\boldsymbol{y}^c_{t+b})^\top \mathbf{S}^{-1}_{k,t+b|t} (\boldsymbol{y}^c_{t+b}) - w^2_{\ell,t+b|t} (\boldsymbol{y}^c_{t+b})^\top \mathbf{S}^{-1}_{\ell,t+b|t} (\boldsymbol{y}^c_{t+b}) && (87) \\
& + 2 w^2_{\ell,t+b|t} \hat{\boldsymbol{y}}^\top_{\ell,t+b|t} \mathbf{S}^{-1}_{\ell,t+b|t} \boldsymbol{y}^c_{t+b} - 2 w^2_{k,t+b|t} \hat{\boldsymbol{y}}^\top_{k,t+b|t} \mathbf{S}^{-1}_{k,t+b|t} \boldsymbol{y}^c_{t+b} && (88) \\
& + w^2_{k,t+b|t} \hat{\boldsymbol{y}}^\top_{k,t+b|t} \mathbf{S}^{-1}_{k,t+b|t} \hat{\boldsymbol{y}}_{k,t+b|t} - w^2_{\ell,t+b|t} \hat{\boldsymbol{y}}^\top_{\ell,t+b|t} \mathbf{S}^{-1}_{\ell,t+b|t} \hat{\boldsymbol{y}}_{\ell,t+b|t} && (89) \\
& + w^2_{k,t+b|t} \log\det \mathbf{S}_{k,t+b|t} - w^2_{\ell,t+b|t} \log\det \mathbf{S}_{\ell,t+b|t} + \frac{d\log(2\pi)}{2} (w^2_{k,t+b|t} - w^2_{\ell,t+b|t}) \,, && (90)
\end{aligned}$$

with (89) and (90) independent to $\boldsymbol{y}^c_{t+b}$. Hence, we focus our attention on (87) and (88).

Since for any value $x, y$, we have $|x+y| \leq |x| + |y|$, the difference in log-marginals can be written as

$$|\Delta^c_{\ell,k,t+b}| \leq \frac{1}{2}\bigg(|(87)| + |(88)| + |(89)| + |(90)|\bigg)$$

where we will attempt to bound the absolute log-differences by prove boundedness for each term within this bound.

Now define $\lambda_{\max} : \mathbb{R}^{d\times d} \to \mathbb{R}$ as the function that maps any square matrix to its largest eigenvalue. We first show that the second-order term (87) is bounded.

$$\begin{aligned}
|(87)| &\leq |w^2_{k,t+b|t}||(\boldsymbol{y}^c_{t+b})^\top \mathbf{S}^{-1}_{k,t+b|t}(\boldsymbol{y}^c_{t+b})| + |w^2_{\ell,t+b|t}||(\boldsymbol{y}^c_{t+b})^\top \mathbf{S}^{-1}_{\ell,t+b|t}(\boldsymbol{y}^c_{t+b})| \\
&\leq |w^2_{k,t+b|t}|\lambda_{\max}(\mathbf{S}^{-1}_{k,t+b|t})\|\boldsymbol{y}^c_{t+b}\|_2^2 + |w^2_{\ell,t+b|t}|\lambda_{\max}(\mathbf{S}^{-1}_{\ell,t+b|t})\|\boldsymbol{y}^c_{t+b}\|_2^2 \\
&\leq (\lambda_{\max}(\mathbf{S}^{-1}_{k,t+b|t}) + \lambda_{\max}(\mathbf{S}^{-1}_{\ell,t+b|t}))C_2 < \infty
\end{aligned}$$

where the last inequality comes directly from the assumption of $\sup_{\boldsymbol{y}_{t+b}\in\mathbb{R}^d} w^2_{k,t+b|t}\|\boldsymbol{y}_{t+b}\|_2^2 < C_2 < \infty$ for some $C_2 > 0$. Furthermore, $\mathbf{S}_{k,t+b|t}, \mathbf{S}_{\ell,t+b|t} \succ 0$ and real-valued, hence $0 < \lambda_{\max}(\mathbf{S}^{-1}_{k,t+b|t}),\ \lambda_{\max}(\mathbf{S}^{-1}_{\ell,t+b|t}) < \infty$.

Moving onto the first-order term (88), we have

$$\begin{aligned}
|(88)| &\leq 2|w^2_{\ell,t+b|t}||\hat{\boldsymbol{y}}^\top_{\ell,t+b|t}\mathbf{S}^{-1}_{\ell,t+b|t}\boldsymbol{y}^c_{t+b}| + 2|w^2_{k,t+b|t}||\hat{\boldsymbol{y}}^\top_{k,t+b|t}\mathbf{S}^{-1}_{k,t+b|t}\boldsymbol{y}^c_{t+b}| \\
&\leq 2|w^2_{\ell,t+b|t}|\|\mathbf{S}^{-\top}_{\ell,t+b|t}\hat{\boldsymbol{y}}_{\ell,t+b|t}\|_2\|\boldsymbol{y}^c_{t+b}\|_2 + 2|w^2_{k,t+b|t}|\|\mathbf{S}^{-\top}_{k,t+b|t}\hat{\boldsymbol{y}}_{k,t+b|t}\|_2\|\boldsymbol{y}^c_{t+b}\|_2 \\
&\leq 2C_3(\|\mathbf{S}^{-\top}_{\ell,t+b|t}\hat{\boldsymbol{y}}_{\ell,t+b|t}\|_2 + \|\mathbf{S}^{-\top}_{k,t+b|t}\hat{\boldsymbol{y}}_{k,t+b|t}\|_2) < \infty
\end{aligned}$$

where we used $|\boldsymbol{x}^\top\boldsymbol{y}| \leq \|\boldsymbol{x}\|_2\|\boldsymbol{y}\|_2$ in the second inequality, and the last inequality comes from the assumption in (82). Combining the two bounds, we have $|\Delta^c_{\ell,k,t+b}| < \infty$, and therefore

$$|(86)| = |\log\nu^c_{\ell,t+B} - \log\nu^c_{k,t+B}| \leq \sum_{b=1}^{B}|\Delta^c_{\ell,j,t+b}| + |C_{\ell k}| \leq C_{\ell,k,t+B} < \infty$$

since all terms in the sum are bounded. Let $C_{t+B} = \max_{\ell,\ell'} C_{\ell,\ell',t+B} < \infty$, be an upper bound over the log-difference in the contaminated unnormalised weights between any two states. The normalised contaminated distribution over the state trajectories is given by the normalisation

$$P(s_{t+1:t+B} \mid s_t, \boldsymbol{\Psi}_t, \boldsymbol{\Phi}_t, \mathcal{D}^c_{t+1:t+B}) = \frac{\nu^c_{\ell,t+B}}{\sum_{\ell'}\nu^c_{\ell',t+B}} = \left[1 + \sum_{\ell'\neq l}\exp(\log\nu^c_{\ell',t+B} - \log\nu^c_{\ell,t+B})\right]^{-1}.$$

Since $\log\nu^c_{\ell,t+B} - \log\nu^c_{k,t+B} \in [-C_{t+B}, C_{t+B}]$, we can bound the contaminated probabilities

$$0 < 1 + \exp(-C_{t+B}) \leq 1 + \exp(\log\nu^c_{\ell,t+B} - \log\nu^c_{k,t+B}) \leq 1 + \exp(C_{t+B}) \tag{91}$$

$$\implies \frac{1}{1+\exp(C_{t+B})} \leq P(s_{t+1:t+B} \mid s_t, \boldsymbol{\Psi}_t, \boldsymbol{\Phi}_t, \mathcal{D}^c_{t+1:t+B}) \leq \frac{1}{1+\exp(-C_{t+B})} < \infty \tag{92}$$

and conclude the contaminated posterior has non-zero mass over all possible path trajectories $s_{1:t+B}$. It therefore follows that the generalised $\text{PIF}^B_{s_{t+B}}$ is finite. □

**Lemma D.8.** *The iHMM with batched generalised posteriors in Algorithm 1 has a bounded* $\text{PIF}^B_{\boldsymbol{\theta}_t}$ *i.e.,*

$$\text{PIF}^B_{\boldsymbol{\theta}_{t+B}}(\boldsymbol{y}^c_{t+1:t+B}, \boldsymbol{y}_{1:t}) < \infty, \tag{93}$$

*for any weighting function* $W : \mathbb{R}^d \times \mathbb{R}^d \to \mathbb{R}$ *satisfying (9).*

*Proof.* We aim to show the posterior distribution obtained after multiple (outlier) updates remain bounded in observation PIF. Consider the batch from $t+1$ to $t+B$ of size $B$, such that $s_{t+1} = s_{t+2} = ... = s_{t+B}$. Let $b \in [1, B]$ such that $\boldsymbol{y}_{t+b}^c$ be the first contaminated sample within the batch, that is, we have the following observations in the batch

$$(\boldsymbol{y}_t, ..., \boldsymbol{y}_{t+b-1}, \boldsymbol{y}_{t+b}^c, ..., \boldsymbol{y}_{t+B})\,.$$

Given these observations $(\mathcal{D}_{t+1:t+b-1}, \mathcal{D}_{t+b:t+B}^c)$, we show that

$$\begin{aligned}&\mathrm{PIF}^B_{\boldsymbol{\theta}_{t+B}}(\boldsymbol{y}^c_{t+b:t+B}, \boldsymbol{y}_{1:t+b-1})\\ &\quad= \mathrm{KL}(P(\boldsymbol{\theta}_{t+B} \mid s_{t+B}, \boldsymbol{\Psi}_{t-1}, \mathcal{D}_{t+1:t+B})||P(\boldsymbol{\theta}_{t+B} \mid s_{t+B}, \boldsymbol{\Psi}_{t-1}, \mathcal{D}_{t+1:t+b-1}, \mathcal{D}^c_{t+b:t+B})) < \infty\end{aligned} \tag{94}$$

via induction. Since the observations from $\boldsymbol{y}_{1:t+b-1}$ are uncontaminated and identical on both sides, the PIF is equivalent to

$$\begin{aligned}&\mathrm{PIF}^B_{\boldsymbol{\theta}_{t+B}}(\boldsymbol{y}^c_{t+b:t+B}, \boldsymbol{y}_{1:t+b-1})\\ &\quad= \mathrm{KL}(P(\boldsymbol{\theta}_{t+B} \mid s_{t+B}, \boldsymbol{\Psi}_{t+b-1}, \mathcal{D}_{t+b:t+B})||P(\boldsymbol{\theta}_{t+B} \mid s_{t+B}, \boldsymbol{\Psi}_{t+b-1}, \mathcal{D}^c_{t+b:t+B})).\end{aligned} \tag{95}$$

Our base case is the one-step WoLF posterior update, that is

$$\mathrm{KL}(P(\boldsymbol{\theta}_{t+b} \mid s_{t+b}, \boldsymbol{\Psi}_{t+b-1}, \mathcal{D}_{t+b})||P(\boldsymbol{\theta}_{t+b} \mid s_{t+b}, \boldsymbol{\Psi}_{t+b-1}, \mathcal{D}^c_{t+b})) < \infty \tag{96}$$

which we know is bounded from Lemma C.2 in (Duran-Martin et al., 2024). The inductive step for some $n \in [t+b+1, t+B-2]$ assumes

$$\begin{aligned}&\mathrm{KL}(P(\boldsymbol{\theta}_n \mid s_n, \boldsymbol{\Psi}_{n-1}, \mathcal{D}_n)||P(\boldsymbol{\theta}_n \mid s_n, \boldsymbol{\Psi}^c_{n-1}, \mathcal{D}^c_n))\\ &= \mathrm{KL}(\mathcal{N}(\boldsymbol{\theta} \mid \boldsymbol{\mu}_n, \boldsymbol{\Sigma}_n) \;||\; \mathcal{N}(\boldsymbol{\theta} \mid \boldsymbol{\mu}^c_n, \boldsymbol{\Sigma}^c_n))\\ &= \frac{1}{2}\left[\mathrm{Tr}((\boldsymbol{\Sigma}_n)^{-1}\boldsymbol{\Sigma}^c_n) - d + (\boldsymbol{\mu}^c_n - \boldsymbol{\mu}_n)^\top(\boldsymbol{\Sigma}_n)^{-1}(\boldsymbol{\mu}^c_n - \boldsymbol{\mu}_n) + \log\left(\frac{\det \boldsymbol{\Sigma}_n}{\det \boldsymbol{\Sigma}^c_n}\right)\right] < \infty\,,\end{aligned} \tag{97}$$

where $\boldsymbol{\Phi}^c_n = (\boldsymbol{\mu}^c_{\ell,n}, \boldsymbol{\Sigma}^c_{\ell,n})_{\ell=1}^{L_n+1}$ has the contaminated posterior means from previous outliers and we abused notation with $\boldsymbol{\mu}_n$ to refer to $\boldsymbol{\mu}_{s_{t+B},n}$ (the state is fixed throughout the batch so dependence on $s_{t+B}$ is implied, $\forall n, s_n = s_{t+1}$). Recall the update rules to the covariance matrices $\boldsymbol{\Sigma}_n$,

$$\begin{aligned}\mathbf{S}^c_{n+1} &= \mathbf{F}_{n+1}\boldsymbol{\Sigma}_n\mathbf{F}^\top_{n+1} + \mathbf{R}_{n+1}/w^2_{n+1}, & \lim_{\|\boldsymbol{y}^c_t\|^2_2\to\infty} \mathbf{S}^c_{n+1} &= \infty\\ \mathbf{K}^c_{n+1} &= \boldsymbol{\Sigma}_n\mathbf{F}^\top_{n+1}(\mathbf{S}^c_{n+1})^{-1}, & \lim_{\|\boldsymbol{y}^c_t\|^2_2\to\infty} \mathbf{K}^c_{n+1} &= 0\\ \boldsymbol{\Sigma}^c_{n+1} &= \boldsymbol{\Sigma}_n - \mathbf{K}^c_{n+1}\mathbf{S}^c_{n+1}(\mathbf{K}^c_{n+1})^\top, & \lim_{\|\boldsymbol{y}^c_t\|^2_2\to\infty} \boldsymbol{\Sigma}^c_{n+1} &= \boldsymbol{\Sigma}_n.\end{aligned} \tag{98}$$

In general, $\boldsymbol{\Sigma}^c_{n+1} \neq \boldsymbol{\Sigma}_{n+1}$, but it should be clear that

$$\boldsymbol{\Sigma}_n \succeq \boldsymbol{\Sigma}_{n+1}, \boldsymbol{\Sigma}^c_{n+1} \succeq 0. \tag{99}$$

In the following, we show that the posterior for $\boldsymbol{\theta}_{n+1}$ also has a bounded PIF.

The inductive step assumes a finite KL divergence, this implies several useful assumptions under the context of a LG model. We now consider the case for $n+1$

$$\begin{aligned}&\mathrm{KL}(P(\boldsymbol{\theta}_{n+1} \mid s_{n+1}, \boldsymbol{\Psi}_n, \mathcal{D}_{n+1})||P(\boldsymbol{\theta}_{n+1} \mid s_{n+1}, \boldsymbol{\Psi}_n, \mathcal{D}^c_{n+1}))\\ &= \frac{1}{2}\Bigg[\underbrace{\mathrm{Tr}((\boldsymbol{\Sigma}_{n+1})^{-1}\boldsymbol{\Sigma}^c_{n+1})}_{(a)} - d + \underbrace{(\boldsymbol{\mu}^c_{n+1} - \boldsymbol{\mu}_{n+1})^\top(\boldsymbol{\Sigma}_{n+1})^{-1}(\boldsymbol{\mu}^c_{n+1} - \boldsymbol{\mu}_{n+1})}_{(b)} + \underbrace{\log\left(\frac{\det \boldsymbol{\Sigma}_{n+1}}{\det \boldsymbol{\Sigma}^c_{n+1}}\right)}_{(c)}\Bigg]\end{aligned}$$

and show boundedness term by term. Before bounding (a) and (c), we rewrite[5] the covariance update as

$$\begin{aligned}(\boldsymbol{\Sigma}_{n+1})^{-1} &= \boldsymbol{\Sigma}^{-1}_n + w^2_{n+1}\mathbf{H}^\top_{n+1}\mathbf{R}^{-1}_{n+1}\mathbf{H}_{n+1}\\ (\boldsymbol{\Sigma}^c_{n+1})^{-1} &= (\boldsymbol{\Sigma}^c_n)^{-1} + (w^c_{n+1})^2\mathbf{H}^\top_{n+1}\mathbf{R}^{-1}_{n+1}\mathbf{H}_{n+1}\end{aligned} \tag{100}$$

[5] In standard Kalman filters, the update rule for the precision is written as $\boldsymbol{\Sigma}^{-1}_{n+1|n} + w^2_{n+1}\mathbf{H}_{n+1}\mathbf{R}^{-1}_{n+1}\mathbf{H}_{n+1}$, but the LG assumption yields $\boldsymbol{\Sigma}_{n+1|n} = \boldsymbol{\Sigma}_n$.

by the Woodbury identity and write

$$\mathbf{M}_{n+1} = \mathbf{H}_{n+1}^\top \mathbf{R}_{n+1}^{-1} \mathbf{H}_{n+1}.$$

Consider (a), where

$$\mathrm{Tr}(\mathbf{\Sigma}_{n+1}^{-1}\mathbf{\Sigma}_{n+1}^c) = \mathrm{Tr}(\mathbf{\Sigma}_n^{-1}\mathbf{\Sigma}_{n+1}^c) + w_{n+1}^2 \mathrm{Tr}(\mathbf{M}_{n+1}\mathbf{\Sigma}_{n+1}^c). \tag{101}$$

The first term is bounded by the inductive hypothesis, using $\mathbf{\Sigma}_n^c \succeq \mathbf{\Sigma}_{n+1}^c$,

$$\mathrm{Tr}(\mathbf{\Sigma}_n^{-1}\mathbf{\Sigma}_{n+1}^c) \le \mathrm{Tr}(\mathbf{\Sigma}_n^{-1}\mathbf{\Sigma}_n^c) < \infty. \tag{102}$$

Using the same property, see that

$$\mathrm{Tr}(\mathbf{M}_{n+1}\mathbf{\Sigma}_{n+1}^c) \le \mathrm{Tr}(\mathbf{M}_{n+1}\mathbf{\Sigma}_n^c) \le \lambda_{\max}(\mathbf{M}_{n+1})\mathrm{Tr}(\mathbf{\Sigma}_n^c). \tag{103}$$

From the inductive step, we have the assumption

$$\begin{aligned} \frac{\mathrm{Tr}(\mathbf{\Sigma}_n^c)}{\lambda_{\max}(\mathbf{\Sigma}_n)} &\le \mathrm{Tr}(\mathbf{\Sigma}_n^{-1}\mathbf{\Sigma}_n^c) \\ \implies \mathrm{Tr}(\mathbf{\Sigma}_n^c) &\le \lambda_{\max}(\mathbf{\Sigma}_n)\mathrm{Tr}(\mathbf{\Sigma}_n^{-1}\mathbf{\Sigma}_n^c) < \infty. \end{aligned} \tag{104}$$

Hence, combining, we have

$$\mathrm{Tr}(M_{n+1}\Sigma_{n+1}^c) \le \lambda_{\max}(\mathbf{M}_{n+1})\lambda_{\max}(\mathbf{\Sigma}_n)\mathrm{Tr}(\mathbf{\Sigma}_n^{-1}\mathbf{\Sigma}_n^c) < \infty, \tag{105}$$

and (a) is bounded. Now consider (c), which admits a similar manipulation on the posterior covariance estimate,

$$\begin{aligned} \log\frac{\det \mathbf{\Sigma}_{n+1}}{\det \mathbf{\Sigma}_{n+1}^c} &= \log\det \mathbf{\Sigma}_{n+1} - \log\det \mathbf{\Sigma}_{n+1}^c \\ &= \log\det \mathbf{\Sigma}_{n+1} - \log(\det(\mathbf{\Sigma}_{n+1}^c)^{-1})^{-1} \\ &= \log\det \mathbf{\Sigma}_{n+1} + \log\det(\mathbf{\Sigma}_{n+1}^c)^{-1}, \end{aligned} \tag{106}$$

therefore it suffices to bound $\log(\det(\mathbf{\Sigma}_{n+1}^c)^{-1})$. We make use of the inequality $\log\det(\mathbf{A}+\mathbf{B}) \le \log\det(\mathbf{A})+\mathrm{Tr}(\mathbf{A}^{-1}\mathbf{B})$, for any $\mathbf{A} \succ 0$ and $\mathbf{B} \succeq 0$ (see Boyd & Vandenberghe (2004, §3.1.5 on the concavity of $\log\det$) and Bhatia (2009)),

$$\log\det(\mathbf{\Sigma}_{n+1}^c)^{-1} = \log\det\left((\mathbf{\Sigma}_n^c)^{-1} + (w_{n+1}^c)^2\mathbf{H}_{n+1}^\top\mathbf{R}_{n+1}^{-1}\mathbf{H}_{n+1}\right) \tag{107}$$

$$\le \log\det(\mathbf{\Sigma}_n^c)^{-1} + (w_{n+1}^c)^2\mathrm{Tr}\left(\mathbf{\Sigma}_n^c\mathbf{M}_{n+1}\right) \tag{108}$$

since $\mathbf{\Sigma}_n^c \succ 0$ by virtue of $\mathbf{\Sigma}_{(0)} \succ 0$ and $\mathbf{M}_{n+1} = \mathbf{H}_{n+1}^\top\mathbf{R}_{n+1}^{-1}\mathbf{H}_{n+1} \succeq 0$ (it is possible for $\mathbf{H}_{n+1} = 0$ as the input feature). From the inductive hypothesis, we know

$$\log\frac{\det \mathbf{\Sigma}_n}{\det \mathbf{\Sigma}_n^c} = \log\det \mathbf{\Sigma}_{n+1} + \log\det(\mathbf{\Sigma}_n^c)^{-1} < \infty \tag{109}$$

$$\implies \log\det(\mathbf{\Sigma}_n^c)^{-1} < \infty. \tag{110}$$

Coupled with the bound in (104), we conclude (107) is bounded and therefore (c) is bounded.

Finally, consider (b) and see that

$$(b) = (\boldsymbol{\mu}_{n+1}^c - \boldsymbol{\mu}_{n+1})^\top(\mathbf{\Sigma}_{n+1})^{-1}(\boldsymbol{\mu}_{n+1}^c - \boldsymbol{\mu}_{n+1}) \tag{111}$$

$$\le \lambda_{\max}((\mathbf{\Sigma}_{n+1})^{-1})\|\boldsymbol{\mu}_{n+1}^c - \boldsymbol{\mu}_{n+1}\|_2^2 \tag{112}$$

$$= \lambda_{\max}(\mathbf{\Sigma}_{n+1}^{-1})\|\boldsymbol{\mu}_{n+1}^c - \boldsymbol{\mu}_{n+1}\|_2^2\,, \tag{113}$$

and

$$\boldsymbol{\mu}_{n+1}^c - \boldsymbol{\mu}_{n+1} = (\boldsymbol{\mu}_n^c - \boldsymbol{\mu}_n) + (w_{n+1}^c)^2\mathbf{K}_{n+1}^c(\boldsymbol{y}_{n+1}^c - \hat{\boldsymbol{y}}_{n+1}^c) - w_{n+1}^2\mathbf{K}_{n+1}(\boldsymbol{y}_{n+1} - \hat{\boldsymbol{y}}_{n+1})\,. \tag{114}$$

It suffices to bound

$$\begin{aligned}\|\boldsymbol{\mu}_{n+1}^c - \boldsymbol{\mu}_{n+1}\|_2^2 &= \|(\boldsymbol{\mu}_n^c - \boldsymbol{\mu}_n) + (w_{n+1}^c)^2 \mathbf{K}_{n+1}^c(\boldsymbol{y}_{n+1}^c - \hat{\boldsymbol{y}}_{n+1}^c) - w_{n+1}^2 \mathbf{K}_{n+1}(\boldsymbol{y}_{n+1} - \hat{\boldsymbol{y}}_{n+1})\|_2^2 \\ &\leq 2\Big( \|(\boldsymbol{\mu}_n^c - \boldsymbol{\mu}_n)\|_2^2 + \|(w_{n+1}^c)^2 \mathbf{K}_{n+1}^c \boldsymbol{y}_{n+1}^c\|_2^2 + \|(w_{n+1}^c)^2 \mathbf{K}_{n+1}^c \hat{\boldsymbol{y}}_{n+1}^c\|_2^2 \\ &\quad + \|w_{n+1}^2 \mathbf{K}_{n+1} \boldsymbol{y}_{n+1}\|_2^2 + \|w_{n+1}^2 \mathbf{K}_{n+1} \hat{\boldsymbol{y}}_{n+1}\|_2^2 \Big).\end{aligned} \tag{115}$$

The last two terms are uncontaminated, therefore, we show that the first three terms are all bounded. The inductive step implies

$$\lambda_{\max}(\boldsymbol{\Sigma}_n^{-1})\|\boldsymbol{\mu}_n^c - \boldsymbol{\mu}_n\|_2^2 < \infty \implies \|\boldsymbol{\mu}_n^c - \boldsymbol{\mu}_n\|_2^2 < \infty.$$

Therefore, the second term is also bounded as

$$|w_{n+1}^c|^2 \|\mathbf{K}_{n+1}^c \boldsymbol{y}_{n+1}^c\|_2^2 \leq |w_{n+1}^c|^2 \|\boldsymbol{y}_{n+1}^c\|_2^2 \, \|\mathbf{K}_{n+1}^c\|_2^2 \,, \tag{116}$$

and

$$\begin{aligned}\|\mathbf{K}_{n+1}^c\|_2 &\leq \|\boldsymbol{\Sigma}_n\|_2 \|\mathbf{F}_{n+1}^\top\|_2 \|(\mathbf{S}_{n+1}^c)^{-1}\|_2 \\ &= \frac{\|\boldsymbol{\Sigma}_n\|_2 \|\mathbf{F}_{n+1}^\top\|_2}{\lambda_{\min}(\mathbf{S}_{n+1}^c)} \\ &= \frac{\|\boldsymbol{\Sigma}_n^c\|_2 \|\mathbf{F}_{n+1}^\top\|_2}{\lambda_{\min}(\mathbf{R}_{n+1}/w_{n+1}^2)},\end{aligned} \tag{117}$$

where we used the fact that $\mathbf{S}_{n+1}^c - \mathbf{R}_{n+1}/w_{n+1}^2 = \mathbf{F}_{n+1}\boldsymbol{\Sigma}_n \mathbf{F}_{n+1}^\top \succeq 0$. Combining, we have

$$|w_{n+1}^c|^2 \|\mathbf{K}_{n+1}^c \boldsymbol{y}_{n+1}^c\|_2^2 \leq \frac{\|\boldsymbol{\Sigma}_n^c\|_2^2 \, \|\mathbf{F}_{n+1}^\top\|_2^2 \, |w_{n+1}^c|^4 \, \|\boldsymbol{y}_{n+1}^c\|_2^2}{\lambda_{\min}(\mathbf{R}_{n+1})^2} \tag{118}$$

$$\leq \frac{\|\boldsymbol{\Sigma}_n^c\|_2^2 \, \|\mathbf{F}_{n+1}^\top\|_2^2 \, |w_{n+1}^c|^2 \, C_2}{\lambda_{\min}(\mathbf{R}_{n+1})^2}. \tag{119}$$

Since $\|\boldsymbol{\Sigma}_n^c\|_2 < \infty$ by the inductive hypothesis, the second term is bounded. Similarly, the third term is bounded as

$$\|w_{n+1}^c \mathbf{K}_{n+1}^c \hat{\boldsymbol{y}}_{n+1}^c\|_2^2 = \|w_{n+1}^c \mathbf{K}_{n+1}^c \mathbf{F}_{n+1} \boldsymbol{\mu}_n^c\|_2^2 \tag{120}$$

$$\leq |w_{n+1}^c|^2 \|\mathbf{F}_{n+1}\|_2^2 \, \|\boldsymbol{\mu}_n^c\|_2^2 \|\mathbf{K}_{n+1}^c\|_2^2 \tag{121}$$

$$\leq \frac{\|\boldsymbol{\Sigma}_n^c\|_2^2 \, \|\mathbf{F}_{n+1}\|_2^2 \, \|\mathbf{F}_{n+1}^\top\|_2^2 \, |w_{n+1}^c|^4 \, \|\boldsymbol{\mu}_n^c\|_2^2}{\lambda_{\min}(\mathbf{R}_{n+1})^2}\,, \tag{122}$$

where $\|\boldsymbol{\mu}_n^c\|_2^2 = \|(\boldsymbol{\mu}_n^c - \boldsymbol{\mu}_n) + \boldsymbol{\mu}_n\|_2^2 \leq 2\|\boldsymbol{\mu}_n^c - \boldsymbol{\mu}_n\|_2^2 + 2\|\boldsymbol{\mu}_n\|_2^2 < \infty$ from the inductive step. Here, we conclude that term (b) is bounded.

By the rules of induction, we conclude that the KL divergence at step $t + B$ remains finite, and by extent the $\mathrm{PIF}^B_{\boldsymbol{\theta}_{t+B}}(\boldsymbol{y}^c_{t+b:t+B}, \boldsymbol{y}_{1:t+b-1}) < \infty$. □

**Proof of Theorem 4.2**. Follows directly from Lemmas D.7 to D.8.

# E. Additional Experiments

## E.1. Metrics

**Effective sample size** Given the particle weights $\boldsymbol{\omega} = \{\omega^{(i)}\}_{i=1}^N$ such that

$$\sum_{i=1}^N \omega^{(i)} = 1, \tag{123}$$

the ESS is given by

$$\mathrm{ESS}(\boldsymbol{\omega}) = \frac{1}{\sum_{i=1}^N (\omega^{(i)})^2}. \tag{124}$$

| Model | Synthetic | Electricity | OFI |
|---|---|---|---|
| BR-iHMM | **46.1±0.00301** | **0.47±0.04** | **0.616±0.082** |
| WoLF-iHMM | 103.8±0.01155 | 0.63±0.03 | 0.623±0.089 |
| iHMM | 101.7±0.02636 | 0.57±0.03 | 0.620±0.080 |
| offline-iHMM | 2.9 | 0.32 | 0.552 |
| BOCD | 123.12 ± 0.01365 | 0.80 ± 0.11 | 0.733 |

*Table E.5.* Forecast performance in terms of one-step ahead RMSE (Mean ± stdev) over 100 runs. The most precise online model is bolded. The BOCD implementation for the OFI experiment is from Tsaknaki et al. (2025) and has no randomisation. offline-iHMM is only ran once.

**Root Mean Squared Error (RMSE).** The root mean squared error measures the square root of the average squared difference between the predicted and true values. It penalises larger deviations more heavily and is defined as

$$\text{RMSE} = \sqrt{\frac{1}{T}\sum_{t=1}^{T}(\boldsymbol{y}_t - \hat{\boldsymbol{y}}_t)^2}, \tag{125}$$

where $\boldsymbol{y}_t$ and $\hat{\boldsymbol{y}}_t$ denote the true and predicted observations at time $t$, respectively.

**Mean Absolute Error (MAE).** The mean absolute error measures the average magnitude of the absolute differences between predicted and true values, treating all errors equally regardless of direction:

$$\text{MAE} = \frac{1}{T}\sum_{t=1}^{T}|\boldsymbol{y}_t - \hat{\boldsymbol{y}}_t|. \tag{126}$$

**Detection Delay (DD).** We define the DD as follows. For a set of predicted MAP states $(s_t)_{t\leq T}$, we calculate the changepoints such that

$$\mathcal{C} = \{t \geq 2 : s_t \neq s_{t-1}\}. \tag{127}$$

Let $\mathcal{C}^* = \{t : s_t^* \neq s_{t-1}^*\}$ be the set of true CPs. The DD is then defined as

$$\text{DD}(\mathcal{C}, \mathcal{C}^*) = \frac{1}{|\mathcal{C}^*|}\sum_{t^*\in\mathcal{C}^*}\min_{t\in\mathcal{C}, t\geq t^*}(t - t^*)\,, \tag{128}$$

that is the time required from the true CP to an estimated CP. Indeed, missing one single regime will lead to a large DD if the regime missed has a long run-length. Intuitively, the detection delay is calculated as the average number of observations after a true CP for the HMM to recognise a change in regimes.

**Regime classification errors.** We calculate the recall and precision of the segmentation problem, treating correct state inference as TP (true positive), incorrect state inference as either FN (false negative) or FP (false positive). For non-trivial state labelling, we employ the Algorithm 9 to obtain TP, FN, and FP values, given the predicted states, and a benchmark set of CPs.

**Algorithm 9** State matching and labelling

**Input:** true change-points $\mathcal{C}^\star = \{t_1^*, \dots, t_K^*\}$ (0-based, sorted), predicted state sequence $\hat{s}_{1:T}$, run threshold $L \in \mathbb{N}_{>0}$
**Output:** global counts $\mathrm{TP}, \mathrm{FP}, \mathrm{FN}$ and rates $\mathrm{TPR}, \mathrm{PPV}$
Initialise $\mathrm{TP} \leftarrow 0,\ \mathrm{FP} \leftarrow 0,\ \mathrm{FN} \leftarrow 0$
Define boundaries $B \leftarrow [0, t_1^*, t_2^*, \dots, t_K^*, T]$
Track prev majority label $\ell_{\mathrm{maj}}^{\mathrm{prev}} = -1$
**for** $i = 0$ **to** $|B| - 2$ **do**
  $a \leftarrow B[i],\ b \leftarrow B[i+1]$ (segment indices: $t \in [a, b-1]$)
  $S \leftarrow \hat{s}_{a:b-1}$ (subsequence of predicted states in segment)
  **if** $|S| = 0$ **then**
    **continue**
  **end if**
  Compute majority label $\ell_{\mathrm{maj}} \leftarrow \arg\max_\ell \#\{t \in [a, b-1] : \hat{s}_t = \ell\}$
  `failed_detection` = $\ell_{\mathrm{maj}}^{\mathrm{prev}} = \ell_{\mathrm{maj}}$
  **if** `failed_detection` **then**
    $\mathrm{FN} \leftarrow \mathrm{FN} + \#\{t \in [a, b-1] : \hat{s}_t = \ell_{\mathrm{maj}}\}$
  **else**
    $\mathrm{TP} \leftarrow \mathrm{TP} + \#\{t \in [a, b-1] : \hat{s}_t = \ell_{\mathrm{maj}}\}$
  **end if**
  // Scan maximal consecutive runs of minority labels inside the segment
  $j \leftarrow 0$
  **while** $j < |S|$ **do**
    **if** ($S[j] = \ell_{\mathrm{maj}}$ and $\neg$`failed_detection`) or ($S[j] \neq \ell_{\mathrm{maj}}$ and `failed_detection`) **then**
      $j \leftarrow j + 1$
      **continue**
    **end if**
    $k \leftarrow j$
    **while** $k < |S|$ **and** $S[k] = S[j]$ **do**
      $k \leftarrow k + 1$
    **end while**
    $r \leftarrow k - j$ (run length)
    **if** $r \geq L$ **then**
      $\mathrm{FP} \leftarrow \mathrm{FP} + r$
    **else**
      $\mathrm{FN} \leftarrow \mathrm{FN} + r$
    **end if**
    $j \leftarrow k$
  **end while**
**end for**
$\mathrm{TPR} \leftarrow \dfrac{\mathrm{TP}}{\mathrm{TP} + \mathrm{FN}}$ **if** $(\mathrm{TP} + \mathrm{FN}) > 0$ **else** $0$
$\mathrm{PPV} \leftarrow \dfrac{\mathrm{TP}}{\mathrm{TP} + \mathrm{FP}}$ **if** $(\mathrm{TP} + \mathrm{FP}) > 0$ **else** $0$
**return** $\{\mathrm{TP}, \mathrm{FP}, \mathrm{FN}, \mathrm{TPR}, \mathrm{PPV}\}$

TPR (recall) and PPV (precision) are calculated as

$$\mathrm{TPR} = \frac{\mathrm{TP}}{\mathrm{TP} + \mathrm{FN}}, \qquad \mathrm{PPV} = \frac{\mathrm{TP}}{\mathrm{TP} + \mathrm{FP}}.$$

### E.2. Runtimes

As the `BR-iHMM` model can skip sampling full trajectories given a batch size $B > 1$, the algorithm is computationally quicker than `iHMM` and `WoLF-iHMM`. This advantage is reflected in the average run-time across the experiments in Table

E.6.

| Dataset | BR-iHMM | WoLF-iHMM | iHMM |
|---|---|---|---|
| Synthetic | **64.0±0.27** | 66.1±0.35 | 66.3±0.20 |
| Electricity | **274.2±0.5** | 351.1±1.7 | 351.9±1.4 |
| LOB | 95.2±0.6 | **94.3±1.8** | 96.1±1.0 |

*Table E.6.* Average runtime in seconds across 100 runs across experiments (Mean ± stdev). Fastest model is highlighted.

### E.3. Bounded PIFs from batching

Figure E.5 illustrates the empirical PIF under different batch sizes. A batch size of zero recovers the WoLF-iHMM model without state-space robustness. We consider a sandbox setting with a two-state 1D observation model. State 0 corresponds to the observation distribution $\mathcal{N}(\mu_0, \sigma_0)$, and similarly for state 1. Observations are shifted relative to the first state's mean, $\boldsymbol{y} = \mu_0 +$ offset, with offsets ranging from [-1.5, 1.5]. Batches of size $B$ stabilises the PIF by pooling observations with $\mu_0$, yielding bounded behaviour.

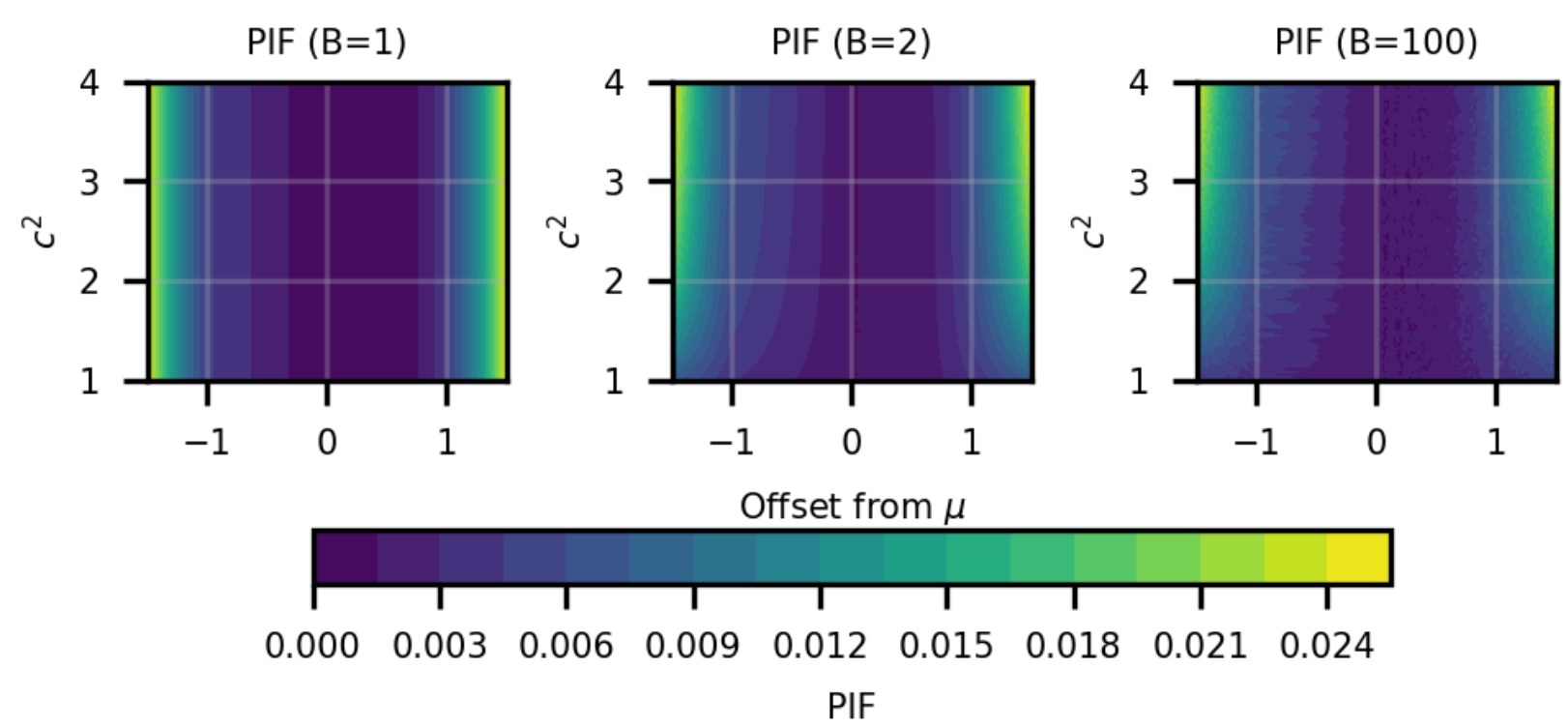

*Figure E.5. State-space PIF.* T he non-batched PIF grows unbounded and is irrespective of the IMQ weights. The batched PIFs remains bounded even under large offsets in contrast. $c^2$ is the IMQ scale.

### E.4. Preference of regimes

We provide supplementary plots in this section to explore the preference of states in the online iHMM. We construct a toy two-state model with known posterior parameters and study the difference in log-marginal likelihood given an outlier observation. We assume state 0 have distribution $\mathcal{N}(0, \sigma_0)$ and state 1 have $\mathcal{N}(5, \sigma_1)$. In both plots of E.6, a lower value indicates reference to state 0. The left hand plot shows that the system struggles to assign a state when model precision is low. In the right hand plot, we consider the scenario where an outlier is coincidentally explained very well by state 1. Consider two possible state trajectories $s_{1:t}^{\text{degen}} = (0, ..., 0)$ and $s_{1:t}^{\text{switch}} = (1, 0, ...0)$. We assume the outlier is the first observation of the batch for convenience, the location of the outlier within the batch is irrelevant. As the outlier $\boldsymbol{y}_1^c$ moves closer to 5 (the mean of the incorrect state 1), the system prefers $s_{1:t}^{\text{switch}}$ which harms interpretability. This justifies our claim that a degenerate HDP prior preserves interpretability.

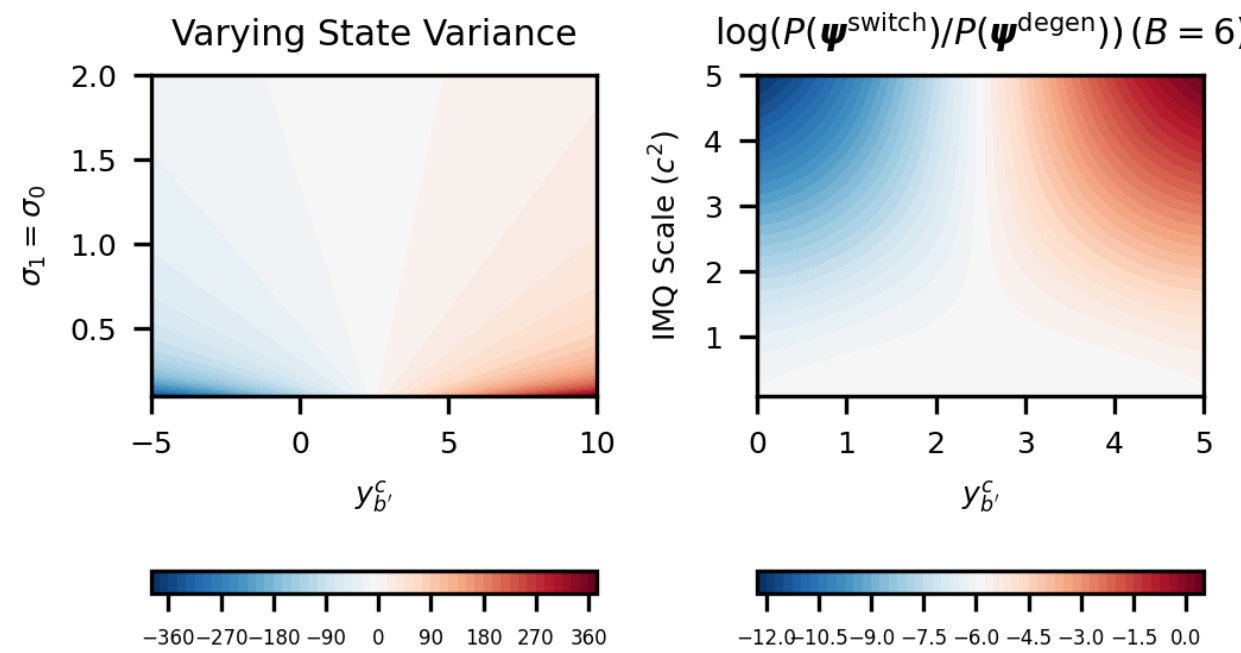

*Figure E.6.* The left hand plot shows a lack of preference if state variance is high. In the right hand plot, the batch size is $B = 6$ and an outlier $\boldsymbol{y}_1^c$ is in the batch.

### E.5. Inference with heavy-tailed posterior predictives

As mentioned in the literature review, a common robustification tactic is to replace the likelihood model with a heavy-tailed distribution to accommodate outliers. This experiment aims to illustrate that a heavy-tailed distribution is not sufficient to achieve bounded PIFs and is non-robust theoretically.

Consider the univariate LG observation model with unknown variance with unknown mean. The setup admits a posterior predictive under the Student-t distribution. For simplicity, assume (1D) observations are generated by

$$y_t \mid \mu_t, \sigma_t \sim \mathcal{N}(\mu, \sigma^2) \tag{129}$$

with Normal-inverse-gamma prior on

$$(\mu, \sigma^2) \sim \mathrm{NIG}(\mu_0, \kappa_0, \alpha_0, \beta_0) \tag{130}$$

where $\mu_0, \kappa_0, \alpha_0, \beta_0$ are initialising hyperparameters. Under this conjugate specification, the posterior remains in the same family and admits closed-form updates. Marginalising out $(\mu, \sigma^2)$ , the resulting posterior predictive distribution for a new observation is a Student-t distribution. In particular, conditioning on data $y_{1:t}$ the posterior predictive takes the form

$$y_{t+1} \mid y_{1:t}, \mu_t, \kappa_t, \alpha_t, \beta_t \sim t_{2\alpha_t}\left(\mu_t, \frac{\beta_t}{\alpha_t}\left[1 + \frac{1}{\kappa_t}\right]\right). \tag{131}$$

The update rules for the sufficient statistics $\mu_t, \kappa_t, \alpha_t, \beta_t$ are given by

$$\kappa_t = \kappa_0 + t\,, \tag{132}$$

$$\mu_t = \frac{\kappa_0\mu_0 + t\bar{y}_t}{\kappa_0 + t}\,, \tag{133}$$

$$\alpha_t = \alpha_0 + \frac{t}{2}\,, \tag{134}$$

$$\beta_t = \beta_0 + \frac{1}{2}\sum_{i=1}^{t}(y_i - \bar{y}_t)^2 + \frac{\kappa_0 t}{2(\kappa_0 + t)}(\bar{y}_t - \mu_0)^2\,. \tag{135}$$

Now consider the difference in log-likelihood between two Student-t distributions $t^{(1)}_{\nu_1}$ and $t^{(2)}_{\nu_2}$ with degrees of freedom $\nu_1$ and $\nu_2$ respectively,

$$\begin{aligned}\Delta(y_t) = \log\Gamma\left(\frac{\nu_1 + 1}{2}\right) - \log\Gamma\left(\frac{\nu_1}{2}\right) - \frac{1}{2}\log(\nu_1\pi\sigma_1^2) - \frac{\nu_1 + 1}{2}\log\left(1 + \frac{(y_t - \mu_1)^2}{\nu_1\sigma_1^2}\right)\\ - \left[\log\Gamma\left(\frac{\nu_2 + 1}{2}\right) - \log\Gamma\left(\frac{\nu_2}{2}\right) - \frac{1}{2}\log(\nu_2\pi\sigma_2^2) - \frac{\nu_2 + 1}{2}\log\left(1 + \frac{(y_t - \mu_2)^2}{\nu_2\sigma_2^2}\right)\right].\end{aligned} \tag{136}$$

When the outlier magnitude tends to infinity, i.e. $|y_t| \to \infty$, then the asymptotic behaviour becomes

$$\lim_{|y_t|\to\infty} \log P(y_t \mid t_\nu(\mu, \sigma^2)) = -(\nu+1)\log|y_t| + C \tag{137}$$

$$\implies \lim_{|y_t|\to\infty} \Delta(y_t) = (\nu_2 - \nu_1)\log|y_t| + C, \tag{138}$$

where $\nu_1 = 2\alpha_1$ and $\nu_2 = 2\alpha_2$. Therefore the log-likelihood difference tends to positive infinity if $\alpha_2 > \alpha_1$. Recall the update rules of $\alpha_t$, $\alpha_2 > \alpha_1$ when distribution 2 is updated more times than distribution 1. Hence, for large outliers, the iHMM system a less updated/newer state with heavy prejudice as the observation's magnitude becomes large.

We illustrate this empirically with a toy two-state example, the new state has a degree of freedom (df) of 5 and the old state has a df of 1000, both are centred with a zero mean. In Figure E.7, as we move the observation away from 0, the iHMM prefers the new state with heavy preference.

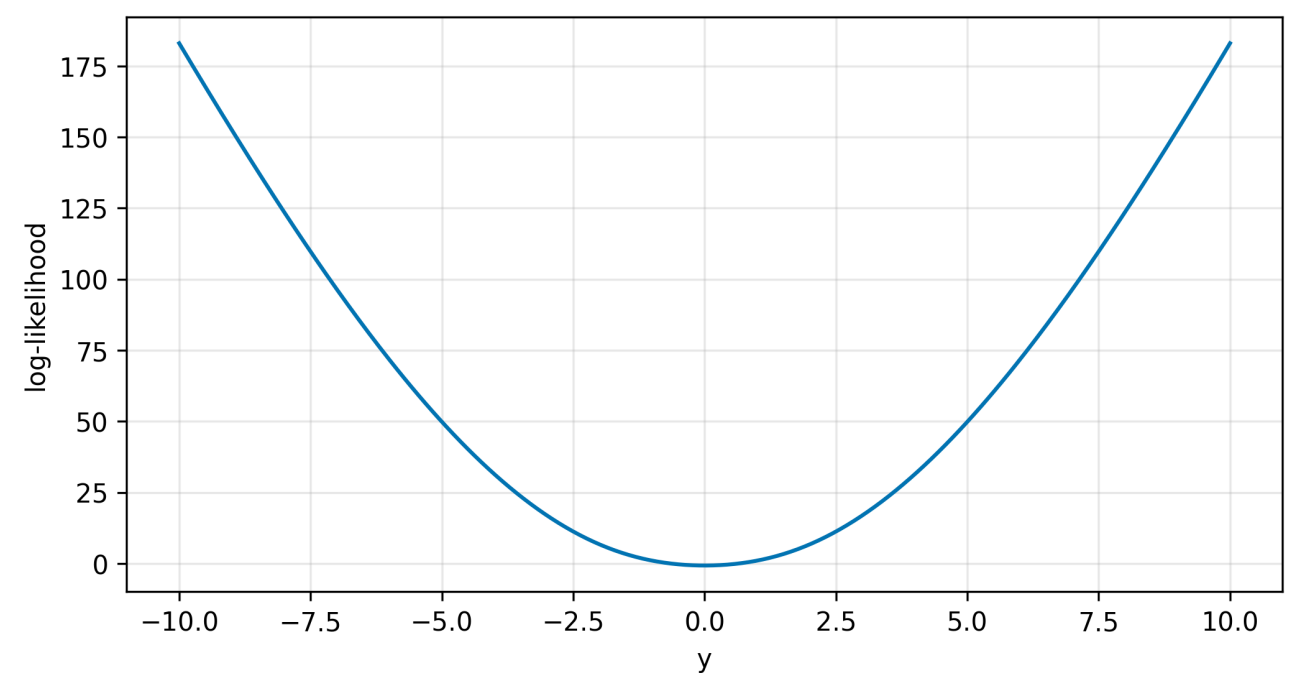

*Figure E.7.* Difference in log-likelihood for the unknown mean unknown variance case.

Finally, to illustrate the state-switching and observation-space non-robustness exists in real-life data, we apply the unknown-variance unknown-mean iHMM on the well-log data (Figure E.8). In particular, the values between 100 and 500 are plotted, there is a window of outliers near the 350-th observations. The `BR-iHMM` remains robust to the outlier in the observation space as well as the state inference path, yet the classical `iHMM` displays noticeable change in its predictions.

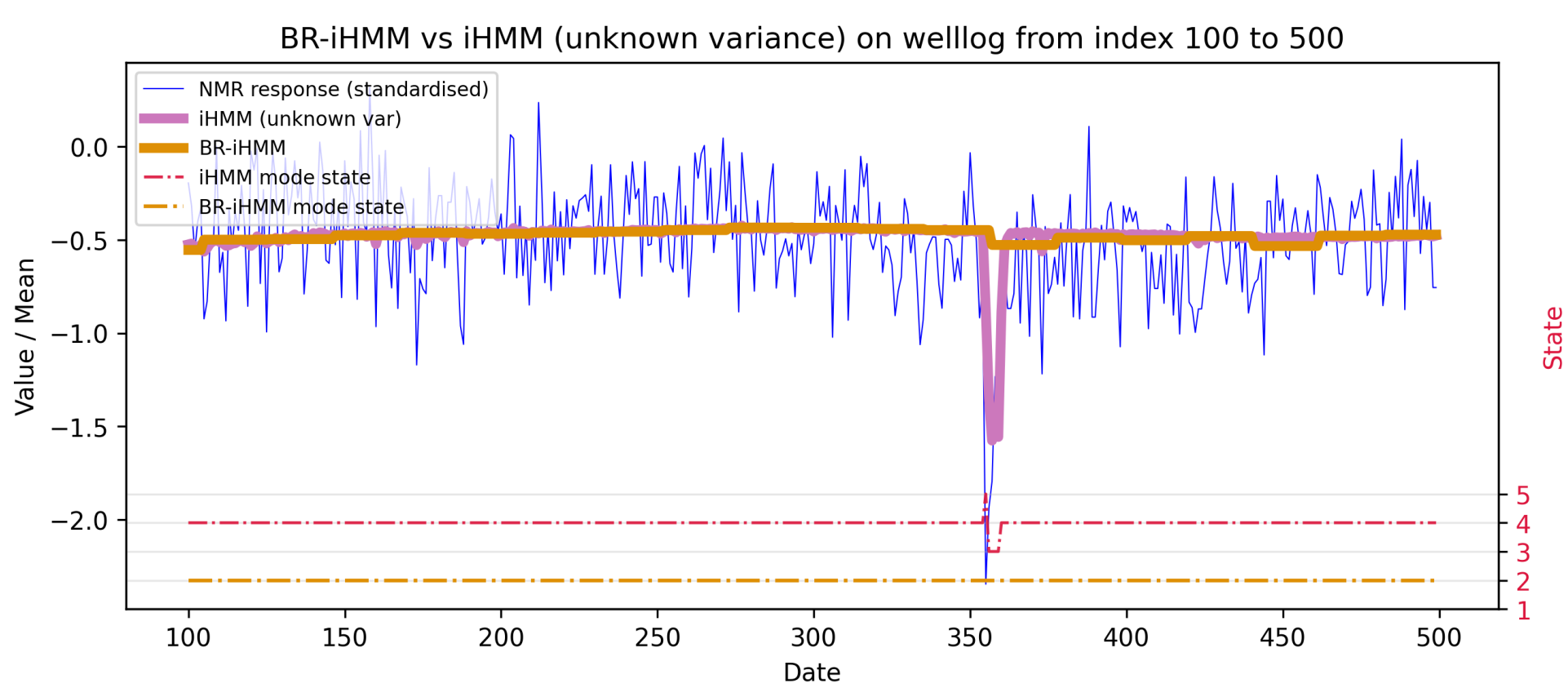

*Figure E.8.* `BR-iHMM` (with known variance) vs classical `iHMM (unknown-var)`. The unknown variance iHMM has a Student-t posterior predictive and is still sensitive to outliers. Student-t remains non-robust and changes states, yet the `BR-iHMM` model is robust.

Hence, using a t-distribution is not a guarantee that the PIF is bounded, in fact, given large enough observations, a Student-t posterior predictive runs into the same problems as our LG example, recovering a similar non-robust observation space result illustrated in Lemmas D.5. We emphasise that our paper focuses on the linear-Gaussian case to illustrate analytical results, but we believe the general intuition of doubly robustness applies to other distributions in the Bayesian paradigm. In that, given there is state-space robustness, the joint PIF explodes for non-robust observation models. Intuitively, as we update

existing observation posteriors over time, the parameters get increasingly confident and fixated in their expectation. Any outliers that are reasonably out of range of the existing states will inevitably be assigned to a new state, as is the intended outcome for classical iHMMs, and the pathology that our `BR-iHMM` is trying to prevent to improve prequential tasks.

### E.6. Summary of fitted hyperparameters

| Dataset | Hyperparameter | `BR-iHMM` | `WoLF-iHMM` | `iHMM` |
|---|---|---|---|---|
| Well-log | $\tau_{\text{ESS}}$ | 10 | 10 | 54 |
| | $c^2$ | 2.124 | 1.657 | – |
| | $B$ | 21 | – | – |
| Synthetic | $\tau_{\text{ESS}}$ | 2 | 1 | 0 |
| | $c^2$ | 1.348 | 3.589 | – |
| | $B$ | 2 | – | – |
| OFI | $\tau_{\text{ESS}}$ | 80 | 75 | 73 |
| | $c^2$ | 4.122 | 4.334 | – |
| | $B$ | 1 | – | – |
| | $\Sigma_0$ | 0.001 | 0.001 | 0.107 |
| | $\rho^*$ | 0.114 | 0.087 | 0.030 |
| Electricity | $\tau_{\text{ESS}}$ | 26 | 50 | 14 |
| | $c^2$ | 1.0 | 3.909 | – |
| | $B$ | 6 | – | – |
| | $\Sigma_0$ | 90.482 | 99.619 | 31.334 |
| | $p$ | 2 | 1 | 1 |

*Table E.7.* Summary of fitted hyperparameters obtained from Bayesian optimisation. Same hyperparameters are used in subsequent runs throughout the same dataset. Extra hyperparameters are included where appropriate. $\Sigma_0$ is the initial variance for newly initiated states, for an $m$-dimensional state vector, the initial covariance matrix is then set to $\Sigma_0 \mathbf{I}_m$ where $\mathbf{I}_m$ is the $m$-dimensional identity matrix. For the Well-log and Synthetic experiment, $\Sigma_0$ is set to $1$. $\rho^*$ is the autoregressive coefficient in the OFI DGP (141). $p$ denotes the degree of feature engineering of the weather covariates, see Section E.11. `MAX_STATES` = 30 for all three online iHMM models.

We specifically highlight that $B$ is data-dependent. For instance, in the Synthetic experiment, $B = 2$ as outliers are isolated incidents, and dampening its effect with one other non-contaminated observation is enough (see Figure E.12). In the well-log dataset, $B = 21$ and is enough to combat the burst-window outliers that are triggered by events that last for a certain period of time (see Figure E.10). In the OFI experiment, $B = 1$, and recovers similar hyperparameters as the `WoLF-iHMM` model as the improvement using batches is not apparent. Our `BR-iHMM` model therefore fallbacks into the `WoLF-iHMM` benchmark. In other words, the `BR-iHMM` model performs at least as good as `WoLF-iHMM` and `iHMM` in cases where batching does not improve the predictive accuracy.

### E.7. Well-log data

A complete plot of predicted values and MAP states are in Figure E.9.

Both `iHMM` and `WoLF-iHMM` frequently created artefact states, `BR-iHMM` was able to distinguish a selection of reusable states but created artefact states at changepoint boundaries, `DSM-BOCD` changepoints are used as a benchmark but suffers from predictive capabilities due to the creation of a new state at every changepoint. The bottom plot reflects the non-reusability problem of `DSM-BOCD` by plotting each changepoint as a new state. Furthermore, we included `iHMM (unknown-var)` to show that even a Student-t posterior predictive is susceptible to the pathology of creating new states for a large enough outlier. As iHMMs do not restart the training process from scratch, all iHMMs produced predictions visibly closer to the well-log samples at those regions.

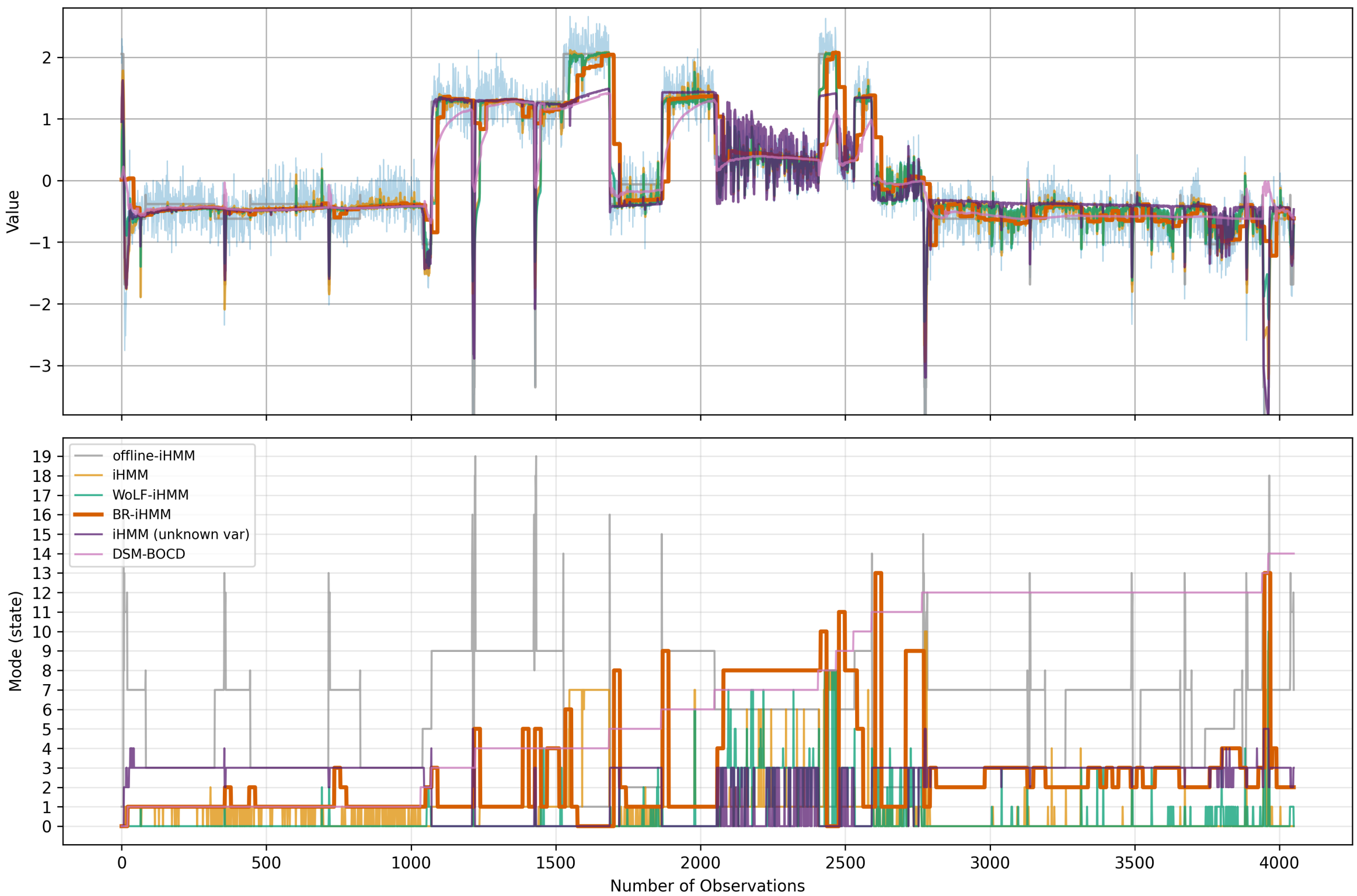

*Figure E.9.* Comparison of iHMM filtered mean and maximum a-posteriori (MAP) state inference with different degrees of robustness on the well-log data. Top subplot overlays the well-log data with the filtered values of `iHMM`, `iHMM (unknown-var)`, `WoLF-iHMM`, `BR-iHMM`, `DSM-BOCD`, and `offline-iHMM`. Bottom subplot shows the MAP state inference.

**Detection delay** We use the `offline-iHMM` MAP states as the benchmark CPs, we remove CPs that last less than 50 data points and treat those as transient outliers. The list of CPs is as follows:

```
    [20   84  360  445  720  825 1071 1223 1433 1527 1718 1867 2049 2409
 2470 2532 2593 2784 3191 3261 3520 3743 3890 3966]
```

The DD across models is in Table E.8 below.

| Model | PPV | TPR | Delay |
|---|---|---|---|
| `BR-iHMM` | $\mathbf{0.4739 \pm 0.0636}$ | $\mathbf{0.5797 \pm 0.0885}$ | $78.06 \pm 46.95$ |
| `WoLF-iHMM` | $0.3384 \pm 0.588$ | $0.3384 \pm 0.0588$ | $63.20 \pm 16.50$ |
| `iHMM` | $0.4086 \pm 0.0635$ | $0.4970 \pm 0.0924$ | $\mathbf{19.19 \pm 11.38}$ |

*Table E.8. Regime classification performances on synthetic data.* PPV, TPR, and delays are averaged over 100 runs for online models. Higher PPV and TPR is better, lower detection delay is better.

For this particular experiment, the true positives (TP), false positive (FP), and false negatives (FN) are matched post-process as state labels are mismatched and differing state labels do not imply an incorrect classification. We produce the TP, FP, and FN labels for the predicted states with Algorithm 9.

**Different batch sizes (Well-log)** We investigate the effect of different batch sizes with respect to the well-log data. Following the optimisation process, we plot the lookahead MAE instead of RMSE for this experiment. The results are presented below in Figure E.10.

As expected, the detection delay generally increases as we increase the batch size. This reaffirms the intuition that CPs have

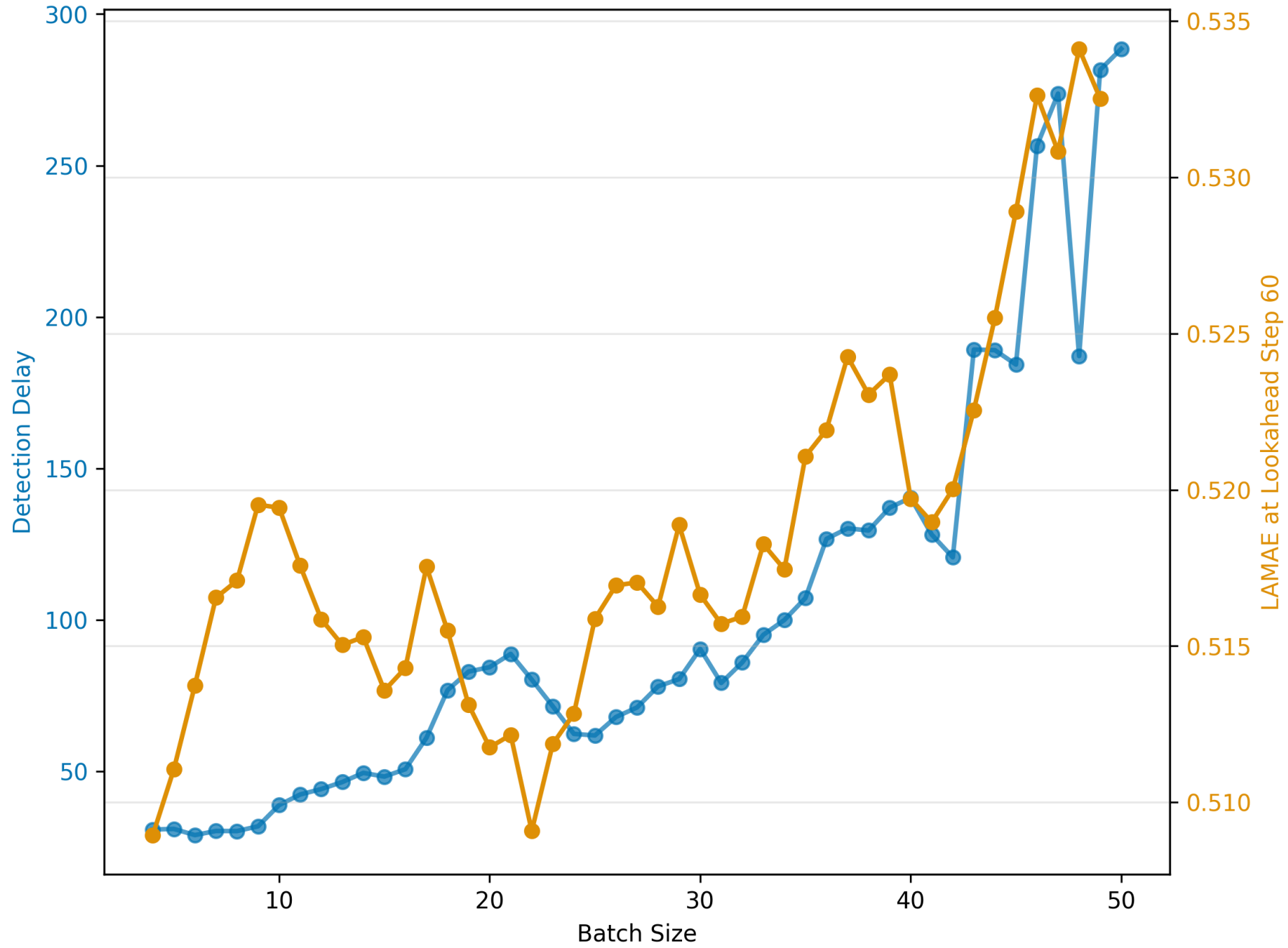

*Figure E.10. Lookahead MAE and detection delay for different batch sizes.* The blue graph, corresponding to the axis on the left, plots the detection delay of `BR-iHMM` as we increase the batch size. The orange graph, corresponding to the axis on the right, plots the lookahead MAE for different batch sizes.

increased chance to be intra-batch as batch size increases and hence requires waiting for the next batch to arrive before recognising a change in regime. For lookahead MAE, we see a dip at around batch size of 20, this concurs with the optimal batch size of 21 that we obtained in via `bayesian-optimization`.

### E.8. Synthetic linear data

Complete plot of Figure 3 in Figure E.11.

**Different batch sizes** We further investigate the effect of batch size on detection delays. Specifically, we run the `BR-iHMM` model with batch sizes ranging from $B \in [1, 100]$ and calculate the detection delay on the synthetic linear data we generated above.

We find that the RMSE is optimal at $B = 2$, which is expected as the outliers are isolated anomalies. For such cases, a batch size of 2 suffices to mitigate its effect on the path posterior as (i) the path posterior penalises the outlier's contribution and revert to the non-contaminated observation in the same batch, and (ii) the batch size of 2 is the minimum cost to propagate the correct prediction to the next batch. Increasing the batch size simply increases detection delay which also increases the RMSE as predictions may originate from the previous regime. Furthermore, the figure confirms the intuition that increasing the batch size can only increase the detection delay as the CP has more chances of being mid-batch than at inter-batch indices.

**Regime detection** Table E.9 shows the results pertaining to changepoint detection, and demonstrates that `BR-iHMM` enjoy more accurate regimes detection at the expense of a small detection delay.

| Model | PPV | TPR | Delay |
|---|---|---|---|
| `BR-iHMM` | $\mathbf{0.963 \pm 0.001}$ | $\mathbf{0.996 \pm 0.001}$ | $3.4 \pm 0.0$ |
| `WoLF-iHMM` | $0.533 \pm 0.002$ | $0.344 \pm 0.002$ | $\mathbf{0.0 \pm 0.0}$ |
| `iHMM` | $0.551 \pm 0.002$ | $0.363 \pm 0.008$ | $0.2 \pm 0.0$ |
| `offline-iHMM` | 0.996 | 0.997 | 0.0 |

*Table E.9.* Values are averaged over 100 runs for online models. Higher PPV and TPR is better, lower DD is better.

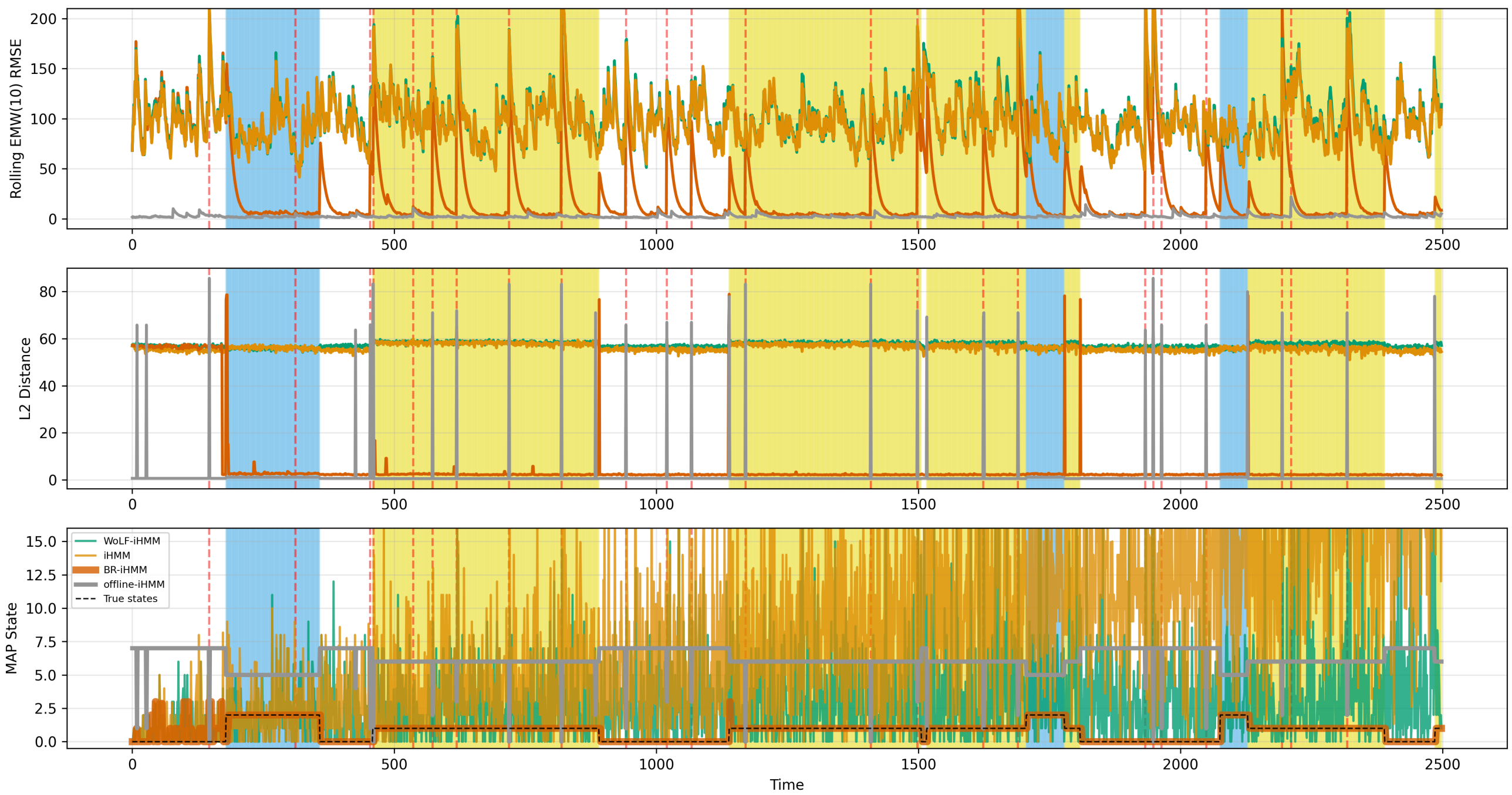

*Figure E.11. Synthetic linear data.* Outlier observations are marked by **red** vertical dashes. **Yellow** and **blue** regions correspond to regimes 2 and 3. First subplot tracks rolling RMSE of mean predictions across the 100 runs. Second subplot compares the L2 distance between average latent state (per model) versus the true data generating state $\boldsymbol{\mu}_t$. The bottom plot shows average MAP state. Each plot overlays the results for all four models. All data points are plotted.

For PPV and TPR, the closer to 1, the better. For delays, the closer to 0, the better.

### E.9. IID synthetic data with burst-like outliers

Consider the following DGP

$$\begin{aligned}
s_t &\in \{1,2,3\}, \quad s_0 = 1 \\
\boldsymbol{\theta}_t &= \boldsymbol{\theta}_{s_t}, \\
y_t &= \boldsymbol{\theta}_{s_t} + e_t, \\
P(s_t \mid s_{t-1}) &= \begin{bmatrix} 0.995 & 0.0025 & 0.0025 \\ 0.0025 & 0.995 & 0.0025 \\ 0.0025 & 0.0025 & 0.995 \end{bmatrix},
\end{aligned} \tag{139}$$

where $e_t \sim \mathcal{N}(0, 0.25)$ for data with Gaussian noise and $2\sqrt{2/3}\, e_t \sim \mathrm{t}(3)$ for data with $t$ noise. We first generate 1180 uncontaminated data points based on this DGP. We then simulate burst-like outliers by first randomly selecting 3 data points to be the start of an outlier window, $c_1, c_2, c_3 \in [1, 1175]$. This experiment was originally constructed to emulate the outliers seen in the well-log data set as a controlled environment of whether the method is effective against multiple outliers in a batch. For each $c_i$, we contaminate the next 5 values with an additive offset, that is, for each $i$

$$\begin{aligned}
y^c_{c_i} &= y_{c_i} + 1 \\
y^c_{c_i+1} &= y_{c_i+1} + 3 \\
y^c_{c_i+2} &= y_{c_i+1} + 5 \\
y^c_{c_i+3} &= y_{c_i+1} + 3 \\
y^c_{c_i+4} &= y_{c_i+1} + 1.
\end{aligned} \tag{140}$$

In our experiments, $c_1 = 134, c_2 = 411, c_3 = 900$.

**Gaussian noise** We compare the state inference process of the online iHMM models `BR-iHMM`, `WoLF-iHMM`, and `iHMM`. `BR-iHMM` is successful in inferring the correct number of states, whereas `iHMM` and `WoLF-iHMM` all created artefact

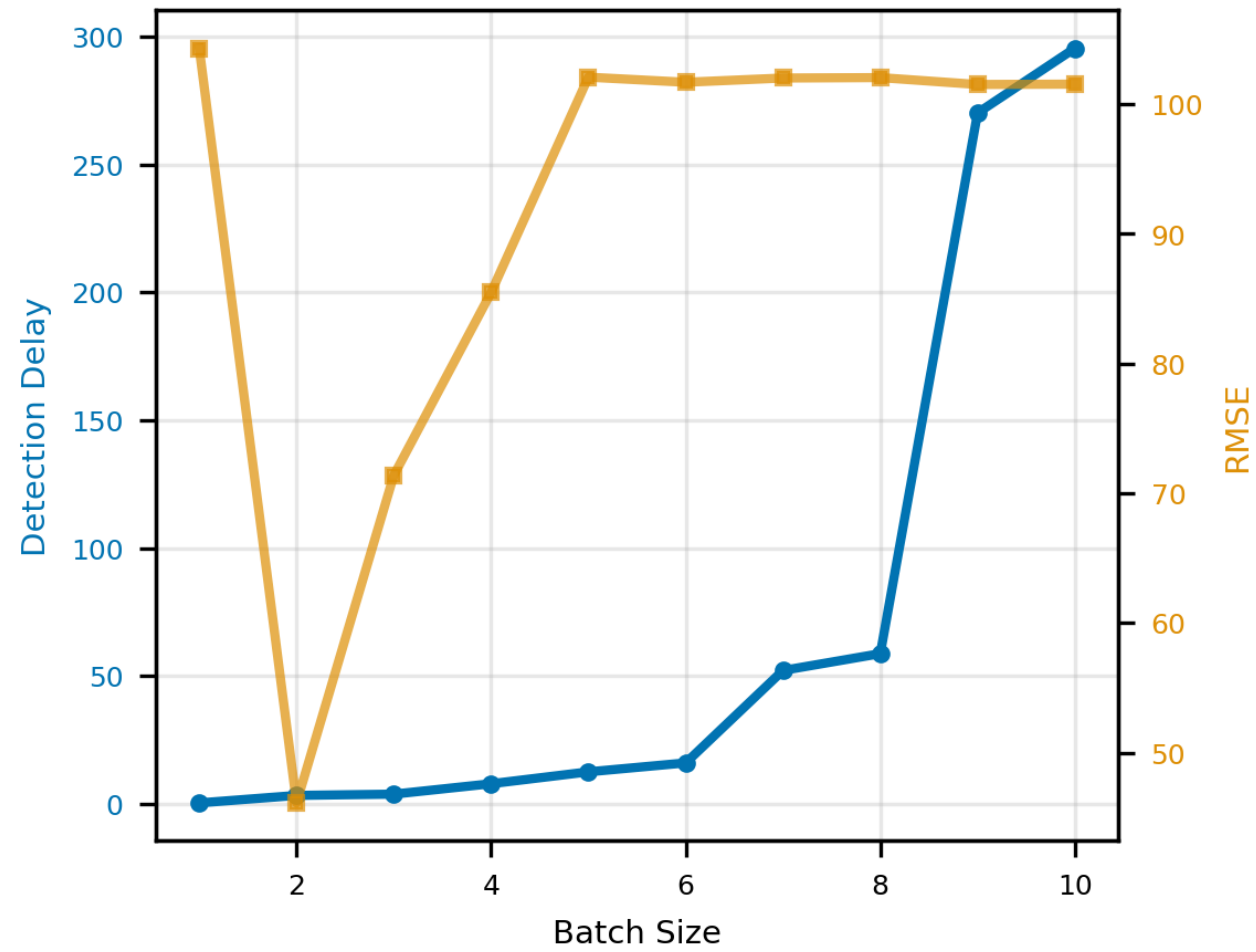

*Figure E.12. RMSE and detection delay for different batch sizes.* The blue graph, corresponding to the axis on the left, plots the detection delay of BR-iHMM as we increase the batch size. The orange graph, corresponding to the axis on the right, plots the RMSE for different batch sizes.

states. The filtered values of BR-iHMM is also resilient against the burst-window of outliers. In particular, both iHMM and WoLF-iHMM exhibited the rapid-switching phenomena in different parts of the time series.

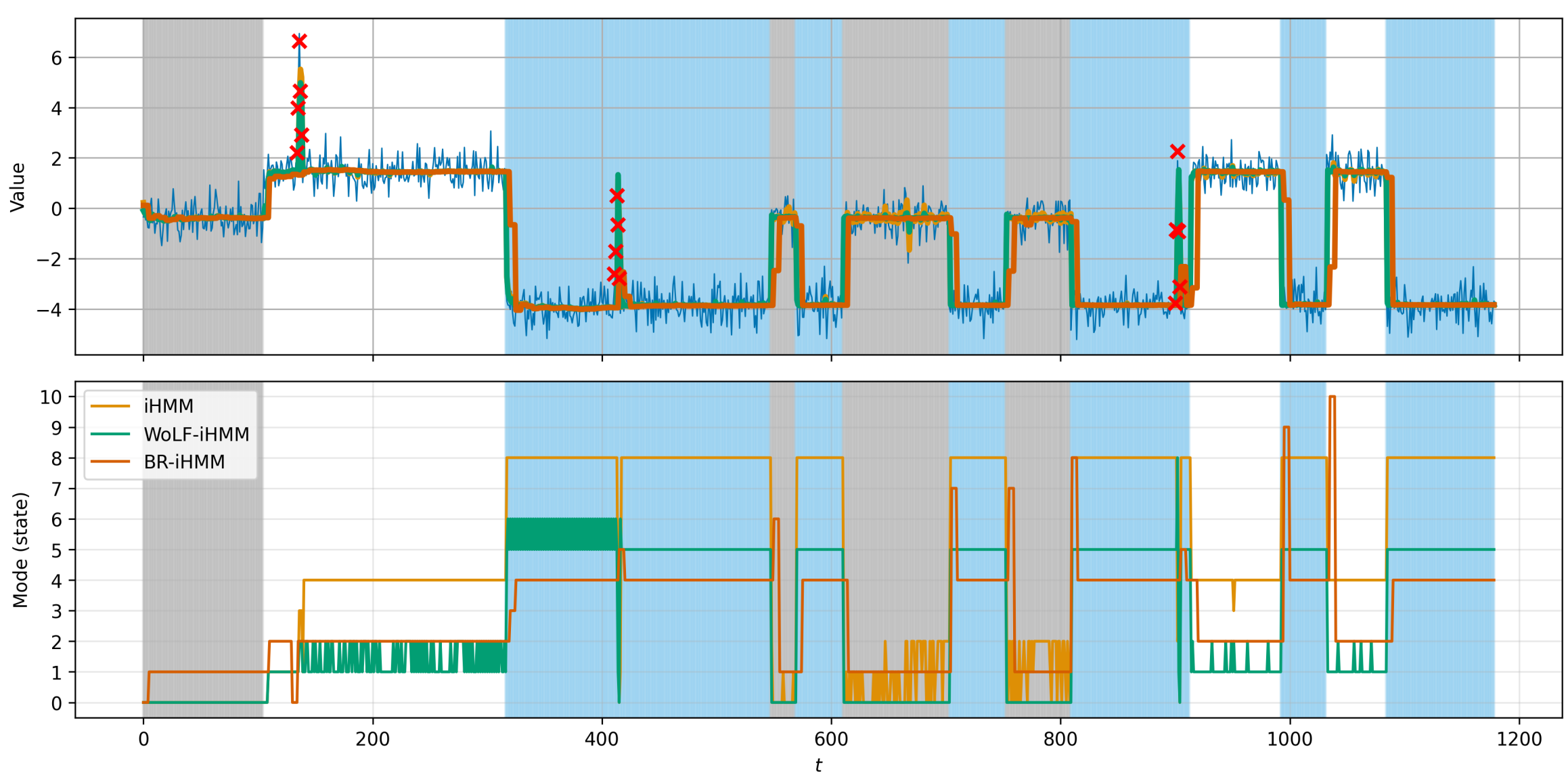

*Figure E.13. IID synthetic data with Gaussian noise.* Outliers are marked with a red cross. Top panel compares predicted value, lower panel compares MAP states.

**T-noise** The rapid-switching problem is exacerbated in the presence of t-noise (see Figure E.14). The MAP state of the BR-iHMM method remains the same trajectory as that inferred in the Gaussian experiment, demonstrating it's resilience against fat tailed noise. On the contrary, WoLF-iHMM was visibly switching to states due to outliers, demonstrating rapid-switching at times, but also creating artefact states to switch to state that happens to be close to the outlier. We deliberately underestimate the LG observation noise $R_t = 0.25 < e_t$ in the model to make the model more misspecified. This exacerbates the rapid-switching pathology in WoLF-iHMM and iHMM.

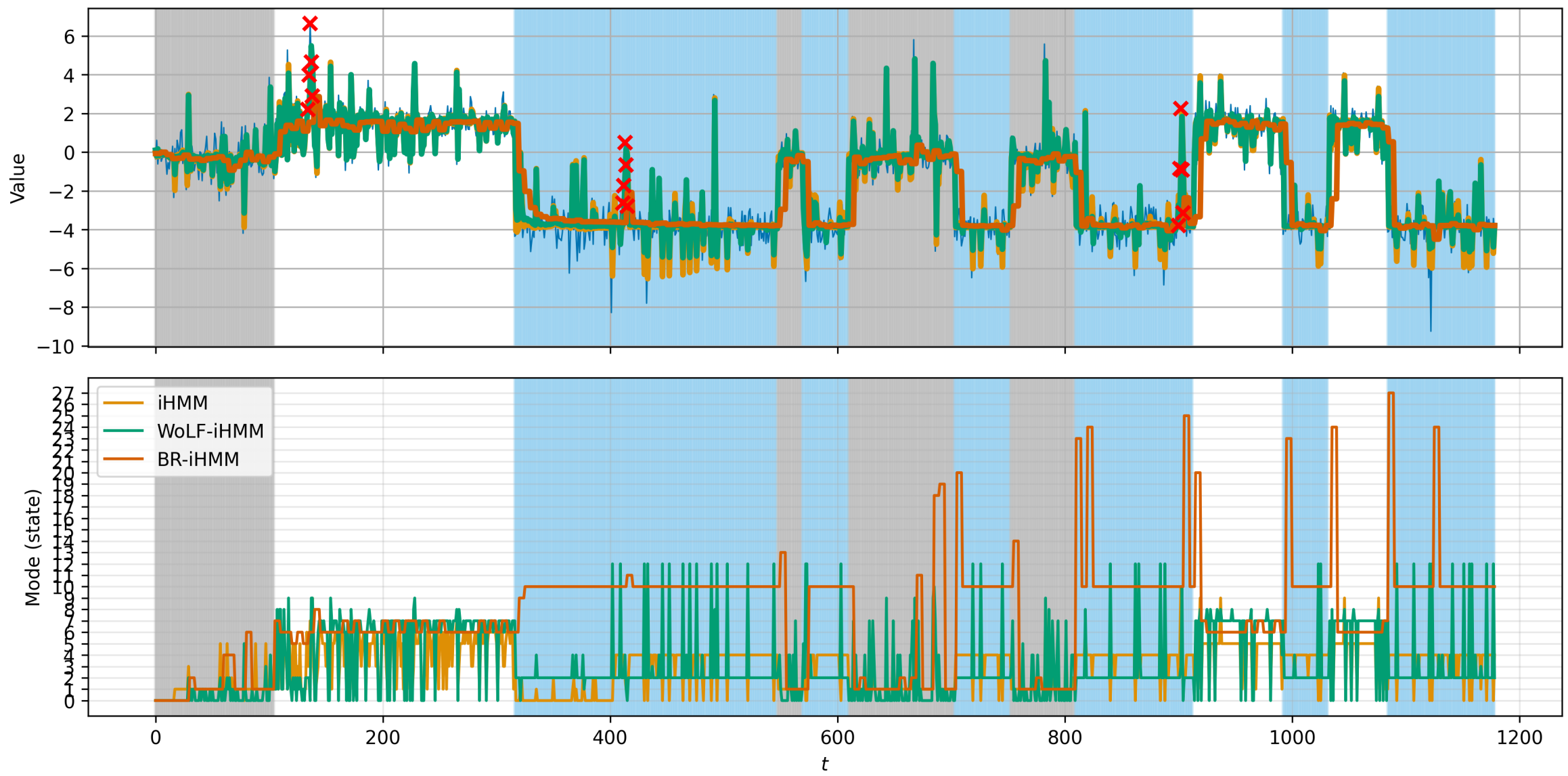

*Figure E.14. IID synthetic data with t-noise.* Outliers are marked with a red cross. Top panel compares predicted value, lower panel compares MAP states.

### E.10. Order flow imbalance

**Calculation of order flow imbalance.** We replicate as simplified version of the experiment in Tsaknaki et al. (2025), where order flow imbalance (OFI) is the sum of signed volumes of trades that occurred within a time interval. Let $V$ denote the total number of trades in a day and fix a bucket size of $z = 400$. Bucketing with respect to $z$, yields a time series of $T = \lfloor V/z \rfloor = 8273$ observations, with each bucket corresponding to approximately one minute on the trade clock. The value $z = 400$ is chosen heuristically for the Microsoft LOB to roughly partition each day's LOB into one minute intervals. Let $v_1, \ldots, v_V$ denote the signed trade volumes throughout the day, a trade is buyer-initiated if $v_i > 0$ and seller-initiated otherwise. The OFI time series is then defined as

$$y_t = \sum_{j=1}^{z} v_{(t-1)z+j}, \qquad t = 1, \ldots, T$$

which is a simple sum of the traded quantity within the interval.

**Autoregressive DGP** We assume a regime-dependent autoregressive DGP,

$$y_t = \rho^* y_{t-1} - (1 - \rho^*)\, \boldsymbol{\theta}^*_{s_t} + e_t, \tag{141}$$

where $e_t \sim \mathcal{N}(0, \sigma^2(1 - \rho^{*2}))$, $\sigma^2$ is known, and $\rho^*$ is a hyperparameter. The observations are standardised, and no explicit outlier removal is performed. The model is parameterised via

$$\boldsymbol{\theta}_t = \begin{bmatrix} \boldsymbol{\theta}_{s_t} & 1 \end{bmatrix}^\top, \quad f(\boldsymbol{x}_t) = \begin{bmatrix} \rho^* - 1 & \rho^* \, y_{t-1} \end{bmatrix}.$$

**`BR-iHMM` falls back to `WoLF-iHMM`** From Table E.5, it is clear that there is a lack of statistical significance in the improvement of `BR-iHMM` over `WoLF-iHMM` and `iHMM`. Indeed, the t-statistic between `BR-iHMM` and `iHMM` is about $t \approx 0.35$, and the t-statistic between `BR-iHMM` and `WoLF-iHMM` is about $t \approx 0.58$, which is far from significant. Recall Table E.7, in which $B = 1$ for `BR-iHMM` in Bayesian optimisation. We reiterate the advantage of `BR-iHMM` as it falls back to `WoLF-iHMM` when the batching does not provide gains in predictive accuracy, in that in the worst case (either from a severely mismatched assumed DGP or extremely rapid-switching data) `BR-iHMM` performs *at least as well* as `WoLF-iHMM` and `iHMM`.

### E.11. Hourly electricity demand

**Feature engineering and model setup** The regression features combine exogenous covariates $\boldsymbol{x}_t$ (temperature, pressure, humidity, wind) with periodic basis functions. For a chosen polynomial degree $p \in \mathbb{N}$, we define the feature map

$$f_p(\boldsymbol{x}_t, t) = \big[1,\ \boldsymbol{x}_t^{(1)}, \ldots, \boldsymbol{x}_t^{(p)},\ \cos(2\pi t/Q),\ \sin(2\pi t/Q)\big], \tag{142}$$

where $\boldsymbol{x}_t^{(k)}$ denotes element-wise powers of $\boldsymbol{x}_t$, and the periodic terms correspond to hourly ($Q = 24$) and semi-annual ($Q = 24 \times 182$) cycles. The resulting model is

$$y_t = f_p(\boldsymbol{x}_t, t)\, \boldsymbol{\theta}_{s_t} + e_t,$$

with fixed observation noise variance $\mathrm{var}(e_t) = R$. The initial parameter variance $P_0$ and the degree $p$ are treated as hyperparameters.

**Extra plots** We present some extra plots in this section.

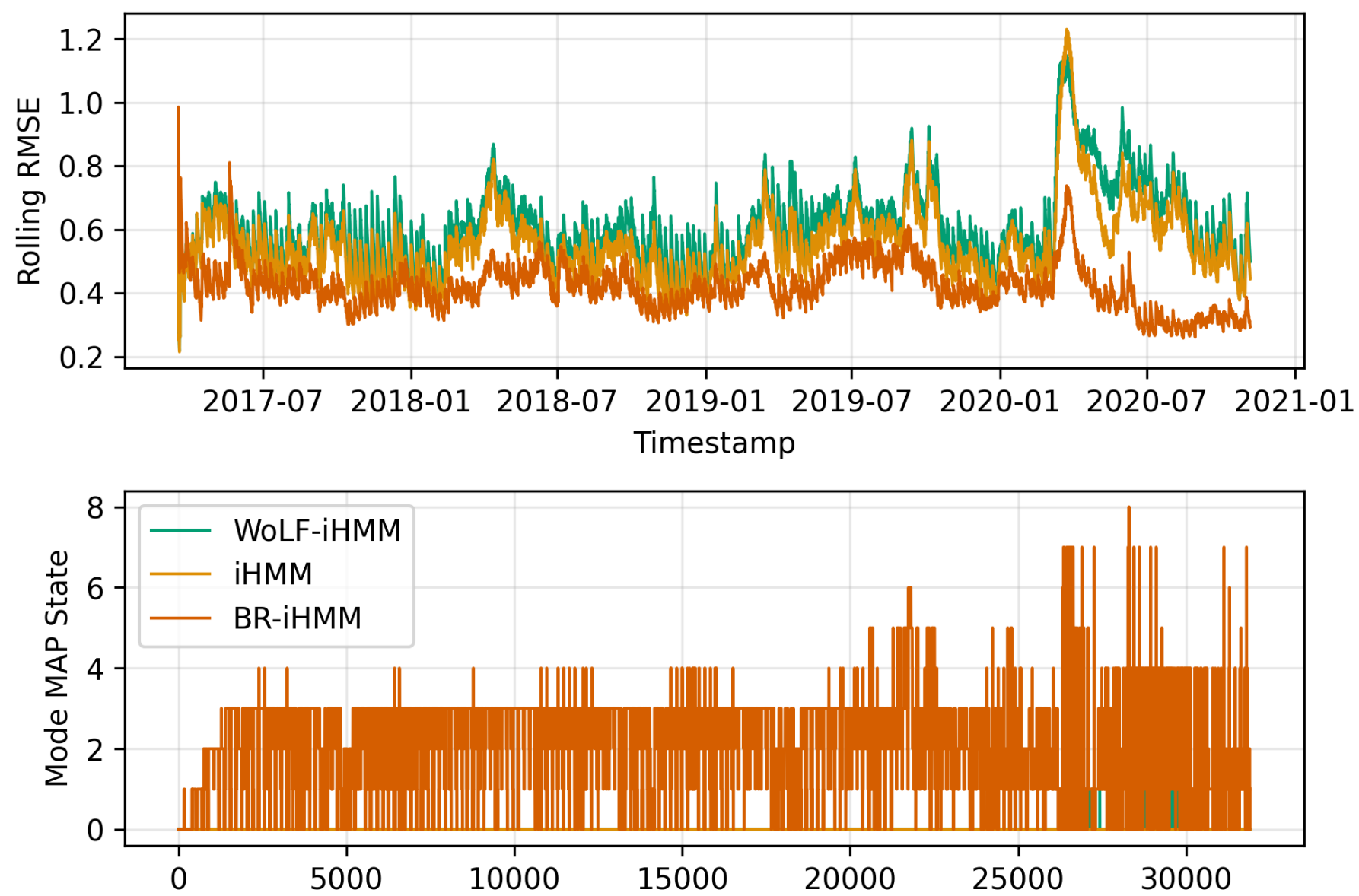

*Figure E.15. Hourly electricity demand.* Top panel plots the rolling RMSE with a window of 100 for each model. Bottom panel plots MAP state. Each plot overlays the results for all three models. The MAP state of `iHMM` did not change, `WoLF-iHMM` detected a small change towards the end.

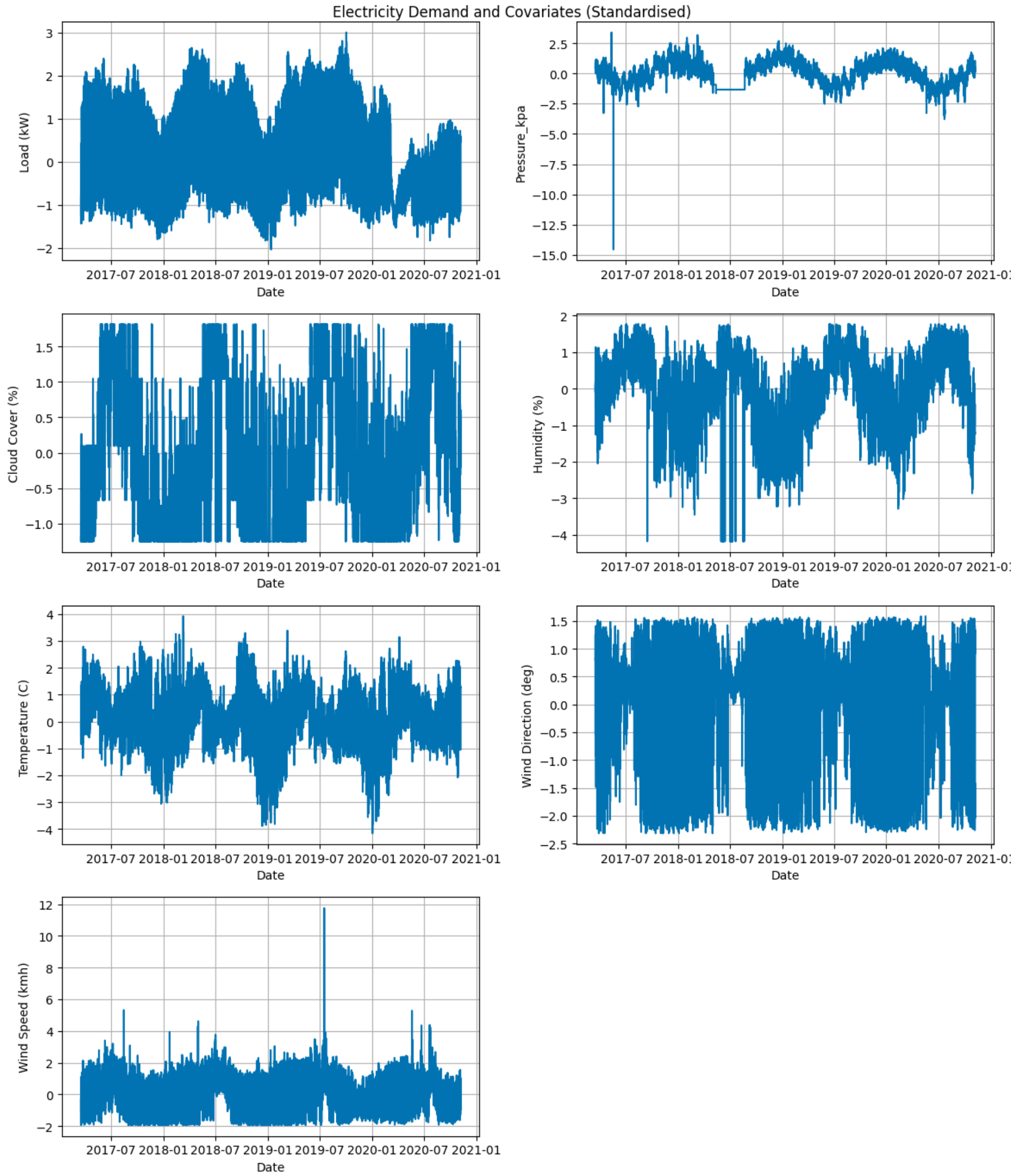

*Figure E.16. Electricity data.* Standardised data for all 31912 data points standardised per dimension. Electricity demand is measured in `Load (kW)`. Data shows clear periodicity.

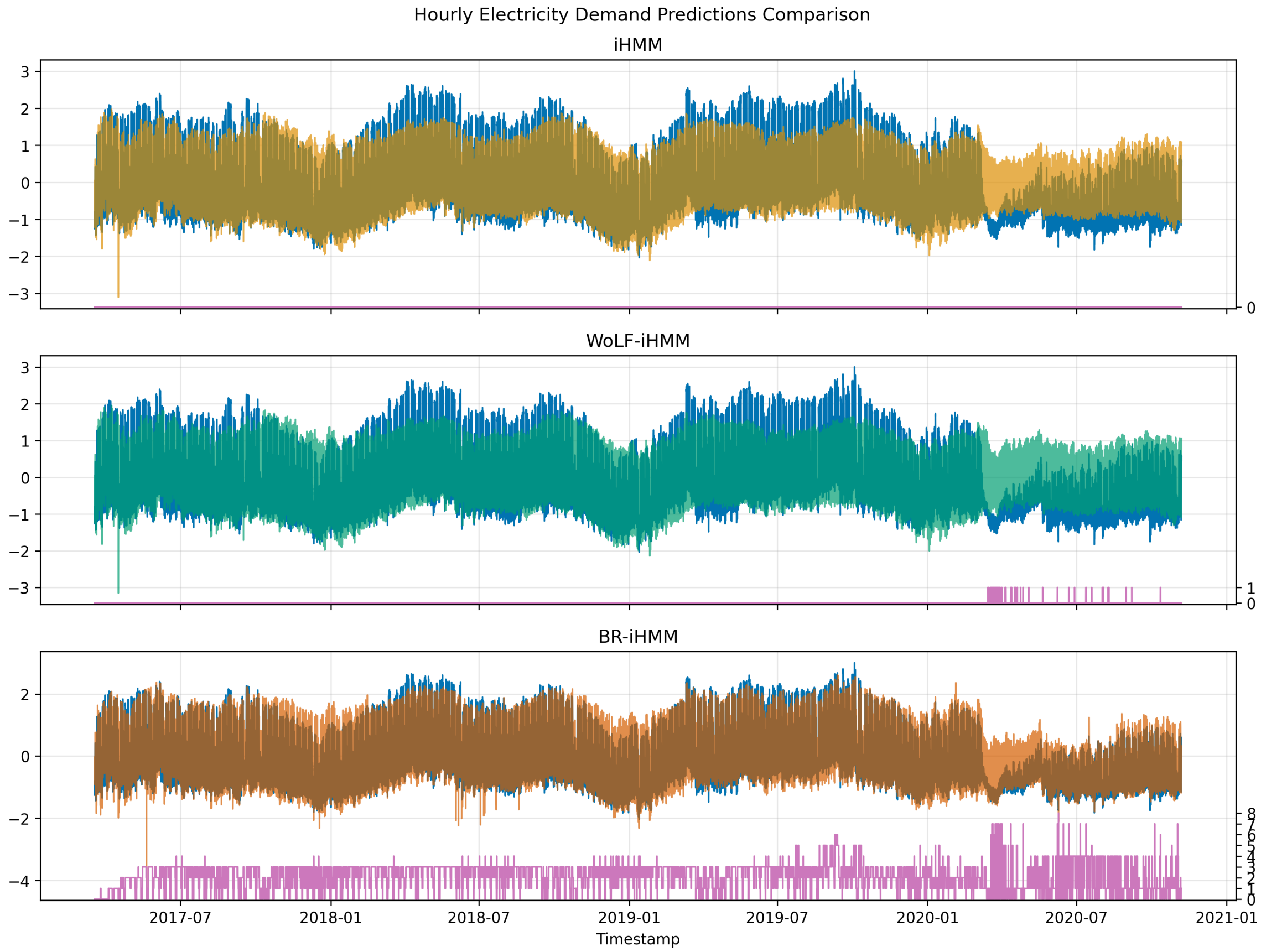

*Figure E.17.* Electricity demand forecasts produced by `iHMM`, `WoLF-iHMM`, and `BR-iHMM` respectively. MAP states are plotted on the right hand axis for each model in **purple**. The true standardised electricity demands are plotted in **blue** in the background.

# F. Limitations and extensions

## F.1. Fixed batch size $B$

Currently, the batch size $B$ is fixed a priori and is dependent on the data. This restricts the state-inference to an arbitrary partition of time.

To alleviate this shortcoming, we explore methods to model $B$ adaptively in the online context. One possible extension is to place a hierarchical Bayesian structure and estimate $B$ as a random variable instead. A natural choice of prior for $B$ would be the Gamma distribution with Poisson likelihood, in which the expected number of occurrences can be estimated for a size of $B$ as well as the gaps between such events (which we know have exponential gaps). Concretely, define

$$\lambda_t \sim \mathrm{Gam}(a_\lambda, b_\lambda) \tag{143}$$

as the latent rate of outlier occurrences. Let

$$B_t \sim \mathrm{Gam}(a_\Delta, b_\Delta) \tag{144}$$

be the size of the batch and

$$\Delta_t \mid B_t, \lambda_t \sim \mathrm{Poi}(B_t\,\lambda_t) \tag{145}$$

as the number of events occurring within the batch of size $B_t$ with Poisson likelihood. See that integrating out $\lambda_t$ yields a negative Binomial distribution on $\Delta_t$. From this formulation, we know event gaps between two consecutive outliers $g_t \in \mathbb{N}$ are exponential

$$g_t \mid \lambda_t \sim \mathrm{Exp}(\lambda_t). \tag{146}$$

We require a method to retrospectively count the number of outliers $\Delta_t$. A naive method is to define a threshold $\tau_w \in (0,1)$, such that the number of outliers is defined as the number of observations within the batch with a weight lower than $\tau_w$, with respect to the assigned state $s'_t$,

$$\Delta_t = \sum_{t'=t+1}^{t+B} \mathbb{1}(w^2_{s_{t'},t'} < \tau_w). \tag{147}$$

This allows us to update the latent rate of outlier rates with

$$\lambda_t \mid \Delta_t, B_t \sim \mathrm{Gam}(a_\lambda + \Delta_t, b_\lambda + B_t), \tag{148}$$

in closed-form, and hence the estimation of $B_t$ sequentially based on the probability $P(\Delta_t = n \mid B_t) < \tau_g$ (i.e. the probability of $n$ outliers in a batch size of $B_t$) versus some user-determined threshold $\tau_g \in (0,1)$. Adaptability and robustness trade-off is therefore encoded probabilistically.

### F.2. Results specific to LG emissions

We acknowledge that our results in Section D are specific to the LG model, however the intuition behind the proofs translate to other emission distributions. We now provide more general claims to the conditions that induce bounded joint PIFs without proofs.

**General claim on bounded state-space PIF** In Lemma D.4, we show that the LG log-likelihood infinitely favours a newer state over an existing one. In Section B.3, we show that the same pathology exists for a univariate Student-t distribution. The sketch of the proof requires us to consider the asymptotic behaviour of the difference in log-likelihoods between a new state and an existing state's posterior predictive as the (contaminated) observation $\|\boldsymbol{y}_t\|_2^2 \to \infty$. Let $g(\boldsymbol{y}_t)$ be the rate of growth of the log-likelihood difference, such that

$$\Delta(\boldsymbol{y}_t) = \log P(\boldsymbol{y}_t \mid \boldsymbol{\Phi}_{t-1}, s_t = \text{new state}) - \log P(\boldsymbol{y}_t \mid \boldsymbol{\Phi}_{t-1}, s_t = \text{existing state}) \in O(g(\boldsymbol{y}_t)). \tag{149}$$

In the LG case, $g(\boldsymbol{y}_t) = \|\boldsymbol{y}_t\|_2$. In the univariate Student-t case, $g(y_t) = \log(|y_t|)$. Therefore, it suffices to bound $\Delta(\boldsymbol{y}_t)$ as $\|\boldsymbol{y}_t\|_2^2 \to \infty$. A sufficient condition to enforce this bound is the WoLF update, consider a weighting function that satisfies the following new assumptions

$$\begin{aligned} \sup_{\boldsymbol{y} \in \mathbb{R}^d} W(\boldsymbol{y}, \hat{\boldsymbol{y}}) &< \infty, \\ \sup_{\boldsymbol{y} \in \mathbb{R}^d} W(\boldsymbol{y}, \hat{\boldsymbol{y}}) g(\boldsymbol{y}) &< \infty. \end{aligned} \tag{150}$$

When this is satisfied, the asymptotic behaviour of the WoLF log-likelihood difference between any two states is finite, which yields non-zero state distributions for all possible state-paths. The state PIF is therefore bounded subsequently. This requirement is also satisfied in the batched posterior predictive (12).

**General claim on bounded observation-space PIF** Recall the observation-space PIF amounts to a bounded KLD between two posterior distributions, one contaminated with an outlier, and the other without. Furthermore, the batched observation-space PIF is bounded as long as the one-step posterior update is bounded by induction (see Lemma D.8). For analytical purposes, we assume it is possible to write the KLD's rate of growth as

$$\mathrm{KL}\big(P(\boldsymbol{\theta}_t \mid \mathcal{D}^c_{1:t}) \mid\mid P(\boldsymbol{\theta}_t \mid \mathcal{D}_{1:t})\big) \in O(h(\boldsymbol{y}^c_t)) \tag{151}$$

for some function $h(\boldsymbol{y}^c_t)$. In the LG case, $h(\boldsymbol{y}^c_t) = g(\boldsymbol{y}^c_t) = \|\boldsymbol{y}^c_t\|_2$. Given the LG update rules, the boundedness is satisfied by the same weighting function $W$, bounding the state-space and observation-space PIF simultaneously. The precise $W$ depends on the choice of posterior and the corresponding update rules. We plan to explore whether it is possible to analytically define the admissible $W$'s for all continuous emission models.

**Emission models with no closed-form updates** As a simple demonstration, we experimented on a regime-switching GARCH model with iHMM via SMC. The stochastic volatility model does not have closed-form updates, so we resort to SMC-type updates. The regime $s_t$ determines which $\mathrm{GARCH}(1,1)$ coefficients are active at time t. Regime 0 of

the DGP corresponds to a low-volatility parameter set, and regime 1 to a high-volatility parameter set. We consider the regime-switching GARCH-model with the model setup:

$$\begin{aligned} s_t &\in 0, 1 \\ y_t &= \sigma_t \varepsilon_t, \\ \varepsilon_t &\sim \mathcal{N}(0, 1), \\ \sigma_t^2 &= \omega_{s_t} + \alpha_{s_t} y_{t-1}^2 + \beta_{s_t} \sigma_{t-1}^2. \end{aligned} \tag{152}$$

Inference is performed using SMC.

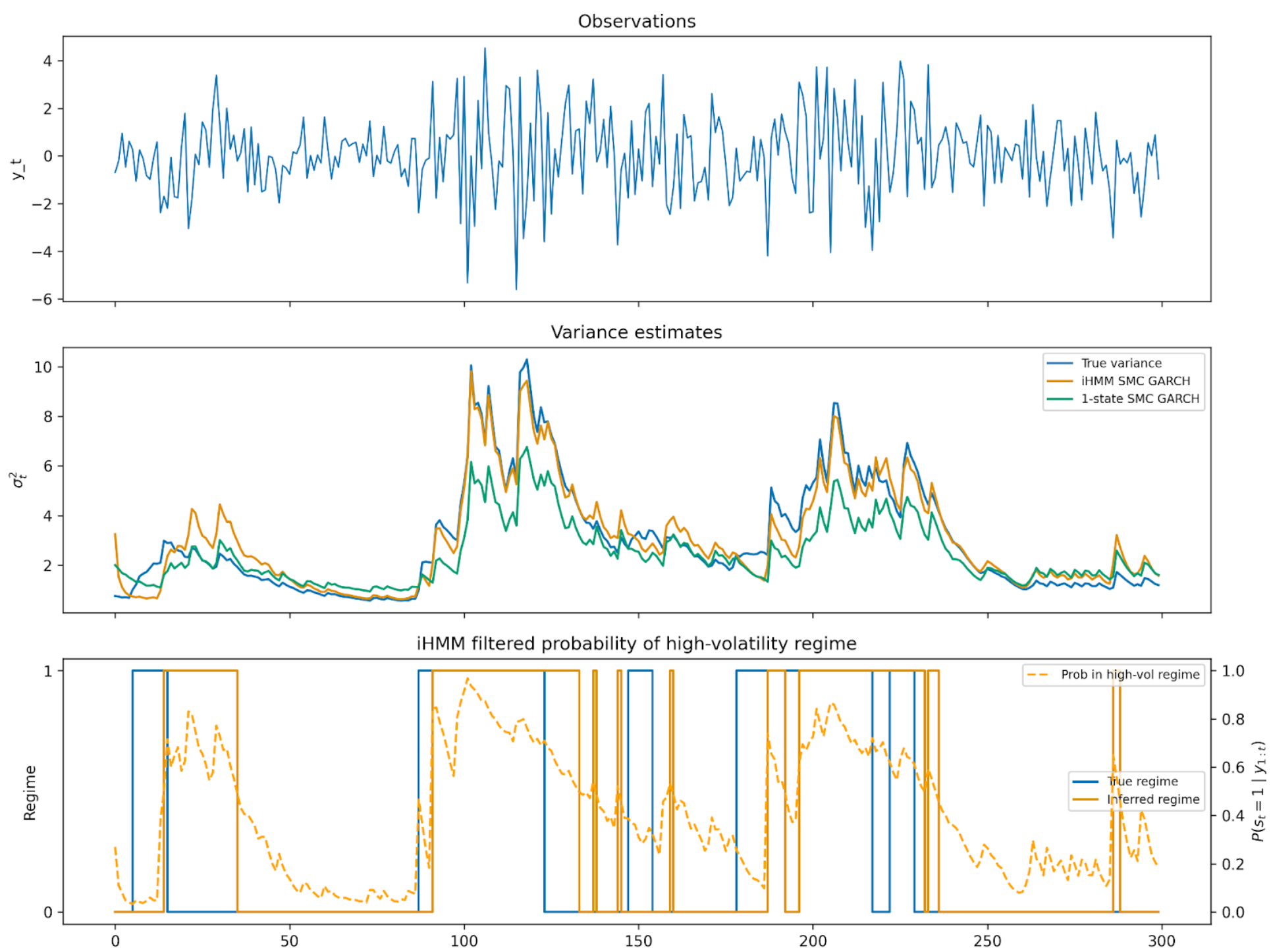

*Figure F.18.* Top panel plots the observations. Middle panel plots the actual variance, along with variance estimates between a naive one-state GARCH and an iHMM of GARCH models. Bottom panel plots inferred regime probabilities for the iHMM GARCH model.

The resulting plots illustrate that periods of elevated variability in the observations coincide with the high-volatility regime, and that the filtered regime probabilities closely track these changes over time. Moreover, the estimated conditional variance adapts dynamically to both recent observations and regime switches, demonstrating that the proposed framework remains applicable in settings where volatility is specified deterministically rather than via latent stochastic processes.

### F.3. Towards linking bounded PIFs to higher predictive accuracy

In this work, robustness is characterised through bounded posterior influence functions (PIFs), which quantify the sensitivity of posterior distributions to arbitrary contamination. While bounded PIF ensures that extreme observations cannot arbitrarily distort the posterior, its relationship to predictive performance is not trivial. Nevertheless, bounded PIF provides a necessary stability condition for reliable prediction: without it, a single outlier can arbitrarily corrupt the posterior and degrade future predictions. Empirically, we observe that enforcing bounded PIF leads to improved predictive performance in settings with outliers or model misspecification (Section E), suggesting that stability of the posterior is a key driver of prequential accuracy in these regimes. A formal characterisation of the relationship between bounded PIF and predictive accuracy remains an open problem and is an interesting direction for future work. Concretely, one such results requires us to relate

$$\mathcal{R}_{t+1} = \mathbb{E}[(\boldsymbol{y}_{t+1} - \hat{\boldsymbol{y}}_{t+1})^2]$$

which denotes the predictive error. Similarly, let $\mathcal{R}^c_{t+1}$ denote the corresponding error under contaminated data. A natural target is to establish a bound of the form

$$|\mathcal{R}^c_{t+1} - \mathcal{R}_{t+1}| \leq C\sqrt{\mathrm{KL}(P(\boldsymbol{\theta} \mid \mathcal{D}_{1:t}) \,\|\, P(\boldsymbol{\theta} \mid \mathcal{D}^c_{1:t}))} = C\sqrt{\mathrm{PIF}(\boldsymbol{y}^c_t)},$$

for some constant $C > 0$. Extending this argument to the iHMM setting requires controlling both state and parameter perturbations through the joint PIF decomposition, which introduces additional challenges due to the mixture structure of the predictive distribution. This suggests a concrete direction for future work: establishing predictive-error stability bounds of the form $\Delta\mathrm{RMSE} \leq C\sqrt{\mathrm{PIF}}$ for online HSSMs.

### F.4. Long-term sensor failures

When consecutive outliers exceed $B$, the guarantee no longer applies directly as multiple batches are affected. This arises in scenarios when there are long-term sensor failures. Particularly, assuming the observations produced by the sensor failures follow the same distribution, then the model typically allocates such observations to separate states rather than distorting existing ones; persistent outliers are grouped into consistent states across batches. This contains their effect at the level of state allocation rather than parameter drift. From a practical perspective, this behaviour can be desirable. Persistent sensor failures or systematic corruptions are treated as distinct regimes, which can improve interpretability. In addition, rarely visited states can be pruned over time, limiting long-term impact.