# OpenReview forum: "Doubly Outlier-Robust Online Infinite Hidden Markov Model"
_ICML.cc/2026/Conference — ICML 2026 regular_

### Official Review · Reviewer_bFCF · 2026-03-06

**Soundness:** 4
**Presentation:** 4
**Significance:** 3
**Originality:** 4
**Overall Recommendation:** 6
**Confidence:** 5

**Summary:**

The paper introduces a doubly-robust update rule for the online infinite hidden Markov model. The proposal is based on the observation that the posterior influence function (PIF) is unbounded even when observation updates are robust. Combining WoLF-based observation updates with batched state inference yields a bounded PIF. The authors also provided some applications.

**Compliance With Llm Reviewing Policy:**

Affirmed.

**Key Questions For Authors:**

1) The manuscript contains several minor typographical and punctuation errors like misplaced commas and periods. A careful proofreading would improve readability.
2) The intra-batch is a key concept in this proposal. Could you clarify how you choose the intra-batch in both theory and practice?
3) The inverse-multiquadratic weights are used for the robust estimation. Are there any other weighting methods available?

**Limitations:**

Yes.

**Strengths And Weaknesses:**

Soundness:
The manuscript clearly introduces the relevant theoretical concepts, and formal proofs support the associated results. The logical development is coherent, and the arguments appear internally consistent.

Presentation:
The logical flow of the manuscript is clear, and the motivation behind the contributions is well articulated. The authors clearly explain what is being proposed and why it is necessary.
In addition, the inclusion of empirical data analysis strengthens the practical relevance.

Significance:
The lack of robustness in this model class has been well recognized in the literature. By introducing a clear theoretical framework, the authors provide a principled way to address this limitation.
This contribution substantially improves the understanding and treatment of the robustness issue, making the work highly significant.

Originality:
As discussed above, the manuscript proposes a novel method grounded in a well-formulated theoretical framework. The approach is not merely an incremental modification, but is motivated and supported by new theoretical insights.
In addition, its practical applicability is empirically demonstrated, which further strengthens the originality and overall contribution.

---

> ### Author Rebuttal · Authors · 2026-03-31
>
> We thank the reviewer for their kind comments. We respond to your questions below.
>
> **R4.1 Typographical errors**
>
> We have proofread the manuscript and corrected typographical and punctuation issues.
>
> **R4.2 Intra-batch**
>
> Thank you for this question. In the revised version, we clarify how the intra-batch size B is chosen. In practice, B is treated as a hyperparameter and selected via validation or Bayesian optimisation, as in our experiments.
>
> We also expand the discussion to outline directions for adapting B online. In particular, one possible approach is to model B as a random variable and infer it during training.
>
> **R4.3 Other weighting functions**
>
> In theory, any weighting function satisfying the boundedness conditions in Eqn (9) can be used. In practice, this includes the IMQ, MD, and TMD weighting functions introduced in [DuranMartin2024], as well as weighting schemes derived from [West1981], where the score can be expressed as a scalar reweighting of the Gaussian score.
>
> We clarify this flexibility and provide additional discussion in the revised version.
>
> **References**
>
> [DuranMartin2024]: Duran-Martin, G., Altamirano, M., Shestopaloff, A., Sanchez-Betancourt, L., Knoblauch, J., Jones, M., Briol, F.-X., and Murphy, K. P. Outlier-robust kalman filtering through generalised Bayes. In Salakhutdinov, R., Kolter, Z., Heller, K., Weller, A., Oliver, N., Scarlett, J., and Berkenkamp, F. (eds.), Proceedings of the 41st International Conference on Machine Learning, volume 235 of Proceedings of Machine Learning Research, pp. 12138–12171. PMLR, 21–27 Jul 2024. URL https://proceedings.mlr.press/v235/duran-martin24a.html.
>
> [West1981]: West, Mike. "Robust sequential approximate Bayesian estimation." Journal of the Royal Statistical Society Series B: Statistical Methodology 43.2 (1981): 157-166.

---

> > ### Author Rebuttal · Reviewer_bFCF · 2026-03-31
> >
> > Thank you for the rebuttal. I am satisfied with the clarifications provided by the authors.

---

### Official Review · Reviewer_bNnS · 2026-03-13

**Soundness:** 3
**Presentation:** 3
**Significance:** 2
**Originality:** 2
**Overall Recommendation:** 5
**Confidence:** 4

**Summary:**

This paper shows a robust method with both theoretical guarantees and practical performance which can solve the weak robustness of the online infinite Hidden Markov Model (iHMM) in streaming data with outliers and model misspecification, First, this paper defines the robustness of the online iHMM based on the Posterior Influence Function (PIF). Then, it gives a key conclusion: the robustness of the online iHMM is essentially a two-fold problem, which requires simultaneous constraints on both the observation parameter posterior and the latent state posterior. Constructing robustness only in the observation space cannot achieve overall robustness of the model. Based on this conclusion, the authors propose a doubly-robust online iHMM method called the Batch Robust infinite Hidden Markov Model (BR-iHMM). This method uses the Weighted observation Likelihood Filter (WoLF) to achieve robust updates in the observation space, and introducing a batched state inference mechanism that enforces intra-batch state persistence via a degenerate sticky HDP prior. Through rigorous theoretical derivation, the authors show that when the WoLF weighting function satisfies certain boundedness conditions, BR-iHMM guarantees bounded joint posterior PIF, which provides provable theoretical guarantees for the model’s performance under outlier contamination. The authors conduct systematic experiments, the results show that, the batched mechanism also yields computational speed-ups by reducing state sampling from every time step to once per batch, thereby avoiding the exponential path enumeration that would otherwise arise within each batch (Proposition C.1).

**Compliance With Llm Reviewing Policy:**

Affirmed.

**Final Justification:**

The rebuttal addressed my main concerns.

**Key Questions For Authors:**

1. In the inductive step of Lemma D.4, the authors claim that $\Sigma^c_{(n+1)} = \Sigma_{(n+1)}$, based on the assertion that "covariance updates in linear-Gaussian systems are independent of observations". However, the innovation covariance $S_{s_t}$ in the WoLF update formula (Equation (11)) includes the term $R_t / w^2_{s_t, t|t-1}$, and the weight $w^2$ explicitly depends on the observation $y_t$. This means that under the WoLF framework, covariance updates depend on observations through the chain $w^2 \to S \to K \to \Sigma_{\text{post}}$. If $\Sigma^c \neq \Sigma$, how should the trace and log-det terms that were eliminated in the proof of Lemma D.4 be handled?
2. The results for the OFI dataset in Table 1 show that the RMSE differences among the three online iHMM variants are minimal (BR-iHMM: $0.616\pm0.082$, WoLF-iHMM: $0.623\pm0.089$, iHMM: $0.620\pm0.080$). A two-sample t-test ($n=100$) based on the reported means and standard deviations shows that the difference between BR-iHMM and iHMM ($t \approx 0.35$) and the difference between BR-iHMM and WoLF-iHMM ($t \approx 0.58$) are far from reaching statistical significance. Did the authors conduct statistical significance tests on the experimental results? In the case of insignificant differences, should the labeling method and related discussions be adjusted?
3. In Table 1, the synthetic experiment reports standard deviations of exactly 0 for all three online iHMM variants (e.g., BR-iHMM: $46.1 \pm 0$) over 100 runs. Given that particle learning involves stochastic resampling and state sampling, exactly zero variance is surprising. Is this a rounding artifact? Clarifying this would strengthen the credibility of the reported results.
4. The paper defines order-$B$ batch robustness and proves the corresponding boundedness of PIF. When the duration of consecutive outliers exceeds $B$ (e.g., long-term sensor failure), abnormal observations will influence multiple batches. In this case, could the cumulative effect cause the state posterior to deviate? Do the authors have theoretical or empirical analyses of this scenario?

**Limitations:**

The limitations discussion in the conclusion is brief and could benefit from a more explicit analysis of the trade-off between robustness and adaptivity governed by the fixed batch size $B$.

**Strengths And Weaknesses:**

# Strengths
1. The paper accurately identifies a key under-explored point in existing robust online iHMM research: even after completing robust updates in the observation space, outliers can still trigger spurious state generation, resulting in unbounded PIF in the state space.
2. Based on generalized Bayesian inference, the authors evaluate robustness using PIF and prove the boundedness of the joint posterior PIF of BR-iHMM in Theorem 4.2. The theoretical derivation has clear and reasonable core assumptions and largely self-consistent core mathematical logic, though one step in Lemma D.4 requires correction (see Weaknesses).
3. The experimental design is comprehensive, covering synthetic data with controlled outliers, real-world time series prediction tasks, and state segmentation tasks. The multi-dimensional evaluations are conducted from three aspects, providing preliminary empirical support for the effectiveness of the proposed BR-iHMM method.
4. The paper has a clear structure and complete logic, following a full narrative from problem motivation, related work, background, model design, theoretical proof to experimental validation. The overall presentation is smooth and easy to follow.
5. The authors use intuitive figures to show the key advantages of BR-iHMM, including the boundedness of PIF,  RMSE comparisons, and MAP state inference results, clearly showing the model’s robustness to outliers.
6. The related work section systematically reviews existing research on robustness in the observation space and state space for state-space models, and clearly claims the contribution of this paper relative to previous work.
7. The proposed method has important practical application value for robust online time-series prediction and online state learning, since parameter corruption and spurious state generation caused by outliers widely exist in finance, energy, geology, and other fields.
8. The doubly-robust framework proposed in this paper is not limited to linear-Gaussian iHMM, but also provides a reusable framework for the robustness research of other online Bayesian nonparametric state-space models, and is expected to inspire future work in the field of robust sequential learning.
9. This paper reveals the inherent trade-off between robustness and adaptivity in online state-switching models, provides a new perspective for the analysis of the robustness–adaptivity trade-off in online learning.
10. This paper is the first work to simultaneously achieve provable robustness in both the observation space and latent state space for online iHMM, filling the gap in existing research that only optimizes robustness for a single space.
11. The authors combine the degenerate sticky HDP prior with batch inference, which not only ensures the boundedness of the state-space PIF but also brings significant improvements in computational efficiency.

# Weaknesses
1. There is one step in the proof of Lemma D.4 which claims that "covariance matrices are independent of observations". This seems invalid under the WoLF framework. When observations are outliers, the posterior covariance $\Sigma^c$ shrinks from the prior, while it shrinks normally for normal observations, leading to $\Sigma^c_{(n)} \neq \Sigma_{(n)}$.
2. There is a gap in the experimental comparisons in related work: although the paper reviews the literature of existing robust iHMM methods in the main text and Appendix B, it does not provide direct comparisons with these methods in the experiments.
3. The discussion of limitations is not sufficiently developed.
4. The practical performance improvement of the method varies greatly across datasets: on synthetic data, BR-iHMM significantly reduces RMSE compared with baselines (up to 67\%), but the improvement on real-world datasets is relatively limited. In particular, on the OFI financial dataset, BR-iHMM performs almost the same as WoLF-iHMM (RMSE reduction is about 1.1\%), and Bayesian optimization selects $B=1$, which means that the batching component does not provide additional benefits in this scenario. These results show that the practical gain of the doubly-robust mechanism may highly depend on data characteristics.
5. The individual technical components of this paper draw heavily on existing work: WoLF (Duran-Martin et al., 2024) for observation-space robustness, the generalized Bayesian online learning framework (Duran-Martin et al., 2025) for batch inference, the sticky HDP prior (Fox et al., 2008b), and the particle learning paradigm (Rodriguez, 2011). The original contributions lie in the principled combination of these components (particularly the degenerate scheduling mechanism $\kappa_t = 0, \infty$).
6. For the core issue of the trade-off between robustness and adaptivity, the paper adjusts the balance between them using a fixed batch size $B$ as a hyperparameter, and relying on Bayesian optimization to select $B$, but does not explore an online adaptive mechanism for adjusting the batch size $B$. The authors list this as one of the future work directions in the conclusion, but considering that this is a key requirement for the practical deployment of the method, this limitation needs more sufficient discussion.

---

> ### Author Rebuttal · Authors · 2026-03-31
>
> We thank the reviewer for the thorough review.
>
> **R3.1 There is one step in the proof [...] seems invalid under the WoLF**
>
> Thank you for pointing this out. You are correct that under WoLF, the covariance depends on observations via the weight, so the original argument required correction.
>
> In the revision, we fix this and show that both $Tr(\Sigma_{n+1}^{-1}\Sigma_{n+1}^c)$ and $\log\frac{\det \Sigma_{n+1}}{\det \Sigma_{n+1}^c}$ remain bounded.
>
> Using the precision-form update $(\Sigma_{n+1})^{-1} = \Sigma_n^{-1} + w_{n+1}^2 H_t^\top R_t^{-1} H_t$, the trace term is bounded via $\Sigma_{n+1}^c \preceq \Sigma_n^c$ and the inductive hypothesis. The log-determinant term is bounded using $\log\det(A+B)\le \log\det(A)+Tr(A^{-1}B)$ with the same decomposition.
>
> We provide details in the revised text.
>
> **R3.2 Not direct comparison in the Appendix**
>
> Thank you for the suggestion. We add direct comparisons to existing robust iHMM methods in the appendix (Appendix E), evaluated under the same settings as BR-iHMM. We include pointers to these results in the revision.
>
> **R3.3 Limitations not sufficiently developed**
>
> Thank you for this suggestion. We consider the following limitations of our work
>
> (1) The batch size B is fixed a priori and dataset-dependent, motivating adaptive selection.
>
> (2) Theory relies on LG emissions for closed-form PIF, and a general treatment remains open.
>
> (3) The link between bounded PIF and predictive performance is not yet characterised.
>
> (4) We do not provide a formal analysis of the adaptivity–robustness trade-off in  B, relying instead on empirical evidence (e.g., Fig. E.8, E.10).
>
> We make these limitations more explicit in the revised manuscript.
>
> **R3.4 Dataset-dependent gains.**
>
> We acknowledge that empirical gains vary across datasets, reflecting the trade-off between robustness and adaptivity.
>
> The OFI result is instructive: Bayesian optimisation selects $B=1$, recovering WoLF-iHMM. This indicates that when regime changes are rapid and outliers are less impactful, batching is unnecessary. Importantly, BR-iHMM does not degrade performance and adapts accordingly.
>
> We view this as a strength: the method interpolates between standard and doubly-robust inference depending on the data.
>
> We emphasise this point more clearly in the revised version.
>
> **R3.5 Use of existing components.**
>
> We agree the method builds on existing components. The contribution is their principled integration to achieve bounded joint PIF, rather than a heuristic combination.
>
> Specifically, Theorem. 4.1 shows observation-space robustness alone (e.g., WoLF) is insufficient, necessitating a mechanism at the state level. The batched scheme and degenerate scheduling follow directly from this requirement.
>
> We deliberately reuse standard tools (WoLF, HDP, particle learning) for tractability. We view it as a strength that provable robustness is achieved through this construction, and clarify this in the revision.
>
> We emphasise this point in the revised version.
>
> **R3.6 Fixed-size B**
>
> We agree that fixing $B$ limits adaptivity and that online selection is important in practice. We expand this limitation in the revision and outline directions for adaptive batching, e.g., placing a hierarchical prior over $B$ and inferring it online.
>
> **R3.7 Questions**
>
> 1. Covariance update
>
> Please see R3.1 for corrected details.
>
> 2. T-test
>
> We acknowledge the differences on the OFI dataset are not statistically significant (see also R3.4). This is consistent with the method: Bayesian optimisation selects $B=1$, so BR-iHMM reduces to WoLF-iHMM, indicating batching is unnecessary in this regime.
>
> More generally, BR-iHMM matches non-batched performance when batching is not beneficial, while improving performance when outlier-driven instability is present. We clarify this in the revision.
>
> 3. Zero-variance
>
> Thank you for pointing this out. This is a rounding artifact: the standard deviations were small and rounded to zero at two decimal places.
>
> We now report higher precision:
> ```
> batched: Mean RMSE = 46.11293, Std = 0.00301
> wolf: Mean RMSE = 103.80185, Std = 0.01155
> vanilla: Mean RMSE = 101.72198, Std = 0.02636
> beam(single run) : Mean RMSE = 2.85839, Std = -
> ```
>
> 4. On order-B batch robustness
>
> We acknowledge that when consecutive outliers exceed $B$, the guarantee no longer applies directly as multiple batches are affected.
>
> In this case, the model typically allocates such observations to separate states rather than distorting existing ones; persistent outliers are grouped into consistent states across batches. This contains their effect at the level of state allocation rather than parameter drift.
>
> From a practical perspective, this behaviour can be desirable. Persistent sensor failures or systematic corruptions are treated as distinct regimes, which can improve interpretability. In addition, rarely visited states can be pruned over time, limiting long-term impact.
>
> We clarify this limitation and its implications in the revision.

---

> > ### Author Rebuttal · Reviewer_bNnS · 2026-04-02
> >
> > Thank you, this addresses my main concerns.

---

### Official Review · Reviewer_GcDW · 2026-03-13

**Soundness:** 3
**Presentation:** 3
**Significance:** 3
**Originality:** 3
**Overall Recommendation:** 4
**Confidence:** 4

**Summary:**

This paper fuses weighted likelihood generalized Bayes updates and online batched inference with
the infinite Hidden Markov Model to ensure robustness to outliers. Significant improvements have
been demonstrated across a range of tasks.

The idea here is to use a tempered likelihood function to deal with observations far from the predicted.

**Compliance With Llm Reviewing Policy:**

Affirmed.

**Key Questions For Authors:**

Your tempered likelihood function can be closely linked to robust inference methods such as iteratively reweighted least squares (IRLS) and Huber robustness. IRLS in turn is closely related to parametric outlier modelling with models such as Student-t, which can be written as scale mixtures of normals. Can you discuss this link and whether there is an implicit choice of robust distribution that your method may be equivalent to? And how would your method compare with a Student-t noise model, which is easily constructed using mixture  KF approaches (see Chen and Liu (JRSS 2000) and Doucet et al (2000 Stats and C), which should I suggest be your primary citations on R-B PFs)? Can you provide such a comparison? The choice of weighting function W is every bit as informative as choosing a parametric outlier distribution.

Your batched state inference constrains the model significantly. I understand why you do it, but a more principled approach here would be to perform retrospective smoothing in inference, thus allowing a lookahead of B datapoints before decision-making about the state. Such an approach may also induce the type of robustness you require - please comment on this and can you test this hypothesis in theory and/or in simulations?

**Limitations:**

The batched state dynamics limits switching behaviour to an arbitrary grid of times.

**Strengths And Weaknesses:**

Strengths
1. The paper has proposed a convincing method for dealing with outlier updates. The writing
and analysis are both clean and easy to follow.
2. Unlike other probabilistic approaches for handling heavy-tailed outliers, the method presented
here does not seem to sacrifice much computational efficiency.
3. Experiments have been conducted on a range of datasets, showing improve-
ments for the method to applications of infinite hidden markov models.

Weakness
1. The method has introduced additional hyper-parameters for batched inference and likelihood
weighting, but the sensitivity of the performance to the choices has not been extensively
discussed.

---

> ### Author Rebuttal · Authors · 2026-03-31
>
> We thank the reviewer for their comments and for their suggested references. We have now cited [Chen2000] and [Andrieu2002] for our RBPF implementation, in addition to Doucet et al. (2001).
>
> We respond to your questions below.
>
> **R2.1 Implicit choice of robust distribution.**
>
> Thank you for raising these points. We agree that the choice of weighting function W is closely related to classical robust inference methods such as IRLS, Huber losses, and heavy-tailed noise models.
>
> The WoLF update can be interpreted as a reweighted likelihood, where observations are downweighted based on their prediction. In particular, the update can be viewed as an approximation of the robust state-space model introduced in [Ting2007]. A discussion of implicit choice of robust distribution under the IMQ weight can be found in Appendix B.2 in [DuranMartin2024].
>
> An important distinction is that choosing W plays a role similar to selecting a robust loss (or score) or implicit noise model, but without committing to a specific distribution such as Student-t. This point resembles [West1981]
>
> **R2.2 Student-t noise model**
>
> Regarding Student-t noise models, we emphasise that observation-space robustness alone is not sufficient to ensure joint robustness in the iHMM. Even with heavy-tailed predictive densities, outliers can still induce spurious state creation, which is precisely the pathology addressed by our batched state inference. Empirically, we observe that Student-t-based models do not achieve the same doubly-robust behaviour as BR-iHMM. We include a comparison with mixture KF approaches in the revised version that shows this:
>
> https://drive.google.com/file/d/1jybqD9NleJOLdwXtAwqXWTc5_6za4lum/view?usp=drive_link
>
> In particular, we note that having a Student-t predictive, while robust, does not guarantee the doubly-robust condition that prevents the creation of outlier regimes.
>
> We clarify these connections and add the suggested references in the revised version.
>
> **R2.3 Batching vs smoothing**
>
> Thank you for this suggestion. We agree that retrospective smoothing with a lookahead of B observations is a natural alternative.
>
> Our focus, however, is on one-step-ahead prediction in a fully online setting, where decisions must be made without revisiting past states. This motivates the batched formulation, which approximates a limited lookahead while preserving causal, forward-only inference.
>
> In contrast, smoothing-based approaches require performing inference over past observations and state trajectories within each batch. This introduces additional computational cost, and more importantly, departs from the online setting by requiring backward passes over previously processed data. As a result, such methods are less suitable for prequential tasks where predictions must be produced sequentially.
>
> The batched mechanism provides a practical compromise by aggregating evidence over B observations while maintaining forward-only updates and avoiding the combinatorial complexity of exploring multiple state paths.
>
> We agree that smoothing-based methods may provide an alternative route to robustness when online constraints are relaxed.
>
> We expand the discussion to clarify this trade-off and the associated computational considerations in the revised version.
>
>
> **R2.4 Limitation (batching rigidity)**
>
> We thank the reviewer for highlighting the limitation that batching constrains switching behaviour to a fixed grid of times. We agree that this introduces rigidity.
>
> We make this limitation more explicit in the revised version.
>
> **References**
>
> [Chen2000] Chen, R. and Liu, J. S. Mixture kalman filters. Journal of the Royal Statistical Society: Series B (Statistical Methodology), 62(3):493–508, 2000. doi:https://doi.org/10.1111/1467-9868.00246. URL https://rss.onlinelibrary.wiley.com/doi/abs/10.1111/1467-9868.00246.
>
> [Andrieu2002] Andrieu, C. and Doucet, A. Particle filtering for partially observed gaussian state space models. Journal of the Royal Statistical Society Series B: Statistical Methodology, 64 (4):827–836, 2002.
>
> [Ting2007] Ting, Jo-Anne, Evangelos Theodorou, and Stefan Schaal. "Learning an outlier-robust Kalman filter." ECML, 2007.
>
> [DuranMartin2024]: Duran-Martin, G., Altamirano, M., Shestopaloff, A., Sanchez-Betancourt, L., Knoblauch, J., Jones, M., Briol, F.-X., and Murphy, K. P. Outlier-robust kalman filtering through generalised Bayes. In Salakhutdinov, R., Kolter, Z., Heller, K., Weller, A., Oliver, N., Scarlett, J., and Berkenkamp, F. (eds.), Proceedings of the 41st International Conference on Machine Learning, volume 235 of Proceedings of Machine Learning Research, pp. 12138–12171. PMLR, 21–27 Jul 2024. URL https://proceedings.mlr.press/v235/duran-martin24a.html.
>
> [West1981]: West, Mike. "Robust sequential approximate Bayesian estimation." Journal of the Royal Statistical Society Series B: Statistical Methodology 43.2 (1981): 157-166.

---

> > ### Author Rebuttal · Reviewer_GcDW · 2026-04-03
> >
> > The rebuttal answers most of my questions.
> >
> > You picked up a different, later, paper by Andrieu et al (2002) which has a different focus on nonlinear observations rather than Doucet et al  (2000, Stats and Comp) which introduced RB in same year as Chen et al.
> >
> > I thought your batching method required waiting until the current batch is observed, in which case there is a lag? But perhaps you can clarify.
> >
> > My recommendation remains the same

---

> > > ### Author Response · Authors · 2026-04-06
> > >
> > > **Re: citations**. Thank you for pointing this out. We will update the references accordingly, including:
> > >
> > > * Chen, Rong, and Jun S. Liu. "Mixture kalman filters." Journal of the Royal Statistical Society: Series B (Statistical Methodology) 62.3 (2000): 493-508.
> > >
> > > * Doucet, Arnaud, Simon Godsill, and Christophe Andrieu. "On sequential Monte Carlo sampling methods for Bayesian filtering." Statistics and computing 10.3 (2000): 197-208.
> > >
> > > ---
> > >
> > > **Re: batching method and lag.** Thank you for raising this point. There is an important distinction here: our method does not introduce a lag in predictions, but rather in state updates.
> > >
> > > At each timestep, predictions are produced online using the current state and parameter posterior, i.e., there is no fixed-lag smoothing or delayed prediction. Algorithmically, the next $B$ predictions are produced before observing the next batch of observations. The batching mechanism only affects when the state posterior is updated: the latent state is held fixed over a batch of size B, and the update is performed once the batch is observed.
> > >
> > > As a result, the method does not delay predictions, but it does introduce an adaptation lag in state updates, of up to B timesteps. This is precisely the mechanism that enables robustness: aggregating evidence over a batch helps distinguish persistent regime changes from transient outliers.
> > >
> > > An equivalent interpretation is that inference is performed on a re-indexed state-space at the batch level, where each  observation corresponds to a \\((B \times o)\\) block.
> > >
> > > We will clarify this distinction explicitly in the revised version to avoid confusion.

---

### Official Review · Reviewer_FGom · 2026-03-13

**Soundness:** 3
**Presentation:** 2
**Significance:** 2
**Originality:** 2
**Overall Recommendation:** 3
**Confidence:** 3

**Summary:**

This paper studies robust online inference for infinite hidden Markov models in the presence of outliers. Specifically, the paper proposes BR-iHMM that combines robust observation updates with batched state inference to improve robustness in both the observation space and the latent-state space.

**Compliance With Llm Reviewing Policy:**

Affirmed.

**Key Questions For Authors:**

See weaknesses.

**Limitations:**

No, not sufficiently. The paper mainly emphasizes the robustness of the method and its empirical performance, but the discussion of limitations is not very systematic.

**Strengths And Weaknesses:**

The method and theory rely on a relatively strong linear-Gaussian assumption, which may limit applicability in more complex settings.

Besides, the proposed batching mechanism improves robustness but inevitably introduces an adaptivity–robustness trade-off, since larger batch sizes can delay the detection of true regime changes.

In addition, the theoretical results mainly establish bounded posterior influence, which is useful but does not yet provide stronger guarantees on predictive accuracy, consistency, or broader statistical optimality.

The theoretical presentation is also somewhat informal: the assumptions are not clearly organized, and the paper lacks more explicit mathematical bounds on prediction error, estimation error, or other performance metrics.

---

> ### Author Rebuttal · Authors · 2026-03-31
>
> We thank the reviewer for their comments. To our knowledge, this is the first online iHMM framework with provably bounded joint posterior influence across both observation and state spaces. We reply to their questions below.
>
>
> **R1.1 linear-Gaussian assumption**
>
> We agree that the current theoretical results and experiments focus on the LG setting.
> However, as noted in P2 (L88–91), the framework is not restricted to this case and can accommodate a broader class of emission models. In the revised version, we provide additional empirical evidence to support this. Specifically, we include two new examples: (E.1) a switching Gaussian model with unknown mean and variance, and (E.2) a regime-switching stochastic volatility model estimated via SMC. These illustrate that the method extends beyond closed-form LG updates while retaining its robustness properties.
>
> Results for E.1 and E.2 are in: Section B. For results, see
>
> https://drive.google.com/file/d/1jybqD9NleJOLdwXtAwqXWTc5_6za4lum/view?usp=drive_link
>
> https://drive.google.com/file/d/18yqjpNAZNaOqEeiYJBUk4N5NbBCtqUEd/view?usp=drive_link
>
> On the theoretical side, the boundedness result does not fundamentally rely on the linear-Gaussian assumption. While closed-form expressions for the PIF are no longer available, bounded influence can still be established under appropriate conditions on the emission model. For state-space PIF, the boundedness holds as long as the log-likelihood difference for any two state’s posterior predictive is bounded as $|y_t^c| \rightarrow \infty$; this is given by batched WoLF weighting. For batched observation-space PIF, the inductive proof holds as long as the observation PIF is bounded for one outlier.
>
> We add the experiments and corresponding discussion on the theoretical side in the revised manuscript.
>
> **R1.2  Adaptivity-robustness trade-off of batching mechanism.**
>
> Thank you for pointing this out. We agree that the batching mechanism introduces an inherent adaptivity-robustness trade-off.
>
> One of the key contributions of our work is to make this trade-off explicit. While prior approaches focus on improving robustness of updates, increased robustness necessarily reduces sensitivity to regime changes. Our framework exposes this tension through the batch size B, which directly controls the balance between robustness and adaptivity.
>
> We emphasise this point more clearly in the revised version and discuss how the choice of B reflects application-dependent preferences.
>
>
> **R1.3 The results mainly establish bounded posterior influence, which is useful but does not yet provide stronger guarantees on predictive accuracy, consistency, or broader statistical optimality.**
>
> We agree that our theoretical results focus on bounded posterior influence and do not yet establish guarantees on predictive accuracy, consistency, or broader statistical optimality.
>
> We view this as a necessary first step. In the presence of model misspecification and outliers, controlling posterior sensitivity is a prerequisite for stable inference. The PIF provides a formal way to capture this robustness, which is the primary objective of our work.
>
> Extending these results to predictive accuracy and statistical optimality is an important direction. We clarify this limitation and outline it more explicitly in the revised version.
>
> **R1.4 The theoretical presentation is also somewhat informal**
>
> We improve clarity by explicitly organising assumptions and conditions. Specifically, we:
>
> State modelling assumptions in the main text (LG emissions, HDP transitions, posterior factorisation);
>
> Specify conditions on the weighting function $W$ for bounded one-step observation PIF;
>
> State conditions on log-likelihood differences ensuring bounded state-space PIF;
>
> Clarify that conjugacy is specific to LG, while non-conjugate models can be handled via SMC;
>
> Make the proof structure explicit (one-step bound + induction).
>
> **R1.5 on limitations**
>
> We expand the discussion as follows:
>
> (1) The batch size B is fixed a priori and dataset-dependent, motivating adaptive selection.
>
> (2) Theory relies on LG emissions for closed-form PIF, and a general treatment remains open.
>
> (3) The link between bounded PIF and predictive performance is not yet characterised.
>
> (4) We do not provide a formal analysis of the adaptivity–robustness trade-off in B, relying instead on empirical evidence (e.g., Fig. E.8, E.10).

---

> > ### Author Rebuttal · Reviewer_FGom · 2026-04-04
> >
> > The rebuttal addresses most of my concerns through clarification and better framing, but it still does not fully resolve the underlying theoretical and practical limitations.

---

### Decision · Program_Chairs · 2026-04-30

**Decision:**

Accept (regular)

**Comment:**

Authors presented an interesting method for robust online HMMs. The primary novelty of the paper is that they extend the robust Bayesian learning framework for HMMs. This is a non-trivial extension as the original method was presented for i.i.d. distributed data with a continuous prior. Extending this into HMMs requires novel contributions that are significant enough for ICML. Moreover, they present an online algorithm for updating the model as new data arrive, which is also a significant departure from the robust Bayes framework as it is not obvious how to update the model using an online algorithm.  The authors have satisfactorily responded to the reviewers' questions and have demonstrated the utility of their proposed method. Therefore, I vote to accept the paper.